# FAB: A First-Order AB-based Gradient Algorithm for Distributed Bilevel Optimization over Time-Varying Directed Graphs

**Yaoshuai Ma** [1 2]   **Xiao Wang** [3]   **Wei Yao** [4 5]   **Jin Zhang** [1 2 6]

## Abstract

Distributed optimization over time-varying directed graphs has shown promising performance in addressing challenges posed by complex communication constraints in real-world scenarios. In many practical settings, however, the direct application of distributed optimization algorithms encounters additional difficulties, most notably hyperparameter tuning, which our empirical observations suggest can be effectively mitigated by integrating bilevel optimization. Motivated by these findings, we study distributed bilevel optimization over time-varying directed networks, a problem that remains largely unexplored due to the compounded challenges arising from consensus bias in dynamic unbalanced communication and the nested optimization structure. In this work, we propose a fully first-order distributed gradient-based algorithm that integrates the Push–Pull (also known as AB) communication strategy with a value function-based penalty method and establish its non-asymptotic convergence properties. Notably, a simplified variant of our analysis framework for nonconvex single-level distributed optimization establishes a convergence rate for the Push–Pull algorithm, thereby resolving an open question concerning its convergence over time-varying directed graphs. Experiments across hyperparameter tuning, data hyper-cleaning, and reinforcement learning validate FAB's effectiveness and efficiency.

[1]Peng Cheng Laboratory, Shenzhen, China [2]Department of Mathematics, Southern University of Science and Technology, Shenzhen, China [3]School of Computer Science and Engineering, Sun Yat-Sen University, Guangzhou, China [4]School of Artificial Intelligence, Southern University of Science and Technology, Shenzhen, China [5]National Center for Applied Mathematics Shenzhen, Shenzhen, China [6]Detection Institute for Advanced Technology Longhua-Shenzhen, Shenzhen, China. Correspondence to: Wei Yao <yaow@sustech.edu.cn>, Jin Zhang <zhangj9@sustech.edu.cn>.

*Proceedings of the $43^{rd}$ International Conference on Machine Learning*, Seoul, South Korea. PMLR 306, 2026. Copyright 2026 by the author(s).

## 1. Introduction

In recent years, decentralized optimization has attracted increasing attention in machine learning tasks deployed over distributed learning systems. In decentralized optimization, multiple agents cooperate to minimize the sum of their local objective functions while obeying the underlying network connectivity structure (Nedić & Ozdaglar, 2009; Stephen et al., 2011; Duchi et al., 2012; Yuan et al., 2016; Lian et al., 2017), thereby eliminating the need for a central server or the sharing of raw private data (Nedić et al., 2018).

The earliest studies in decentralized optimization focus on *static, undirected* multi-agent networks, where the primary challenge arises from data heterogeneity (Koloskova et al., 2020). This challenge has been effectively addressed by methods such as EXTRA (Shi et al., 2015), $D^2$/Exact-Diffusion (Tang et al., 2018; Yuan et al., 2023), and gradient-tracking based approaches (Koloskova et al., 2021; Pu & Nedić, 2021). For *directed graphs* (or digraphs), which frequently arise in practical applications such as distributed deep learning (Assran et al., 2019), it may not be possible to construct doubly stochastic weight matrices for information fusion (Gharesifard & Cortés, 2010). To address this issue, the push-sum protocol was introduced in Kempe et al. (2003), and subsequently a push-sum based distributed algorithm for directed graphs was developed in (Tsianos et al. (2012). More recently, the alternating Push–Pull (also known as AB) communication strategy, introduced by Xin & Khan (2018) and Pu et al.(2021), employs a row-stochastic matrix $A$ and a column-stochastic matrix $B$ to facilitate information exchange among agents.

To further address complex communication constraints in real-world scenarios, such as communication delays and straggler effects in multi-agent learning systems (Ren & Beard, 2005; Das et al., 2019; Gaydamaka et al., 2024) and satellite networks (Kaushal & Kaddoum, 2017; Han et al., 2023), distributed optimization algorithms over *time-varying directed graphs* have been developed, including subgradient-push (SGP) (Nedić & Olshevsky, 2015) and Push-DIGing (Nedić et al., 2017), which build upon the push-sum protocol. Another important line of work consists of Push–Pull based methods proposed by Saadatniya et al.(2020) and Nedić et al.(2025). It is worth noting that

the existing convergence analyses of Push–Pull based methods over time-varying directed graphs typically rely on the strong convexity assumption of the objective functions.

One major challenge in applying the aforementioned distributed optimization algorithms to learning tasks is hyperparameter tuning, for which bilevel optimization enjoys several advantages over alternative hyperparameter optimization methods (Franceschi et al., 2018; MacKay et al., 2019). However, its application to distributed optimization over time-varying directed graphs remains unclear. Below, we conduct experiments on a linear classification task over time-varying directed networks with a quadratic regularization parameter to evaluate the effectiveness of integrating bilevel optimization in mitigating hyperparameter tuning.

## 1.1. A Motivating Experiment

We conduct experiments on label-corrupted subsets of MNIST (LeCun et al., 1998) over a time-varying directed network consisting of 100 agents. We compare classical single-level distributed optimization baselines, including subgradient-push (Nedić & Olshevsky, 2015), Push-DIGing (Nedić et al., 2017), and AB/Push–Pull (Nedić et al., 2025) combined with grid search, against FAB, our proposed algorithm that integrates AB communication with bilevel optimization, where the loss on a clean validation set serves as the upper-level objective and the corrupted training loss serves as the lower-level objective. As shown in Figure 1, FAB achieves significantly higher test accuracy across different label corruption rates ($cr$) compared to the single-level baselines.

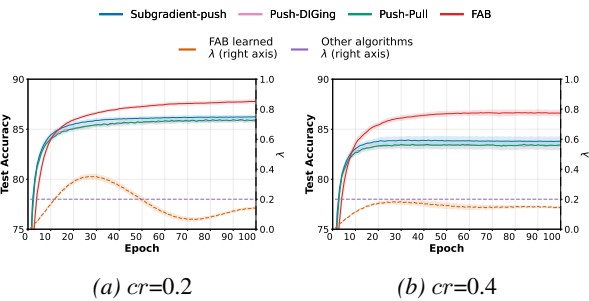

*(a) $cr$=0.2*        *(b) $cr$=0.4*

*Figure 1.* Test accuracy comparison between single-level baselines and FAB (ours) over time-varying directed graphs. The baselines use a fixed regularization parameter $\lambda = 0.2$ selected via grid search, while the dotted curve for FAB (associated with the right vertical axis) depicts the evolution of $\lambda$ during training. The results demonstrate that bilevel optimization enhances the effectiveness of AB/Push–Pull.

## 1.2. Contributions

Motivated by these findings, we study distributed bilevel optimization over time-varying directed networks, a problem that remains largely unexplored due to the compounded

challenges arising from consensus bias induced by dynamic, unbalanced communication and the nested optimization structure. A natural and practically relevant question arises: *Can we develop an efficient algorithm with provable convergence guarantees for distributed bilevel optimization over time-varying directed graphs?*

Addressing this question would advance both the theoretical understanding and practical implementation of decentralized optimization methods across a wide range of machine learning tasks, including meta-learning (Finn et al., 2017; Li et al., 2025) and reinforcement learning (Das et al., 2019; Shen et al., 2025).

Our main contributions are summarized as follows.

(i) We propose a **F**irst-order **AB**/Push-Pull-based **B**ilevel gradient algorithm (termed FAB) for distributed bilevel optimization over time-varying directed graphs. FAB integrates the AB/Push–Pull technique with a value function-based penalty method originally developed for centralized bilevel optimization. To the best of our knowledge, FAB is the first algorithm in the literature to address this problem.

(ii) We establish that FAB converges to a stationary solution at a rate of $\mathcal{O}(K^{-2/3})$, as measured by the hypergradient, when the upper-level objective is nonconvex and the lower-level objective is strongly convex. Notably, a simplified variant of our analysis framework for nonconvex single-level distributed optimization yields a convergence rate for the AB/Push–Pull algorithm, thereby resolving an open question regarding its convergence over time-varying directed graphs.

(iii) We evaluate the effectiveness of FAB through extensive experiments spanning hyperparameter tuning, data hyper-cleaning, and reinforcement learning. These evaluations cover diverse application domains, including computer vision (CV) and natural language processing (NLP). The results consistently demonstrate that FAB outperforms state-of-the-art algorithms, achieving faster convergence and improved robustness to statistical heterogeneity and environmental noise.

## 1.3. Related Work and Challenges

Due to space limitations, we focus on the most relevant work and discuss the main challenges. For a more detailed discussion, we refer the reader to the appendix.

**AB/Push-Pull on single-level optimization.** For static directed networks, there has been extensive work adapting the Push–Pull technique, including (Xin & Khan, 2018; Xin et al., 2019; 2020; Pu et al., 2021; Zhao & Liu, 2024; You & Pu, 2024). While early studies primarily focused on strongly convex objectives, more recent contributions (Liang et al., 2025; You & Pu, 2025) have extended the analysis

to nonconvex settings. However, extending these analyses to time-varying directed networks remains challenging. A critical difficulty is that analyses for static topologies rely on the time invariance of the root eigenvector $\pi$[1]. This invariance plays a crucial role in establishing convergence in static settings (see, e.g., Lemma 1 in Xin & Khan, 2018, Lemma 1 in Pu et al., 2021, and Proposition 1 in Liang et al., 2025). Although AB/Push–Pull based methods over time-varying directed networks have been proposed (Saadatniaki et al., 2020; Nguyen et al., 2024; Nedić et al., 2025), their convergence analyses are largely restricted to the strongly convex setting.

**Distributed bilevel optimization.**  Despite the progress in centralized bilevel optimization, designing provably efficient algorithms for distributed bilevel optimization is far from a straightforward extension of their centralized counterparts, as it requires simultaneously handling data heterogeneity and achieving consensus among multiple agents. Existing algorithms have primarily focused on *static networks* (Yang et al., 2022; Zhu et al., 2024; Chen et al., 2025b; Wang et al., 2025). Among them, the first three rely on second-order derivative information, whereas the latter adopts a first-order gradient-based approach by leveraging a value function-based penalty method.

**Technical challenges.**  Since average consensus is essential for decentralized optimization, the primary challenges in integrating the AB/Push–Pull technique with bilevel optimization arise from consensus bias caused by dynamic, unbalanced communication and from the inherent nested optimization structure. When the upper-level objective is nonconvex, these challenges become twofold.

*1) The convergence rate of the AB/Push–Pull algorithm over time-varying directed graphs remains unknown, even in the single-level distributed optimization setting.* Unlike strongly convex settings, where the objective function induces a global contraction property that suppresses deviations, nonconvex objectives lack such a mechanism. The situation is further complicated in dynamic directed network settings, where the absence of a time-invariant eigenvector causes the consensus reference point to *drift* over time.

*2) A large penalty parameter in the bilevel optimization reformulation can amplify consensus errors.* To approximately converge to a stationary point as measured by the hypergradient, the penalty parameter typically needs to be large. However, an excessively large penalty parameter can significantly amplify consensus errors, potentially leading to divergence. This creates a difficult trade-off between enforcing a large penalty parameter and controlling consensus

---

[1]Defined as the unique nonnegative unit left eigenvector associated with eigenvalue 1, with positive entries corresponding to the root classes of the graph (You & Pu, 2025, Definition 2.2).

errors, especially in nonconvex dynamic settings.

**Notation.**  For the time-varying directed graphs $\{\mathcal{G}^k = ([n], \mathcal{E}^k)\}_{k \geq 0}$, let $\mathcal{N}_{i,\text{in}}^k$ and $\mathcal{N}_{i,\text{out}}^k$ denote the in-neighbor and out-neighbor sets of agent $i$ at iteration $k$, respectively.

**Definitions.** (1) A graph is *strongly connected* if there exists a directed path between any pair of distinct nodes.

(2) A matrix is *row-* (resp. *column-*) *stochastic* if it is nonnegative and each of its rows (resp. column) sums to 1.

(3) A mixing row-stochastic matrix $A = [a_{ij}]_{n \times n}$ is compatible with the directed graph $\mathcal{G}^k = ([n], \mathcal{E}^k)$ if $a_{ij} > 0$ for all $j \in \mathcal{N}_{i,\text{in}}^k \cup \{i\}$, and $a_{ij} = 0$ otherwise. Similarly, a column-stochastic matrix $B = [b_{ij}]_{n \times n}$ is compatible with $\mathcal{G}^k$ if $b_{ji} > 0$ for all $j \in \mathcal{N}_{i,\text{out}}^k \cup \{i\}$, and $b_{ji} = 0$ otherwise. Here, $[n] := \{1, \ldots, n\}$.

## 2. Methodology

We consider a decentralized bilevel optimization problem distributed among $n$ agents over time-varying directed networks $\{\mathcal{G}^k = ([n], \mathcal{E}^k)\}_{k \geq 0}$, formulated as follows:

$$
\min_{x \in \mathbb{R}^{d_x}} \mathcal{F}^*(x) := F(x, y^*(x)) = \frac{1}{n} \sum_{i=1}^n f_i(x, y^*(x)),
$$
$$
\text{s.t. } y^*(x) := \underset{y \in \mathbb{R}^{d_y}}{\arg\min} \ G(x, y) = \frac{1}{n} \sum_{i=1}^n g_i(x, y). \tag{1}
$$

Here, the functions $f_i$ and $g_i$ represent local objective functions accessible only to agent $i$. We assume that each $g_i$ is strongly convex with respect to $y$, whereas $f_i$ may be nonconvex in both $x$ and $y$, but with a lower bound.

To develop a first-order gradient-based algorithm, we adopt the *value function approach* (Ye & Zhu, 1995), which reformulates the original problem (1) into an equivalent constrained optimization problem:

$$
\min_{x,y} \ F(x, y) \quad \text{s.t.} \quad G(x, y) - \min_z G(x, z) \leq 0. \tag{2}
$$

Seeking a stationary solution of this constrained formulation naturally leads to a penalty-based reformulation (Kwon et al., 2023; Wang et al., 2025; Yang et al., 2025):

$$
\min_{x,y} \ F(x, y) + \lambda\big(G(x, y) - \min_z G(x, z)\big), \tag{3}
$$

where the penalty parameter $\lambda > 0$. Importantly, under certain smoothness assumptions, solving problem (3) serves as a reliable proxy for solving (1) when $\lambda$ is sufficiently large (Kwon et al., 2023; Chen et al., 2025a); cf. Lemma D.1 for details. A large $\lambda$, with a properly chosen small stepsize, does not amplify consensus or tracking errors in our setting. By the algorithmic updates in (6) and (7), it amplifies the smoothness constant of the penalty function in (3), which

requires a smaller stepsize. Once the stepsize is reduced accordingly, both errors remain bounded. Besides improving the approximation quality of the bilevel formulation, a sufficiently large $\lambda$ makes the min-max objective in (4) not only strongly concave in $z$, but also strongly convex in $y$. This was also recognized and used by Kwon et al. (Kwon et al., 2023); see also (22) in Appendix D.1, where the strong convexity in $y$ ensures the uniqueness of the minimizer in the definition of $y_\lambda^*(x)$.

### 2.1. The Proposed Method

Since $\lambda > 0$, we can move the minimization operator forward, yielding the following equivalent formulation:

$$\min_{x,y} \max_z \ \mathcal{L}(x,y,z) := F(x,y) + \lambda\big(G(x,y) - G(x,z)\big),$$

where $z$ is an auxiliary variable that tracks $y^*(x)$. This reformulation is well suited for decentralized optimization, as the global objective admits a decomposition into local penalty functions. Specifically,

$$\min_{x,y} \max_z \ \mathcal{L}(x,y,z) = \frac{1}{n}\sum_{i=1}^n \mathcal{L}_i(x,y,z), \qquad (4)$$

where each local penalty function is defined as

$$\mathcal{L}_i(x,y,z) := f_i(x,y) + \lambda\big(g_i(x,y) - g_i(x,z)\big). \quad (5)$$

Now we can introduce the intuition behind our proposed method (termed FAB) in one sentence. FAB integrates the AB/Push–Pull technique with gradient descent ascent updates to solve the distributed optimization problem (4), rather than directly solving (1). Algorithm 1 summarizes the FAB procedure, which consists of three main steps. At iteration $k$, each agent $i$ maintains two groups of variables: $(x_i^k, y_i^k, z_i^k)$ and $(t_{x,i}^k, t_{y,i}^k, t_{z,i}^k)$, where the former are the *decision variables* and the latter are referred to as the *tracking variables*. Each agent $i$ then sends its vectors $(x_i^k, y_i^k, z_i^k)$ and $(b_{ji}^k t_{x,i}^k, b_{ji}^k t_{y,i}^k, b_{ji}^k t_{z,i}^k)$ to its out-neighbor $j \in \mathcal{N}_{i,\text{out}}^k$.

The local gradients in (7) require only first-order oracles for $f_i$ and $g_i$; FAB does not compute Hessian-vector products, Hessian inverses, or unrolled hypergradients.

**Step 1: Decision variable update.** In this step, each agent *pulls* the decision variables from its in-neighbors. Specifically, we use a time-varying row-stochastic matrix $A^k = [a_{ij}^k]_{n \times n}$, compatible with the underlying digraph $\mathcal{G}^k$, to update the decision variables as follows:

$$\begin{cases} x_i^{k+1} = \sum_{j \in \mathcal{N}_{i,\text{in}}^k \cup \{i\}} a_{ij}^k x_j^k - \eta_x^k t_{x,i}^k, \\ y_i^{k+1} = \sum_{j \in \mathcal{N}_{i,\text{in}}^k \cup \{i\}} a_{ij}^k y_j^k - \eta_y^k t_{y,i}^k, \\ z_i^{k+1} = \sum_{j \in \mathcal{N}_{i,\text{in}}^k \cup \{i\}} a_{ij}^k z_j^k - \eta_z^k t_{z,i}^k, \end{cases} \qquad (6)$$

where $\eta_x^k$, $\eta_x^k$, and $\eta_x^k$ denote the learning rates (step sizes).

---

**Algorithm 1** **F**irst-order **AB**/Push-Pull-based **B**ilevel gradient algorithm (FAB)

---

1: **Input:** Initial variables $\{(x_i^0, y_i^0, z_i^0)\}_{i=1}^n$; Initial variables $\{(t_{x,i}^0, t_{y,i}^0, t_{z,i}^0) = (d_{x,i}^0, d_{y,i}^0, d_{z,i}^0)\}_{i=1}^n$, computed via (7); Sequences of row-stochastic matrices $\{A^k\}$ and column-stochastic matrices $\{B^k\}$; Learning rates $\{(\eta_x^k, \eta_y^k, \eta_z^k)\}$; Penalty parameter $\lambda$; Total number of iterations $K$.
2: **for** $k = 0, 1, \ldots, K-1$ **do**
3:    **for** each agents $i \in \{1, \ldots, n\}$ in parallel **do**
4:       **Decision variable update:**
5:       compute $\{(x_i^{k+1}, y_i^{k+1}, z_i^{k+1})\}$ by (6);
6:       **Local gradient evaluation:**
7:       compute $\{(d_{x,i}^{k+1}, d_{y,i}^{k+1}, d_{z,i}^{k+1})\}$ via (7);
8:       **Tracking variable update:**
9:       compute $\{(t_{x,i}^{k+1}, t_{y,i}^{k+1}, t_{z,i}^{k+1})\}$ according to (8);
10:    **end for**
11: **end for**

---

**Step 2: Local gradient evaluation.** In this step, each agent computes the local gradients at the most recent local decision variables, consistent with the gradient descent ascent updates:

$$\begin{cases} d_{x,i}^{k+1} = \nabla_x \mathcal{L}_i(x_i^{k+1}, y_i^{k+1}, z_i^{k+1}), \\ d_{y,i}^{k+1} = \nabla_y \mathcal{L}_i(x_i^{k+1}, y_i^{k+1}, z_i^{k+1}), \\ d_{z,i}^{k+1} = -\nabla_z \mathcal{L}_i(x_i^{k+1}, y_i^{k+1}, z_i^{k+1}). \end{cases} \qquad (7)$$

**Step 3: Tracking variable update.** This step can be viewed as a *push* step, in which we employ gradient tracking and a time-varying column-stochastic matrix $B^k = [b_{ij}^k]_{n \times n}$, compatible with the underlying digraph $\mathcal{G}^k$, to update the tracking variables as follows:

$$\begin{cases} t_{x,i}^{k+1} = \sum_{j \in \mathcal{N}_{i,\text{in}}^k \cup \{i\}} b_{ij}^k t_{x,j}^k + d_{x,i}^{k+1} - d_{x,i}^k, \\ t_{y,i}^{k+1} = \sum_{j \in \mathcal{N}_{i,\text{in}}^k \cup \{i\}} b_{ij}^k t_{y,j}^k + d_{y,i}^{k+1} - d_{y,i}^k, \\ t_{z,i}^{k+1} = \sum_{j \in \mathcal{N}_{i,\text{in}}^k \cup \{i\}} b_{ij}^k t_{z,j}^k + d_{z,i}^{k+1} - d_{z,i}^k. \end{cases} \qquad (8)$$

## 3. Convergence Analysis

### 3.1. Assumptions

**Assumption 3.1** (Communication Graph). For each $k$, the directed graph $\mathcal{G}^k = ([n], \mathcal{E}^k)$ is strongly connected.

**Assumption 3.2** (Compatible Mixing Matrices). For each $k$, the nonnegative mixing matrices $A^k$ and $B^k$ used in FAB are row-stochastic and column-stochastic, respectively, and are compatible with the directed graph $\mathcal{G}^k$. Moreover, there exist constants $a, b > 0$ such that all nonzero entries of $A^k$ and $B^k$ satisfy $a_{ij}^k \geq a$ and $b_{ij}^k \geq b$ for all $k \geq 0$.

Assumption 3.1 and the stochasticity conditions in Assumption 3.2 are standard in the analysis of AB/Push–Pull methods (Xin & Khan, 2018; Pu et al., 2021; Liang et al., 2025). The *uniform lower bound condition* in Assumption 3.2 is also widely adopted in decentralized optimization over time-varying directed graphs (Saadatniaki et al., 2020; Nguyen et al., 2024; Nedić et al., 2025), as it ensures that the interactions among communicating agents remain sufficiently strong, independently of $k$. Similar to Nedić et al. (2025), Assumption 3.1 can be relaxed to a $C$-strong-connected graph sequence, i.e., there exists an integer $C \geq 1$ such that the graph with edge set $\mathcal{E}_{C,k} := \cup_{t=kC}^{(k+1)C-1} \mathcal{E}^t$ is strongly connected for all $k \geq 0$. The proof for this extension mirrors Remark 6.2 of Nedić et al. (2025). This weaker model is often easier to satisfy in practice because directed links are allowed to accumulate over a finite window, meaning that intermittent or asymmetric packet drops at individual iterations need not destroy global connectivity. Specifically, in practical networks, accumulating these sparse links over a moderately small window $C$ easily forms directed rings or overlapping cycles that span all nodes. Thus, temporal accumulation naturally guarantees global strong connectivity, making the algorithm highly robust under realistic network volatility.

**Assumption 3.3** (Smoothness). For each $i \in [n]$,
(i) $f_i(x, y)$ is $L_{f,0}$-Lipschitz continuous in $y$, $L_{f,1}$-smooth (i.e., its gradient is $L_{f,1}$-Lipschitz continuous), and has $L_{f,2}$-Lipschitz continuous Hessian;
(ii) $g_i(x, y)$ is $\mu$-strongly convex in $y$, $L_{g,1}$-smooth, and $L_{g,2}$-Lipschitz continuous Hessian.

Assumption 3.3 is standard in the study of distributed bilevel optimization with strongly convex lower-level problems (Yang et al., 2025; Wang et al., 2025). Notably, all components of this assumption are primarily used to guarantee the $L$-smoothness of $\mathcal{F}^*(x) := F(x, y^*(x))$. Consequently, for the single-level distributed optimization problem

$$\min_x F(x) := \frac{1}{n} \sum_{i=1}^n f_i(x), \qquad (9)$$

Assumption 3.3 reduces, under our consideration, to the standard $L_{f,1}$-smoothness assumption on $f_i(x)$.

### 3.2. Convergence Rate of FAB

Although FAB is designed to directly solve problem (4), its theoretical analysis ultimately focuses on the hypergradient $\nabla \mathcal{F}^*(x)$ of problem (1), which is Lipschitz continuous under Assumption 3.3. See Appendix D for additional properties. We now state the convergence rate of FAB.

**Theorem 3.4.** *Suppose that Assumptions 3.1–3.3 hold. For a given $K$, with properly chosen step sizes $\eta_x^k, \eta_y^k, \eta_z^k = \mathcal{O}(K^{-1/3})$ and a penalty parameter $\lambda = \mathcal{O}(K^{1/3})$, as*

*specified in Theorem D.13, the sequence generated by FAB in Algorithm 1 satisfies*

$$\min_{0 \leq k < K-1} \|\nabla \mathcal{F}^*(\bar{x}^k)\|^2 = \mathcal{O}\left((ab)^{-n} K^{-\frac{2}{3}}\right), \qquad (10)$$

*where $\bar{x}^k := \frac{1}{n} \sum_{i=1}^n x_i^k$, and $a, b > 0$ are the constants defined in Assumption 3.2.*

**Impact of the penalty parameter $\lambda$.** Increasing $\lambda$ improves how well problem (3) approximates problem (1), but it also amplifies consensus errors, thereby necessitating smaller learning rates and potentially slowing convergence in the analysis. With an appropriate but fixed choice of $\lambda = \mathcal{O}(K^{1/3})$, one can guarantee a convergence rate with $\mathcal{O}(K^{-2/3})$ dependence on $K$. It may be possible to further improve this convergence rate by gradually increasing $\lambda$, following recent advances in centralized bilevel optimization (Kwon et al., 2023; Chen et al., 2025a). Indeed, as previously demonstrated in Table 2 of Appendix B.4, a larger $\lambda$ imposes stricter requirements on both the consensus and gradient-tracking errors, forcing the algorithm to maintain a tighter consensus.

**Impact of the network size $n$.** The factor $(ab)^{-n}$ in the convergence rate suggests that linear speedup with respect to the number of agents may not be guaranteed for FAB. This behavior is not unexpected, as it reflects the worst-case analysis inherent to general time-varying directed graphs. Indeed, similar factors have been observed in the theoretical analysis of several distributed optimization algorithms over time-varying directed networks. For instance, in the case of subgradient-push (Nedić & Olshevsky, 2015), a related discussion appears following Theorem 2. Likewise, for the Push-Pull (TV-$\mathcal{AB}$) algorithm, comparable factors can be identified in the constant $m$ in Lemma 4 and in the multistep contraction results for row-stochastic and column-stochastic matrices presented in Lemma 7 and Corollary 2 of Saadatniaki et al. (2020), although such effects do not always appear explicitly in the stated convergence rates. Finally, we conduct experiments to evaluate the impact of the network size $n$. As shown in Figure 6(a), the convergence performance degrades as $n$ increases, but the degradation is not exponential.

We now turn to the local consensus properties of FAB.

**Proposition 3.5.** *Under the same conditions as Theorem 3.4, the average consensus errors satisfy*

$$\begin{cases} \min_{0 \leq k < K-1} \frac{1}{n} \sum_{i=1}^n \|x_i^k - \bar{x}^k\|^2 = \mathcal{O}\left((ab)^{-n} K^{-1}\right), \\ \min_{0 \leq k < K-1} \frac{1}{n} \sum_{i=1}^n \|y_i^k - \bar{y}^k\|^2 = \mathcal{O}\left((ab)^{-n} K^{-1}\right), \\ \min_{0 \leq k < K-1} \frac{1}{n} \sum_{i=1}^n \|z_i^k - \bar{z}^k\|^2 = \mathcal{O}\left((ab)^{-n} K^{-1}\right), \end{cases}$$
(11)

*where $\bar{y}^k := \frac{1}{n} \sum_{i=1}^n y_i^k$, and $\bar{z}^k := \frac{1}{n} \sum_{i=1}^n z_i^k$.*

The difference in the dependence on $K$ between the average consensus errors and the convergence rate arises from the

coefficient $\lambda = \mathcal{O}(K^{1/3})$ of $\mathbf{V}_D^k$ in the following descent property of the Lyapunov function:

$$
\begin{aligned}
&\left\|\nabla_x \mathcal{F}^*\left(\bar{x}^k\right)\right\|^2 + \frac{8\underline{c}n}{5a^n}\mathcal{C}_{b,3}\lambda\mathbf{V}_D^k \\
&\leq \frac{4\mathcal{C}_{gap}}{\lambda^2} + \frac{16\left(\Phi^{(b)}(\hat{x}^k) - \Phi^{(b)}(\hat{x}^{k+1})\right)}{(ab)^n \eta_x^k},
\end{aligned} \tag{12}
$$

whose proof is provided in Theorem D.13. Here, $\underline{c}$, $\mathcal{C}_{b,3}$ and $\mathcal{C}_{gap}$ are time-invariant constants, $\Phi^{(b)}$ denotes the Lyapunov function defined in (34), $\mathbf{V}_D^k$ represents the cumulative consensus error defined in (30), which controls the average consensus errors.

### 3.3. Convergence Rate of Single-Level Push–Pull in the Nonconvex Setting

When the bilevel optimization problem (1) reduces to the single-level distributed optimization problem (9), FAB reduces to the standard Push-Pull (TV-$\mathcal{AB}$) algorithm (Saadatniaki et al., 2020; Nedić et al., 2025). To the best of our knowledge, there is no rigorous convergence analysis establishing its convergence rate over time-varying directed graphs in the nonconvex setting. In this section, we show that a simplified variant of our analysis framework yields such a convergence rate.

**Theorem 3.6.** *For the problem (9), where each $f_i$ may be nonconvex, suppose that Assumptions 3.1 and 3.2 hold and that each $f_i$ is $L_{f,1}$-smooth. Then, with a properly chosen step size $\eta_x^k = \mathcal{O}(1)$ specified in Theorem E.7, the sequence generated by the Push-Pull algorithm satisfies the following convergence rate and average consensus error bounds:*

$$
\min_{0\leq k<K-1}\|\nabla F(\bar{x}^k)\|^2 = \mathcal{O}\left((ab)^{-n}K^{-1}\right), \quad (13a)
$$

$$
\min_{0\leq k<K-1}\frac{1}{n}\sum_{i=1}^{n}\|x_i^k - \bar{x}^k\|^2 = \mathcal{O}\left((ab)^{-n}K^{-1}\right). \quad (13b)
$$

A proof sketch and the complete proof are provided in Appendix E. Since there is no amplification effect arising from a large penalty parameter (as in FAB), the step size can be chosen to be constant and independent of $K$. Consequently, we establish that the AB/Push–Pull algorithm converges to a stationary solution at a rate of $\mathcal{O}((ab)^{-n}K^{-1})$, where the $\mathcal{O}(K^{-1})$ dependence on $K$ matches that of centralized gradient descent in the nonconvex setting.

## 4. Experiments

In this section, we evaluate the practical performance of FAB using three examples[2]: distributed hyperparameter tuning[3], distributed policy evaluation for Reinforcement

---

[2]The source code is available at `https://github.com/studma/FAB`.

[3]See Section 1.1 and Appendix B.

Learning (RL) and distributed data hyper-cleaning. Specifically, FAB is benchmarked against the following methods tailored for time-varying directed networks: SGP/Push-SGD (Nedić & Olshevsky, 2016; Assran et al., 2019), Push-SAGA (Qureshi et al., 2022), Push-ASGD (Chen et al., 2025c), and AB/Push-Pull (Saadatniaki et al., 2020; Nedić et al., 2025).

*Dynamic topology:* Following Chen et al. (2025c), the time-varying directed network topology is based on the Erdős-Rényi (ER) model and directed rings. A visualization of these topologies is provided in Figure 17 in the Appendix, where the edges in an augmented ER graph are generated with a probability $\nu$. This parameter influences the network's sparsity (a smaller $\nu$ results in a sparser network).

Additional experimental results and further details on the experimental setup can be found in Appendix B and Appendix C. We also report network-size, model-size, and communication-volatility diagnostics to distinguish the pessimistic constants in the worst-case theory from the observed behavior of the local FAB update.

### 4.1. Distributed Policy Evaluation for RL

Following Yang et al.(2022) and Zhu et al.(2024), but considering time-varying directed networks, we apply the proposed FAB to conduct policy evaluation in multi-agent RL by attacking Bellman equations. Let $\mathcal{S}$ be the state space, $\pi$ be the policy of interest, and $V^\pi(s)$ denotes the value of being in state $s$ under policy $\pi$. As in Wang et al. (2017), we approximate the value of each state using a linear map of its feature $\phi_s \in \mathbb{R}^d$, where $d < |\mathcal{S}|$ to reduce the dimension. To learn this linear map, we solve a regularized Bellman minimization problem in (16), which can be reformulated as a distributed bilevel optimization problem. To adapt the single-level methods to this problem, we integrate them with the standard Iterative Differentiation (ITD) technique (Franceschi et al., 2018; Grazzi et al., 2020) for bilevel optimization, and refer to the resulting baselines as their double-loop (DL) variants. Further details are provided in Appendix C.3.

Part of the empirical results are shown in Figure 2. Additional results on the effect of noise level and network connectivity, including runtime efficiency, are provided in Figure 11 and Figures 12. The proposed FAB consistently outperforms the adapted double-loop baseline across all connectivity settings ($\nu \in \{0.1, 0.3, 0.5\}$) and noise levels ($\omega \in \{1, 2, 3\}$). Although FAB starts slower initially, it converges significantly faster and achieves lower losses, demonstrating its efficiency and robustness against changes in network topology and gradient noise. This superior efficiency is also reflected in the overall communication burden: while ITD baselines may communicate fewer variables per outer update, their nested inner trajectories necessitate se-

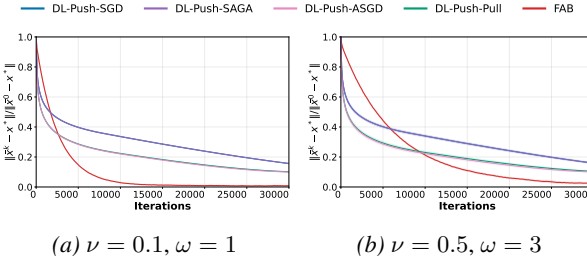

*Figure 2.* Performance of distributed policy evaluation for reinforcement learning over a dynamic network topology, under varying edge connection parameter $\nu$ and gradient noise level $\omega$.

quential synchronization before a hypergradient estimate is available; see Table 4 of Appendix B.4 for the corresponding communication diagnostics.

### 4.2. Distributed Data Hyper-cleaning

We consider a data hyper-cleaning problem (Shaban et al., 2019) over time-varying directed networks. In the decentralized setting, multiple agents cooperate to learn a robust classifier from label-corrupted data by leveraging a small, clean validation set for re-weighting.

We test the performance of the proposed algorithm across two distinct application domains: (i) *Computer Vision (CV):* Training a lightweight Multi-Layer Perceptron (MLP) on the Fashion-MNIST dataset and evaluating performance on standard image classification benchmarks; (ii) *Natural Language Processing (NLP):* Fine-tuning a large-scale BERT model (Devlin et al., 2019) (approx. 110M parameters) on the IMDB (Maas et al., 2011) dataset to enhance its capability in complex sentiment analysis scenarios. Detailed experimental configurations are provided in Appendix C.4.

**Training MLP on Fashion-MNIST.** In this experiment, we employ a two-layer MLP with 203,530 parameters, trained on the Fashion-MNIST dataset. We evaluate the performance of our proposed algorithm and the baselines by varying the label corruption rate $cr$ and the degree of data heterogeneity $\rho$. Additional results regarding the impact of $\nu$ are provided in Figure 14.

*Effect of $cr$:* As demonstrated in Figure 3, FAB consistently outperforms all baselines in terms of both epoch and time-to-accuracy efficiency. Notably, as the corruption rate $cr$ increases, the performance of single-level algorithms degrades significantly, highlighting the advantages of FAB.

*Effect of $\rho$:* The dataset is partitioned using a Dirichlet non-IID scheme with the concentration parameter $\rho \in \{0.1, 0.5, 1.0\}$. As data heterogeneity increases (i.e., $\rho$ decreases), Figure 4 (a)-(c) shows that all algorithms experience performance degradation, with FAB exhibiting superior

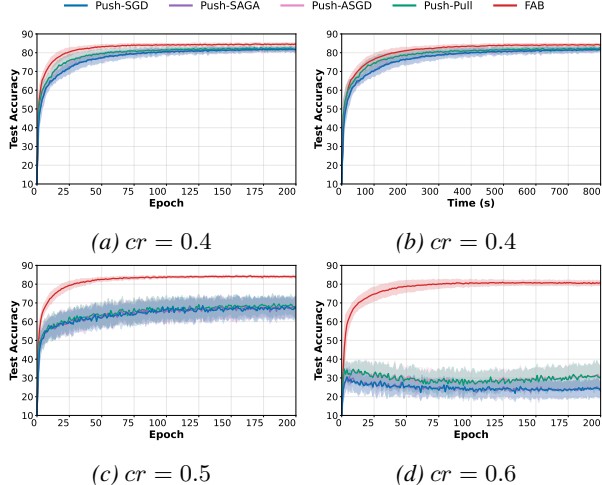

*Figure 3.* Performance on Fashion-MNIST under varying label corruption rates. Subfigures (a), (c), and (d) illustrate test accuracy vs. epochs under corruption rates $cr \in \{0.4, 0.5, 0.6\}$, respectively, while subfigure (b) presents test accuracy vs. runtime. Fixed parameters: connectivity $\nu = 0.5$ and heterogeneity $\rho = 0.5$.

robustness and milder degradation. Moreover, Figure 4(d) demonstrates that the degree of heterogeneity has a negligible impact on peak memory consumption.

**Fine-tuning BERT on IMDB.** We further evaluate the algorithms on the IMDB sentiment analysis task by fine-tuning a BERT model. Although this is a binary classification task, the distributed setting introduces unique challenges due to severe label skew induced by heterogeneous data partitioning. Specifically, agents that possess data from a single class cause local learning to be ineffective. As a result, the initial test accuracy starts near 0% rather than the 50% baseline, and is accompanied by severe oscillations. The results are shown in Figure 5, where the corruption rate $cr$, network connectivity $\nu$, and data heterogeneity $\rho$ are varied. The proposed FAB exhibits superior accuracy and efficiency, along with enhanced robustness on this complex and large-scale task. Moreover, we also evaluate the runtime performance in Figure 15, which indicates that the baselines exhibit a slight speed advantage under low corruption rates ($cr = 0.1, 0.2$) compared to FAB. Although the theoretical guarantees of FAB in Theorem 3.4 rely on the assumption of lower-level strong convexity, these practical BERT experiments demonstrate that our algorithm maintains strong empirical performance and robustness even in highly nonconvex scenarios.

### 4.3. Ablation Study

Next, we evaluate the impact of key factors on the algorithms' performance, including the number $n$ of agents, the minimum values $a$ and $b$ of the nonzero entries in the non-

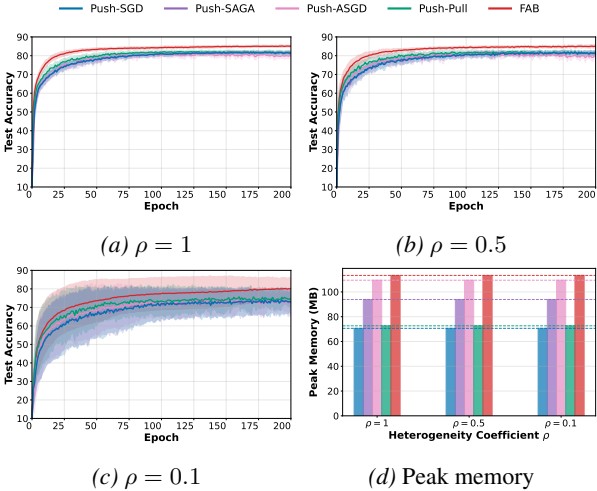

*(a)* $\rho = 1$  *(b)* $\rho = 0.5$

*(c)* $\rho = 0.1$  *(d)* Peak memory

*Figure 4.* Performance on Fashion-MNIST under varying degrees of data heterogeneity $\rho$. Subfigures (a)-(c) illustrate the test accuracy, while subfigure (d) presents the peak memory consumption of the compared algorithms. Fixed parameters: network connectivity $\nu = 0.1$ and label corruption rate $cr = 0.4$.

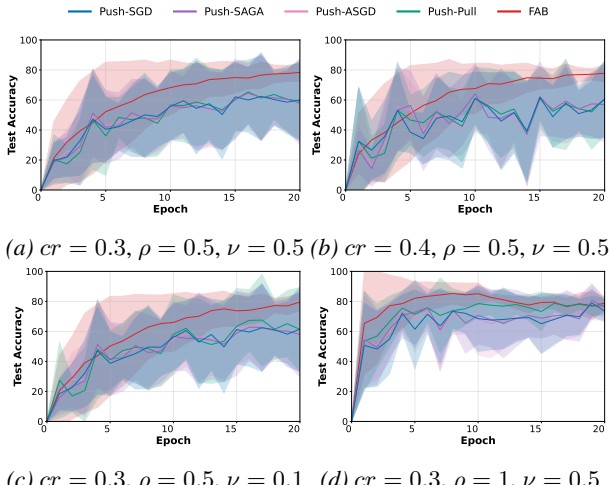

*(a)* $cr = 0.3, \rho = 0.5, \nu = 0.5$  *(b)* $cr = 0.4, \rho = 0.5, \nu = 0.5$

*(c)* $cr = 0.3, \rho = 0.5, \nu = 0.1$  *(d)* $cr = 0.3, \rho = 1, \nu = 0.5$

*Figure 5.* Performance of fine-tuning BERT on IMDB under varying settings. The impact of each parameter is illustrated as follows: corruption rate $cr$ in (a) and (b); network connectivity $\nu$ in (a) and (c); and data heterogeneity $\rho$ in (a) and (d).

negative mixing matrices $A^k$ and $B^k$, as well as the problem dimension. Furthermore, we compare the proposed FAB with other potential integrations of distributed optimization techniques and bilevel optimization methods for distributed bilevel optimization over time-varying directed graphs. Additional results regarding the sensitivity analysis of FAB with respect to the penalty parameter and step sizes are provided in Appendix B.4.

**Impact of the network size and mixing matrix weights.** In Figure 6, we evaluate FAB's performance by varying the number $n$ of agents and the minimum values of the nonzero entries in the dynamic mixing matrices. Figure 6(a) shows that the convergence performance degrades as $n$ increases, suggesting that linear speedup may not be guaranteed for FAB over time-varying directed graphs. Furthermore, Figure 6(b) demonstrates that reducing the minimum values $a$ and $b$ results in a deceleration of convergence.

The milder performance decline in Figure 6(a) suggests that the $(ab)^{-n}$ scaling factor in our analysis is primarily driven by pathological worst-case mixing sequences, rather than the typical dynamic topologies encountered in practice. Together with the model-size study and the larger-network evaluations in Figure 16 and Table 5, these findings demonstrate the practical scalability of our algorithm, complementing the conservative constants in the theorems. Indeed, our large-scale empirical results indicate that real-world networks naturally avoid these worst-case connectivity scenarios, allowing FAB to scale effectively as the network size $n$ increases.

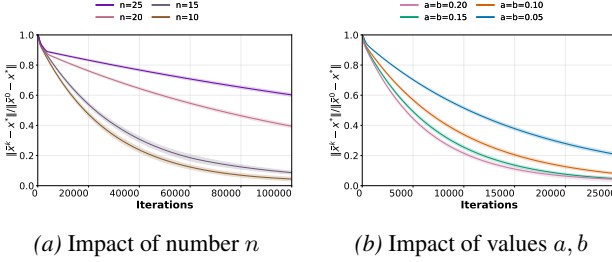

*(a)* Impact of number $n$  *(b)* Impact of values $a, b$

*Figure 6.* Performance of FAB on distributed policy evaluation for RL. Here, $n$ is the number of agents, and $a$ and $b$ represent the minimum values of the nonzero entries in the mixing matrices $A^k$ and $B^k$, respectively.

**Impact of the model size.** In Figure 7, we evaluate the scalability of both Push-Pull and FAB on the Fashion-MNIST data hyper-cleaning task using an MLP architecture, by varying the model size. Both Push-Pull and FAB exhibit approximately linear scaling in communication cost per iteration, peak memory footprint, and runtime per epoch. Moreover, since the weighting variable introduced in FAB as the upper-level variable is independently defined for each sample in the training dataset, the local gradient with respect to the weighting variable for each agent depends only on its own local weighting variable (i.e., the other components of the $x$-gradient are 0). As a result, there is no need to exchange local weighting information, and therefore, FAB exhibits approximately $2\times$ the communication cost of the Push-Pull method.

Accordingly, Appendix B.4 reports the cumulative communication required to reach target accuracy/error levels, complementing the per-iteration cost.

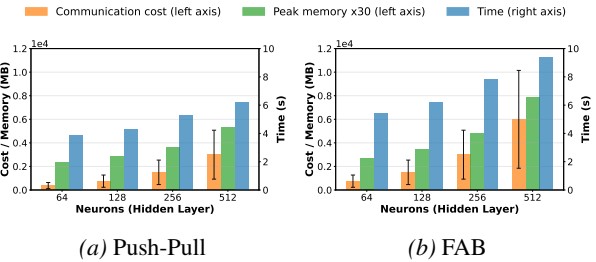

*(a)* Push-Pull       *(b)* FAB

*Figure 7.* Impact of the model size on the Fashion-MNIST data hyper-cleaning task using an MLP architecture. We vary the number of neurons in the hidden layer within $\{64, 128, 256, 512\}$ to control the parameter size. The bars represent the average communication cost per iteration (including fluctuations), peak memory footprint, and runtime per epoch.

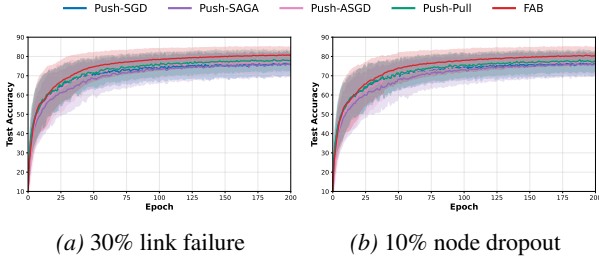

*(a)* 30% link failure      *(b)* 10% node dropout

*Figure 8.* Robustness evaluation on Fashion-MNIST with an MLP under severe communication volatility. The communication settings involve two disruption scenarios: (a) a 30% independent link failure rate and (b) a 10% temporary node dropout probability at each iteration.

**Robustness under communication volatility.** To evaluate the algorithm's resilience to unreliable network conditions, Figure 8 presents performance diagnostics under two severe communication-volatility scenarios on Fashion-MNIST with an MLP: a 30% independent link failure rate and a 10% temporary node dropout probability per iteration. Despite these harsh disruptions, FAB consistently outperforms all baseline methods, maintaining higher test accuracy and exhibiting significantly lower variance (as indicated by the narrower shaded regions). This robust performance aligns with our network connectivity analysis, demonstrating that temporal link accumulation effectively mitigates the impact of instantaneous packet drops. Furthermore, because this neural network task involves a nonconvex lower-level objective, these results provide compelling empirical evidence of FAB's robustness and generalization capacity well beyond the strongly convex lower-level regime covered by our theory. Full experimental details are deferred to Appendix C.

**Comparison of the different integrations of techniques.** We compare the proposed FAB with four variants that integrate different techniques: (1) **Centralized_FAB**: The centralized version of FAB, which is equivalent to F$^2$SA (Kwon

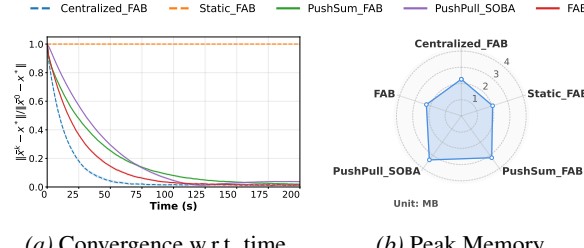

*(a)* Convergence w.r.t. time    *(b)* Peak Memory

*Figure 9.* Performance of algorithms that integrate different techniques for distributed policy evaluation in RL over time-varying directed graphs.

et al., 2023); (2) **Static_FAB**: The static version of FAB, using the initial topology and ignoring temporal edge changes; (3) **PushSum-FAB**: A variant of FAB that replaces the AB/Push–Pull technique with the Push-Sum, as described in Algorithm 3 in the Appendix; (4) **PushPull-SOBA**: A variant of FAB that integrates the SOBA algorithm (Dagréou et al., 2022) with the Push–Pull technique, as described in Algorithm 4 in the Appendix.

The results are shown in Figure 9, from which we observe the following: (i) The divergence of Static_FAB confirms the necessity of dynamic communication protocols in time-varying networks for bilevel optimization; (ii) The proposed FAB performs the best in both computing time and memory cost compared to PushSum_FAB and PushPull-SOBA.

PushPull-SOBA can be competitive at lower accuracy thresholds, but it tracks additional bilevel derivative information compared with FAB's value-function penalty loop.

## 5. Conclusion

This paper introduces FAB, an AB/Push–Pull-based gradient algorithm for distributed bilevel optimization over time-varying directed graphs. By relying on first-order gradient oracles, FAB reduces computational and memory costs. We establish that FAB converges to a stationary solution at an $\mathcal{O}(K^{-\frac{2}{3}})$ rate, as measured by the hypergradient. The current guarantee is limited to lower-level strongly convex problems, and its worst-case constants depend exponentially on the network size through stochastic-matrix contraction factors. Despite these theoretical limitations regarding the exponential dependence, the proposed framework provides an effective and scalable solution for distributed bilevel optimization. Empirically, when the dynamic topology is not pathologically sparse or poorly connected, the algorithm performs well and scales gracefully to a large number of nodes. Improving the convergence rate of FAB (e.g., achieving a polynomial upper bound in $n$) and extending the method to nonconvex lower-level objectives and general stochastic settings are promising directions for future research.

## Impact Statement

This paper presents work whose goal is to advance the field of distributed learning. There are many potential societal consequences of our work, none of which we feel must be specifically highlighted here.

## Acknowledgements

This work was supported by the National Key R&D Program of China (Grant No. 2023YFA1011400), the National Natural Science Foundation of China (Grant Nos. 12326605, 12571321, 12222106, and 12271278), and the Guangdong S&T Programme (No. 2024B0101010003). The authors are listed in alphabetical order.

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

## Contents of Appendices

## A. More Related work

### A.1. Decentralized Optimization

Foundational contributions to decentralized optimization include gradient-based methods (Nedić & Ozdaglar, 2009; Yuan et al., 2016), diffusion strategies (Chen & Sayed, 2012), dual averaging (Duchi et al., 2012), ADMM (Shi et al., 2014), and Newton methods (Mokhtari et al., 2017; Varagnolo et al., 2016). Subsequently, a series of bias-correction techniques have been developed, most notably EXTRA (Shi et al., 2015), Exact-Diffusion (Li et al., 2019; Yuan et al., 2023), and gradient tracking frameworks (Koloskova et al., 2021; Pu & Nedić, 2021). However, these works primarily utilize static network topologies. While recent advancements have extended the analysis to time-varying undirected networks (Kovalev et al., 2021; 2024; Li & Lin, 2024), a critical limitation persists: these algorithms fundamentally rely on undirected communication protocols (i.e., symmetric information exchange), which are often impractical in dynamic environments.

In many practical scenarios, communication among agents is inherently directed. This is particularly evident in applications such as satellite constellation networks (Kaushal & Kaddoum, 2017; Han et al., 2023) and unmanned systems (Ren & Beard, 2005; Das et al., 2019; Gaydamaka et al., 2024) (e.g., UAV swarms), where physical constraints often dictate asymmetric information flow. To address the challenges of directed graphs, sophisticated algorithms based on the Push-Sum mechanism (Tsianos et al., 2012) were developed, including subgradient-push (SGP) (Nedić & Olshevsky, 2015; Assran et al., 2019), SONATA (Tian et al., 2020), and Push-DiGing/ADD-OPT (Nedić et al., 2017; Xi et al., 2018). Although

theoretical research on nonconvex time-varying directed settings remains relatively sparse, recent work by Chen et al. (2025c) proposed the Push-ASGD algorithm and provided theoretical guarantees for it. While these methods utilize asymmetric communication protocols to accommodate directed topologies, their performance in dynamic settings is often compromised. Specifically, their convergence is hindered by inefficient information mixing, a consequence of relying on single-sided weight matrices at each time step (Nedić et al., 2025).

More recently, the Push-Pull (AB) communication mechanism, introduced by Xin et al. (2018) and Pu et al. (2021), has emerged as a robust alternative, as outlined in Algorithm 2. By employing distinct row- and column-stochastic mixing matrices, this framework significantly improves convergence rates compared to Push-Sum-based approaches. However, existing theoretical analyses face distinct limitations. Although Saadatniya et al. (2020) and Nedić et al. (2025) extended the method to time-varying directed networks, their results rely heavily on the restrictive assumption of strong convexity. Conversely, while recent contributions (Liang et al., 2025; You & Pu, 2025) have explored non-convex settings, they are limited to static environments. Consequently, the convergence behavior of Push-Pull for non-convex objectives over dynamic directed networks remains largely unexplored.

---

**Algorithm 2** Push-Pull (TV-$\mathcal{AB}$) Algorithm

---

1: **Input:** Initial iterates $\{x_i^0\}_{i=1}^n$; Initial tracking variables $\{t_{x,i}^0\}_{i=1}^n$ (initialized as $t_{x,i}^0 = \nabla f_i(x_i^0)$); Sequence of column-stochastic matrices $\{A^k\}$ and row-stochastic matrices $\{B^k\}$; Learning rates $\{\eta_x^k\}$; Total iterations $K$.
2: **for** $k = 0, 1, \ldots, K-1$ **do**
3:     for all agents $i \in \{1, \ldots, n\}$ do in parallel:
4:       **Step 1: Decision variable update**
5:       Receive $x_j^k$ from neighbors and update decision variables $x_i^{k+1}$:
6:       $x_i^{k+1} = \sum_{j=1}^n [A^k]_{ij} x_j^k - \eta_x^k t_{x,i}^k$;
7:       **Step 2: Local gradient evaluation**
8:       Evaluate the gradient at the new state $\nabla f_i(x_i^{k+1})$;
9:       **Step 3: Tracking variable update**
10:      Receive $t_{x,j}^k$ from neighbors and update tracking variables $t_{x,i}^{k+1}$:
11:      $t_{x,i}^{k+1} = \sum_{j=1}^n [B^k]_{ij} t_{x,j}^k + \nabla f_i(x_i^{k+1}) - \nabla f_i(x_i^k)$;
12:     **end for**
13: **end for**

---

## A.2. Bilevel Optimization

The study of bilevel optimization originates from Stackelberg games (Stackelberg, 1952), where the lower-level problem serves as a constraint for the upper-level objective (Bracken & McGill, 1973; Ye & Zhu, 1995; Colson et al., 2007). A classical solution strategy involves reformulating the bilevel problem as a single-level Mathematical Program with Equilibrium Constraints (MPEC) by replacing the lower-level optimization with its optimality conditions (Fukushima et al., 1998; Luo et al., 1996; Lv et al., 2011).

In the context of machine learning, gradient-based bilevel methods have seen significant progress (Liao et al., 2018; Ji et al., 2021; Shaban et al., 2019; Liu et al., 2021; Hong et al., 2023). However, these approaches typically rely on evaluating Hessian-inverse-vector products to estimate hypergradients. This requirement imposes substantial computational and runtime overheads, particularly for massive-scale deep models like Large Language Models (LLMs), where accessing second-order information is often intractable (Pearlmutter, 1994; Dagréou et al., 2024).

To address these computational challenges and improve scalability, value-function-based penalty methods and other fully first-order bilevel methods (Shen & Chen, 2023; Kwon et al., 2023; Lu & Mei, 2024; Chen et al., 2025a) have emerged as effective alternatives. By transforming or approximating the bilevel problem with a single-level objective—for example, through lower-level value-function or penalty reformulations—these approaches enable algorithm designs that avoid explicit Hessian inverses. For instance, Liu et al. (2024) utilized the Moreau envelope to relax the value function, developing a first-order algorithm for lower-level non-convex problems without assuming the Polyak-Łojasiewicz (PL) condition; however, their analysis characterizes the stationarity of the penalized problem rather than the original objective. More recently, related first-order ideas have been extended to distributed and federated settings.

These developments motivate our value-function penalty formulation, but the first-order bilevel approaches above are

centralized, server-based, or restricted to static communication structures. More recently, Yang et al. (2025) extended this paradigm to a federated learning framework, while Wang et al. (2025) studied fully first-order decentralized bilevel optimization over static networks. FAB instead targets fully decentralized learning over time-varying directed graphs.

### A.3. Decentralized bilevel optimization

To the best of our knowledge, existing decentralized bilevel optimization methods have predominantly focused on static undirected networks. These approaches typically address Problem (1) by utilizing the hypergradient $\nabla \mathcal{F}^*(x)$:

$$\nabla \mathcal{F}^*(x) = \nabla_x F(x, y^*) - \nabla_{xy}^2 G(x, y^*) \big( \nabla_{yy}^2 G(x, y^*) \big)^{-1} \nabla_y F(x, y^*), \tag{14}$$

where $y^* \coloneqq y^*(x)$. A fundamental challenge in distributed settings arises from the non-separability of the Hessian inverse. Specifically, the average of local hypergradient estimators does not align with the global hypergradient:

$$\frac{1}{n} \sum_{i=1}^n \Big( \nabla_x f_i(x, y^*) - \nabla_{xy}^2 g_i(x, y^*) \big( \nabla_{yy}^2 g_i(x, y^*) \big)^{-1} \nabla_y f_i(x, y^*) \Big) \neq \nabla \mathcal{F}^*(x).$$

Several strategies address this challenge by introducing surrogate techniques to approximate the inverse Hessian. Chen et al. (2025b) pioneered the DBO algorithm, covering both stochastic and deterministic settings, which integrates a decentralized approach to solve the linear system in (14) via the Jacobian-Hessian-Inverse Product (JHIP). This framework was subsequently enhanced by incorporating moving average techniques in Chen et al. (2023). Similarly, Yang et al. (2022) employed a Neumann series expansion combined with gossip communication protocols and provided a rigorous analysis of sample complexity. To further improve efficiency, Gao et al. (2023) introduced momentum and variance-reduction techniques for distributed stochastic bilevel problems. More recently, Nazari et al. (2025) proposed estimating the hyper-gradient of the penalty function through the decentralized computation of matrix-vector products and vector communications. However, these methods typically adopt a double-loop structure, which inevitably imposes significant computational burdens and communication overheads.

To mitigate the latency of double-loop schemes, recent works have pivoted towards single-loop alternatives. For instance, Lu et al. (2022) proposed a linearized augmented Lagrangian method, while Dong et al. (2025) extended the SOBA algorithm (originally by Dagréou et al. (2022)) to the decentralized setting. Although these methods eliminate the inner loop, they generally remain reliant on second-order information. As noted in Dagréou et al. (2024), computing Hessian-vector products can be significantly more expensive—in both time and memory—than gradient computations, particularly in deep learning applications. To eliminate second-order computations entirely, recent breakthroughs have leveraged value-function-based methods to develop fully first-order algorithms, such as those by Wang et al. (2025) and Kong et al. (2025).

Another line of research focuses on personalized DSBO. Unlike previous methods that require consensus on all variables, these approaches only synchronize the upper-level variables, allowing the lower-level variables to remain local to each agent. Representative works include (Liu et al., 2022; Qiu et al., 2023; Liu et al., 2023; Niu et al., 2025; Qin et al., 2025).

Nevertheless, a critical limitation persists: these algorithms rely heavily on symmetric doubly stochastic matrices (implying undirected communication). Consequently, they cannot be directly generalized to time-varying directed graphs, where maintaining such symmetry is often infeasible.

## B. Additional Experimental Results

### B.1. Distributed Hyperparameter Tuning

As shown in Figure 10, we also evaluated the baselines with a fixed $\lambda = 0.15$, which corresponds to the final value learned by FAB. While this setup marginally outperforms the grid-searched version ($\lambda = 0.2$), it still falls short of FAB. This result suggests that FAB's advantage stems from its inherent bilevel structure.

### B.2. Distributed Policy Evaluation for RL

In this subsection, we provide additional experimental results to further evaluate the robustness and efficiency of the proposed FAB algorithm in the context of distributed policy evaluation. Specifically, we conduct comprehensive sensitivity analyses regarding two critical environmental parameters: the observation noise standard deviation $\omega$ and the network connectivity probability $\nu$. The detailed comparisons are presented below.

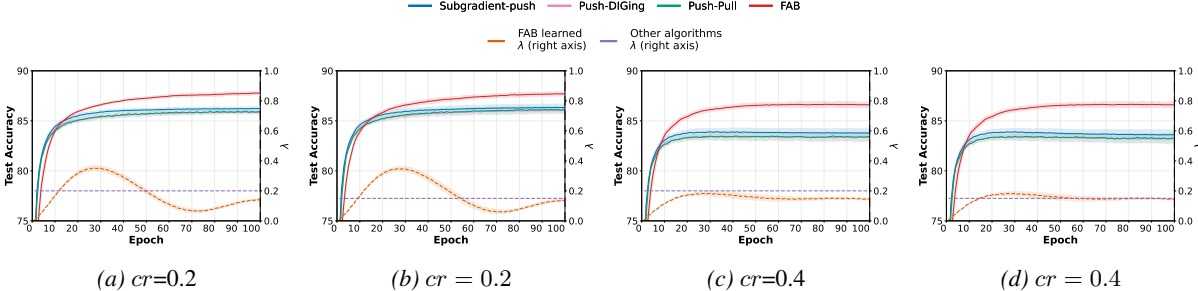

*(a) cr=0.2*  *(b) $cr = 0.2$*  *(c) cr=0.4*  *(d) $cr = 0.4$*

*Figure 10.* **Robustness against label noise over time-varying directed graphs.** Test accuracy comparison of single-level baselines (subgradient-push (Nedić & Olshevsky, 2015), Push-DIGing (Nedić et al., 2017), Push-Pull (Nedić et al., 2025)) against our proposed FAB algorithm under corruption rates ($cr$) of 0.2 and 0.4 . For the baselines, we employ fixed regularization parameters: (a) and (c) use $\lambda = 0.2$ (selected via grid search), while (b) and (d) use $\lambda = 0.15$ (approximating the value learned by FAB). The dotted line (right axis) tracks the evolution of $\lambda$ in FAB, illustrating its capability to *adaptively* adjust the hyperparameter to counteract data corruption.

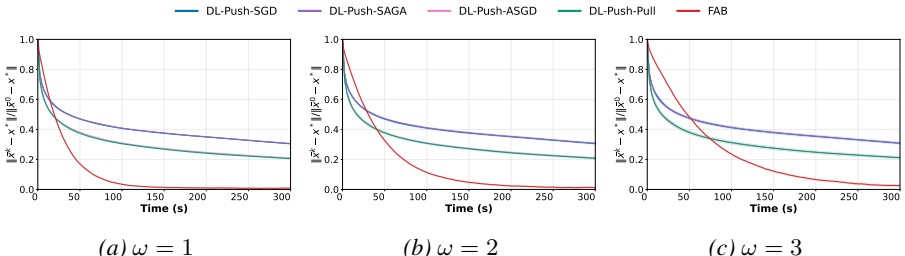

*(a) $\omega = 1$*  *(b) $\omega = 2$*  *(c) $\omega = 3$*

*Figure 11.* Performance comparison on the distributed RL task under varying noise levels. We observe the convergence behavior with noise standard deviation $\omega$ set to 1, 2, and 3, respectively. Fixed parameters: network connectivity $\nu = 0.3$.

**Effect of noise level $\omega$.** Figure 11 presents the convergence results under varying noise standard deviations $\omega \in \{1, 2, 3\}$. Although FAB exhibits higher sensitivity to noise—likely attributable to its single-loop structure—it demonstrates superior runtime efficiency compared to double-loop baselines. While methods like DL-Push-SGD exhibit a sharp initial error drop, their convergence speed stagnates significantly in later stages. Consequently, FAB achieves the best final accuracy.

**Effect of network connectivity $\nu$.** Figure 12 illustrates the convergence performance under varying connectivity probabilities $\nu \in \{0.1, 0.2, 0.3\}$. FAB consistently exhibits the best performance among all compared algorithms. Regarding the impact of topology, a larger $\nu$ clearly accelerates the convergence speed. This is because higher connectivity facilitates faster information mixing across the network. However, this performance gain comes with a trade-off. A larger $\nu$ implies a denser graph structure, which inevitably results in higher communication overhead per iteration, as shown in Figure 2 (d).

### B.3. Distributed Data Hyper-cleaning

In this subsection, we present additional experimental results for the distributed data hyper-cleaning task using the Fashion-MNIST dataset. We specifically investigate the impact of network topology (controlled by $\nu$) on the algorithm's robustness and communication overhead. Furthermore, we provide a comprehensive analysis of runtime efficiency to validate the computational superiority of FAB under diverse environmental settings.

**Effect of $\nu$:** As demonstrated in Figure 14, varying $\nu$ has negligible impact on the peak accuracy of all algorithms. However, a comparison between (a) and (c) reveals that denser networks contribute to improved robustness, albeit at the cost of higher communication overhead, as shown in (d). Specifically, while FAB incurs the highest communication cost, it achieves superior performance. Notably, this performance advantage becomes more pronounced as the corruption rate $cr$ increases, as corroborated by Figure 3.

**Runtime efficiency with BERT.** We evaluate the runtime performance on the IMDB dataset using the BERT model, as illustrated in Figure 15. The results indicate that while single-loop baselines exhibit a slight speed advantage under low corruption rates, FAB achieves significantly higher accuracy, as shown in Figure 5. Moreover, FAB surpasses the baselines

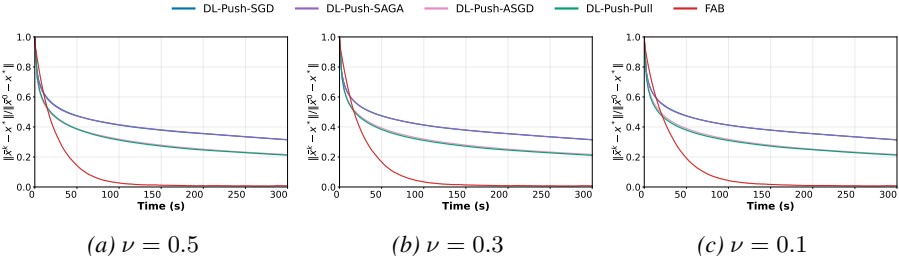

*(a) ν = 0.5*      *(b) ν = 0.3*      *(c) ν = 0.1*

*Figure 12.* Performance comparison on the distributed RL task under varying data heterogeneity levels. We observe the convergence behavior with the network connectivity $\nu$ set to 0.5, 0.3, and 0.1, respectively. Fixed parameters: noise levels $\omega = 1$.

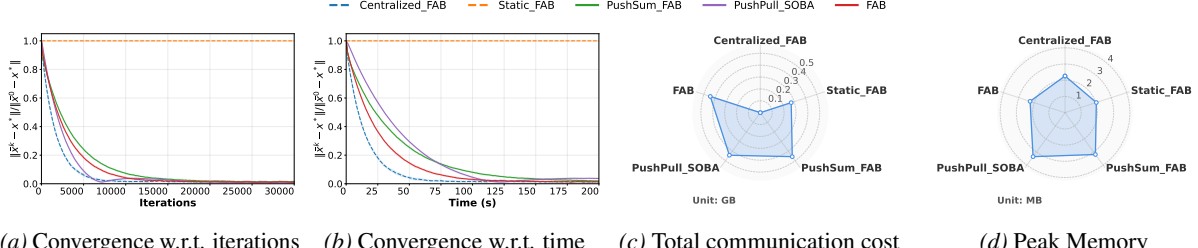

*(a)* Convergence w.r.t. iterations  *(b)* Convergence w.r.t. time  *(c)* Total communication cost  *(d)* Peak Memory

*Figure 13.* Performance of different bilevel algorithms on the RL task. We benchmark FAB against four variants: (1) Centralized_FAB (non-distributed environment); (2) Static_FAB (non-dynamic communication); (3) PushSum_FAB (non-Push-Pull mechanism); (4) PushPull_SOBA (non-fully first-order baseline).

in time efficiency as the corruption level intensifies.

## B.4. Ablation Study

**Sensitivity analysis of FAB.** To rigorously evaluate the robustness of FAB, we conducted a sensitivity analysis on the distributed reinforcement learning task by varying one hyperparameter at a time while keeping others fixed. As detailed in Table 1, we observe the following distinct behaviors: (1) *Penalty parameter ($\lambda$)*: The algorithm demonstrates strong stability within the range $[60, 140]$. However, a trade-off exists: while excessively large values tend to slow down convergence, an insufficiently large $\lambda$ may fail to recover the solution to the original constrained problem, a characteristic inherent to penalty-based methods. (2) *Step sizes ($\eta_y^k, \eta_z^k$)*: The algorithm is notably sensitive to the step sizes $\eta_y^k, \eta_z^k$. Our results indicate that performance is optimal when $\eta_y^k$ and $\eta_z^k$ are balanced (i.e., kept approximately equal). Significant discrepancies between them result in slow convergence or divergence (as seen when varying them independently). However, when scaled simultaneously (the last group in Table 1), the algorithm remains robust. (3) *Upper-level step size ($\eta_x^k$)*: For the upper-level variable, a larger step size (within a stable range) generally yields faster convergence, as it accelerates the update of the primary decision variable.

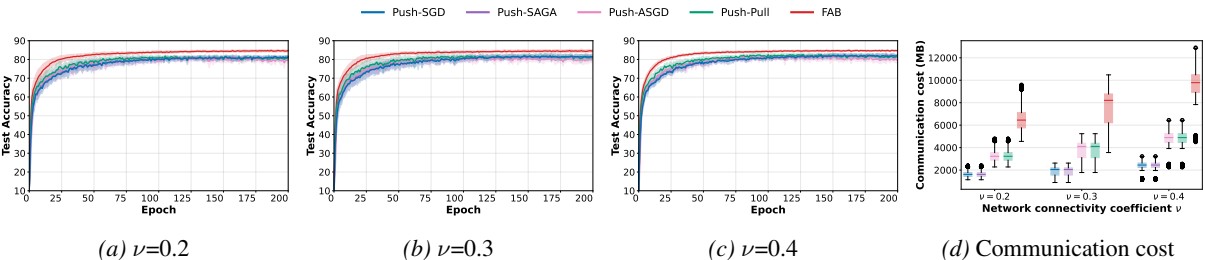

*(a) ν=0.2*      *(b) ν=0.3*      *(c) ν=0.4*      *(d)* Communication cost

*Figure 14.* Performance on Fashion-MNIST under varying network connectivity $\nu$. Subfigures (a)-(c) illustrate the test accuracy, while (d) presents the communication cost per epoch of the compared algorithms. Fixed parameters: corruption rate $cr = 0.4$ and heterogeneity parameter $\rho = 0.5$.

*Table 1.* Sensitivity analysis on the distributed reinforcement learning task. We vary specific hyperparameters while fixing others to the baseline setting ($\lambda = 100, \eta_x^k = \eta_y^k = \eta_z^k = 0.10$). The varied parameters are highlighted with a gray background. The symbol ">Max" indicates that the algorithm failed to converge within the maximum budget of 20,000 iterations.

| $\lambda$ | $\eta_x^k$ | $\eta_y^k$ | $\eta_z^k$ | **Iter** ($< 1\%$) | **Rel. Err.** |
|---|---|---|---|---|---|
| 60 | 0.10 | 0.10 | 0.10 | **3200** | $9.82e-03$ |
| 80 | 0.10 | 0.10 | 0.10 | 4060 | $9.51e-03$ |
| 100 | 0.10 | 0.10 | 0.10 | 4860 | $9.76e-03$ |
| 120 | 0.10 | 0.10 | 0.10 | 5740 | $9.98e-03$ |
| 140 | 0.10 | 0.10 | 0.10 | 6620 | $9.97e-03$ |
| 60 | 0.06 | 0.10 | 0.10 | 5080 | $9.91e-03$ |
| 60 | 0.08 | 0.10 | 0.10 | 4060 | $9.36e-03$ |
| 60 | 0.12 | 0.10 | 0.10 | 2800 | $8.09e-03$ |
| 60 | 0.14 | 0.10 | 0.10 | **2440** | $9.73e-03$ |
| 60 | 0.10 | 0.06 | 0.10 | >Max | $1.24e-01$ |
| 60 | 0.10 | 0.08 | 0.10 | 10760 | $7.66e-03$ |
| 60 | 0.10 | 0.12 | 0.10 | >Max | $1.33e+03$ |
| 60 | 0.10 | 0.14 | 0.10 | >Max | $1.81e+03$ |
| 60 | 0.10 | 0.10 | 0.06 | >Max | $1.80e+03$ |
| 60 | 0.10 | 0.10 | 0.08 | >Max | $1.32e+03$ |
| 60 | 0.10 | 0.10 | 0.12 | 9020 | $8.11e-03$ |
| 60 | 0.10 | 0.10 | 0.14 | 14920 | $8.92e-03$ |
| 60 | 0.10 | 0.06 | 0.06 | 3280 | $9.91e-03$ |
| 60 | 0.10 | 0.08 | 0.08 | 3240 | $9.88e-03$ |
| 60 | 0.10 | 0.12 | 0.12 | 3600 | $8.19e-03$ |
| 60 | 0.10 | 0.14 | 0.14 | 4060 | $9.19e-03$ |

**Penalty parameter, consensus, and tracking diagnostics.**   Table 2 reports diagnostic consensus and tracking errors under the same sensitivity-analysis setting as Table 1. These metrics are empirical counterparts of the consensus and tracking quantities used in the proof. In this fixed-penalty sweep, they indicate that penalty selection should be considered jointly with consensus and tracking behavior; the table is not intended to establish a universal monotonic relationship between $\lambda$ and these errors.

**Cumulative communication to target accuracy/error.**   Since FAB communicates more variables per iteration than single-level Push–Pull, we also report the total communication needed to reach target performance levels in two representative settings. Table 3 reports the communication required to reach target test-accuracy thresholds in the Fashion-MNIST data hyper-cleaning task, whereas Table 4 reports the communication required to reach target relative-error thresholds for the RL algorithmic variants in Figure 9. FAB is not always the cheapest in low-accuracy transient regimes, but it requires the least communication for the stricter $82\%$ and $84\%$ accuracy thresholds in Table 3, and it remains competitive at the strict error

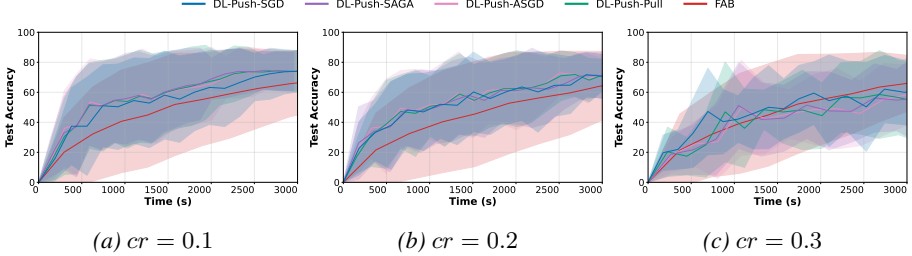

*(a) $cr = 0.1$*        *(b) $cr = 0.2$*        *(c) $cr = 0.3$*

*Figure 15.* Runtime performance evaluation on IMDB with BERT under varying label corruption rates. The network connectivity is fixed at $\nu = 0.5$ and the heterogeneity parameter at $\rho = 0.5$.

*Table 2.* Consensus and tracking diagnostics for fixed penalty values in the distributed reinforcement learning sensitivity experiment.

| Fixed penalty $\lambda$ | 60 | 80 | 100 | 120 | 140 |
|---|---|---|---|---|---|
| Final consensus error | 9.04e−2 | 6.21e−2 | 4.38e−2 | 1.88e−2 | 1.83e−2 |
| Final tracking error | 8.93e−1 | 6.24e−1 | 4.42e−1 | 1.95e−1 | 1.90e−1 |
| Peak consensus error | 1.59e−1 | 1.20e−1 | 9.59e−2 | 7.39e−2 | 5.86e−2 |
| Peak tracking error | 1.44e+0 | 1.10e+0 | 8.82e−1 | 6.96e−1 | 5.60e−1 |

*Table 3.* Cumulative communication volume (MB) required to reach target test-accuracy thresholds in the Fashion-MNIST data hyper-cleaning comparison. Values are mean ± standard deviation over successful runs; the number in parentheses is the number of successful runs out of 10 when not all runs reached the target.

| Algorithm | Acc. $\geq 80\%$ | Acc. $\geq 82\%$ | Acc. $\geq 84\%$ | Acc. $\geq 86\%$ |
|---|---|---|---|---|
| DL-Push-SGD | $51624.0 \pm 8555.4$ | $90694.8 \pm 22475.0$ (8/10) | 116554.0 (1/10) | N/A (0/10) |
| DL-Push-SAGA | $51498.2 \pm 8669.7$ | $92242.5 \pm 18208.8$ (7/10) | 114038.4 (1/10) | N/A (0/10) |
| DL-Push-ASGD | $69469.6 \pm 13968.0$ | $115835.8 \pm 33680.6$ (8/10) | $239070.8 \pm 71367.2$ (2/10) | N/A (0/10) |
| DL-Push-Pull | $65923.0 \pm 10908.4$ | $113653.3 \pm 25917.7$ | $214536.3 \pm 87180.5$ (3/10) | N/A (0/10) |
| FAB (Ours) | $57519.2 \pm 6487.4$ | $85429.4 \pm 15474.4$ | $172098.0 \pm 62980.4$ | $334164.8 \pm 57454.0$ (2/10) |

threshold in Table 4 while maintaining the runtime and memory advantages shown in Figure 9.

**Distributed RL on general dynamic directed graphs at a larger scale.** To examine the applicability of our proposed method on general dynamic directed graphs with larger network sizes, we extended the system to $n = 100$ and $n = 1000$ agents while maintaining a sparse topology ($\nu = 0.05$). As illustrated in Figure 16, the proposed algorithm shows favorable convergence behavior compared with the tested methods in terms of both iterations and runtime. Increasing both the network size $n$ and noise level $\omega$ tends to degrade the performance of all algorithms, but in these experiments the degradation is gradual rather than exponential. Furthermore, among the baselines, the Push-Pull mechanism exhibits better time efficiency as $n$ increases. It is worth noting that while Push-SAGA and Push-SGD show nearly identical convergence behavior in terms of iterations, their runtime performance differs.

**Additional large-network diagnostic on Fashion-MNIST.** The BERT experiments in the main text use $n = 10$ because simulating many decentralized BERT agents requires maintaining independent model weights, gradients, and optimizer states for each agent. To isolate network scaling when the per-agent model footprint is smaller, Table 5 reports an additional Fashion-MNIST MLP diagnostic comparing $n = 10$ and $n = 100$. In this diagnostic, FAB outperforms Push–Pull at both scales, and the degradation from $n = 10$ to $n = 100$ is milder for FAB.

## C. Details of Experiments

In this section, we provide detailed specifications regarding the implementation environment and network configurations to ensure reproducibility. All experiments were implemented in Python 3.9 and conducted on a server equipped with an Intel(R) Xeon(R) Gold 5218R CPU @ 2.10GHz and an NVIDIA A100 GPU (40GB memory). The network topologies employed in our evaluations are visualized in Figure 17. All experimental results are averaged over 10 independent runs. In the plots, solid lines represent the mean values, while shaded regions indicate the standard deviation.

### C.1. Hyperparameter Tuning Strategy for FAB

*In our experiments, hyperparameter tuning is not overly complex.* Specifically, our hyperparameters include the penalty parameter $\lambda$ and the step-sizes for the upper and lower levels, denoted as $\eta_x^k$, $\eta_y^k$, and $\eta_z^k$.

Our sensitivity analysis in Appendix B.4 reveals that different hyperparameters exhibit distinct robustness profiles, allowing for a structured tuning approach. For the penalty parameter $\lambda$, our observations in Table 1 indicate that the algorithm maintains stability over a wide range. As long as the value is large enough to enforce constraints, its exact size has little effect on final accuracy. However, extremely large values may slightly slow down convergence. These observations are empirical and specific to the tested range: once $\lambda$ is large enough to reduce the finite-penalty bias in the tested task, moderate changes

*Table 4.* Cumulative communication volume (MB) required to reach target relative-error thresholds for the algorithmic variants in Figure 9.

| Algorithm | Err. $\leq 0.1$ | Err. $\leq 0.05$ | Err. $\leq 0.02$ | Err. $\leq 0.01$ |
|---|---|---|---|---|
| Static_FAB | N/A | N/A | N/A | N/A |
| PushSum_FAB | $142.33 \pm 0.004$ | $196.38 \pm 0.003$ | $307.73 \pm 0.002$ | N/A |
| PushPull_SOBA | $77.56 \pm 0.006$ | $90.24 \pm 0.005$ | $102.91 \pm 0.005$ | $403.98 \pm 0.004$ |
| FAB (Ours) | $110.84 \pm 0.004$ | $150.42 \pm 0.004$ | $229.65 \pm 0.003$ | $405.57 \pm 0.004$ |

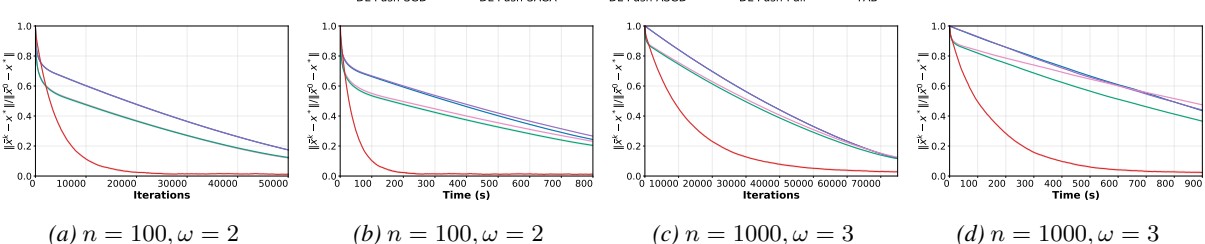

*(a)* $n = 100, \omega = 2$     *(b)* $n = 100, \omega = 2$     *(c)* $n = 1000, \omega = 3$     *(d)* $n = 1000, \omega = 3$

*Figure 16.* Performance comparison on the distributed RL task. We consider different scales of agents ($n = 100$ and $n = 1000$) and noise levels ($\omega = 2$ and $\omega = 3$). The connectivity is fixed at $\nu = 0.05$.

of $\lambda$ have limited impact on final accuracy, while overly large values may require more conservative effective stepsizes and can slow convergence. Regarding the step-sizes, a key finding is the necessity of balance between the lower-level parameters. As shown in the sensitivity analysis, independent variations of $\eta_y^k$ and $\eta_z^k$ can lead to instability. However, when these two are coupled (i.e., $\eta_y^k \approx \eta_z^k$), the algorithm becomes robust to their simultaneous scaling.

Based on these observations, we adopt a decoupled two-tier tuning strategy to efficiently locate optimal configurations:

- *Coarse tuning*: For the penalty parameter $\lambda$, we employ a coarse grid search with large intervals. Given its broad stability region, a coarse estimate is sufficient to ensure convergence without the need for fine-grained adjustment. This statement refers only to the tested experimental configurations; the theoretical conditions still couple $\lambda$ with the stepsizes, as discussed in Lemma D.12.

- *Fine-tuning*: To reduce the search space for step-sizes, we couple the lower-level parameters by setting $\eta_y^k = \eta_z^k = \eta_{low}$. Consequently, we perform a grid search over $\eta_x^k$ and $\eta_{low}$. Guided by our robustness analysis, we prioritize selecting larger values of $\eta_x^k$ within the stable range to accelerate the upper-level optimization process. This strategy effectively reduces the tuning complexity to just two scalar values ($\eta_x$ and $\eta_{low}$).

The above analysis demonstrates that by leveraging the correlation between lower-level step-sizes, FAB's hyperparameter tuning remains computationally efficient and practical for real-world deployment.

## C.2. Distributed Hyperparameter Tuning

We consider a distributed hyperparameter tuning task formulated as a bilevel optimization problem, as defined in Eq. (15). In this setting, we aim to learn a linear classifier while simultaneously tuning the regularization hyperparameters to maximize generalization performance on the test set.

$$
\begin{aligned}
\min_{\boldsymbol{\tau}, \boldsymbol{w}} F(\boldsymbol{\tau}, \boldsymbol{w}) &= \frac{1}{n} \sum_{i=1}^{n} \frac{1}{|\mathcal{D}_{val}^i|} \sum_{(\mathbf{x}_j, y_j) \in \mathcal{D}_{val}^i} \mathcal{L}\big(h(\mathbf{x}_j^\top; \boldsymbol{w}), y_j\big), \\
\text{s.t. } \boldsymbol{w} &\in \arg\min_{\boldsymbol{w}'} f(\boldsymbol{\tau}, \boldsymbol{w}') = \frac{1}{n} \sum_{i=1}^{n} \frac{1}{|\mathcal{D}_{tr}^i|} \sum_{(\mathbf{x}_j, y_j) \in \mathcal{D}_{tr}^i} \mathcal{L}\big(h(\mathbf{x}_j^\top; \boldsymbol{w}'), y_j\big) + \boldsymbol{\tau} \|\boldsymbol{w}'\|^2,
\end{aligned}
\tag{15}
$$

*Network and setup:* We formulate the task as a distributed multiclass linear classification problem with $C = 10$ classes. Let $\boldsymbol{w} \in \mathbb{R}^{d \times C}$ denote the model parameters, where $d = 785$ (flattened input dimension plus bias) and the prediction is given by $h(\mathbf{x}^\top; \boldsymbol{w}) = \mathbf{x}^\top \boldsymbol{w}$. In the lower level, the classifier is trained using local training datasets that are subject to potential

*Table 5.* Test accuracy (%) at four training stages for FAB and Push–Pull under different node scales. Shared settings: Fashion-MNIST, MLP, 200 epochs, batch size 64, label-noise rate 0.5, averaged over 10 seeds.

| Epoch | $n = 10, \alpha = 0.5$ FAB | $n = 10, \alpha = 0.5$ Push–Pull | $n = 100, \alpha = 1.0$ FAB | $n = 100, \alpha = 1.0$ Push–Pull |
|---|---|---|---|---|
| 50 | $82.55 \pm 1.05$ | $63.62 \pm 6.40$ | $70.00 \pm 2.32$ | $39.50 \pm 4.59$ |
| 100 | $83.55 \pm 0.39$ | $66.33 \pm 5.58$ | $74.30 \pm 2.15$ | $39.30 \pm 4.38$ |
| 150 | $83.99 \pm 0.54$ | $68.71 \pm 5.10$ | $77.60 \pm 1.91$ | $42.80 \pm 7.10$ |
| 200 | $83.90 \pm 0.61$ | $67.91 \pm 5.89$ | $81.00 \pm 2.76$ | $45.70 \pm 7.39$ |

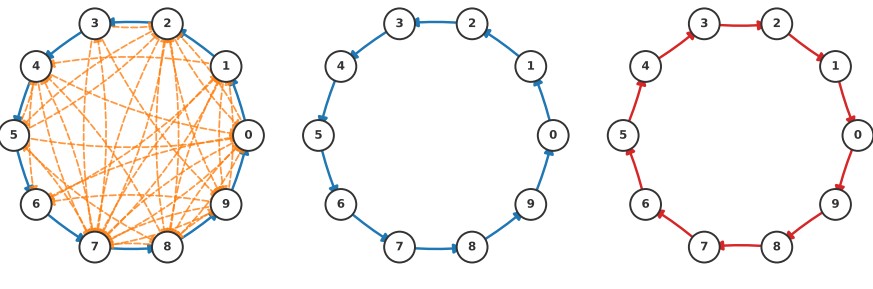

*(a)* Phase I: Augmented ER      *(b)* Phase II: Directed Ring      *(c)* Phase III: Reversed Ring

*Figure 17.* **Illustration of the time-varying network topology.** The system evolves periodically with a cycle of 30 iterations. (a) *Phase I*: An Augmented ER graph combining a fixed cycle (blue solid lines) for strong connectivity and random edges (orange dashed lines) for fast mixing. (b) *Phase II*: A sparse Directed Ring to minimize communication cost. (c) *Phase III*: A Reversed Ring to balance information flow.

label corruption. At the upper level, we aim to optimize the regularization coefficient $\lambda$ to minimize the validation loss on a small, clean validation set $\mathcal{D}_{\text{val}}^i$.

*Implementation details:* We evaluate performance on the MNIST dataset. We partition the dataset across $N = 100$ agents. Each agent is assigned 25 samples, split into 20 for training and 5 for validation. To simulate noisy environments, we inject label corruption with rate $cr$ exclusively into the training set, while the validation set remains clean. We report the top-1 accuracy on the standard MNIST test set. For single-level baselines, we merge the clean validation samples with the corrupted training data to form a unified training set. For FAB, we set the learning rates as $\eta_y^k = 0.02$, $\eta_z^k = 0.02$, and $\eta_x^k = 3 \times 10^{-4}$, with initial regularization $\tau_0 = 0$. For all baseline algorithms, we performed a grid search to select the optimal hyperparameters. The search space for the regularization parameter was $\tau \in \{0, 0.001, 0.01, 0.1, 0.2, 0.3\}$, and the initial step size $\eta$ was selected from $\{1 \times 10^{-5}, 5 \times 10^{-5}, 10^{-4}, 5 \times 10^{-4}, \dots, 0.1\}$. The network consists of $N = 100$ nodes with feature dimension $d = 785$ and 10 classes; the random topology uses connection probability $\nu = 0.05$.

### C.3. Distributed Policy Evaluation for Reinforcement Learning

Following Yang et al.(2022) and Zhu et al.(2024), but considering time-varying directed networks, we evaluate the proposed method on a multi-agent Reinforcement Learning (RL) task, specifically focusing on policy evaluation. Let $\mathcal{S}$ be the state space, $\pi$ be the policy of interest, and $V^\pi(s)$ denotes the value of being in state $s \in \mathcal{S}$ under policy $\pi$. To conduct policy value evaluation, we adopt linear function approximation $\phi_s^\top x$ to estimate the value function as $V^\pi(s)$, where $\phi_s \in \mathbb{R}^d$ denotes the feature vector associated with state $s$, and $x \in \mathbb{R}^d$ represents the unknown parameter vector to be learned, and $d < |\mathcal{S}|$ to reduce the dimension.

To obtain the optimal parameter $x^*$, we consider the Regularized Mean Squared Bellman Error (MSBE) Bellman minimization problem. The global objective is formulated as:

$$\min_{x \in \mathbb{R}^d} F(x) = \frac{1}{n} \sum_{i=1}^{n} \left[ \frac{1}{2|\mathcal{S}|} \sum_{s \in \mathcal{S}} \left( \phi_s^\top x - \mathbb{E}_{s'|s} \left[ r_i(s, s') + \gamma \phi_{s'}^\top x \right] \right)^2 \right] + \frac{\tau}{2} \|x\|_2^2. \tag{16}$$

This objective function $F(x)$ represents a regularized linear least-squares problem. As pointed out by Wang et al. (Wang et al., 2017), problem (16) is $\tau$-strongly convex, guaranteeing the existence of a unique global optimal solution $x^*$ that can be computed efficiently.

To facilitate distributed computation and decouple the target estimation from parameter optimization, we reformulate (16) into a bilevel optimization problem. We introduce an auxiliary variable $y \in \mathbb{R}^{|\mathcal{S}|}$ representing the target value estimates for all states. The problem is decomposed as follows:

The upper-level objective minimizes the fitting error between the linear approximation and the target variable $y$, incorporating the regularization term:

$$f_i(x, y) = \frac{1}{2|\mathcal{S}|} \sum_{s \in \mathcal{S}} (\phi_s^\top x - y_s)^2 + \frac{\tau}{2} \|x\|_2^2. \tag{17}$$

The lower-level problem defines $y$ as the fixed point of the Bellman operator. Each agent $i$ contributes a local loss $g_i$ based on its local reward $r_i$:

$$g_i(x, y) = \sum_{s \in \mathcal{S}} \left( y_s - \mathbb{E}_{s'|s} \left[ r_i(s, s') + \gamma \phi_{s'}^\top x \right] \right)^2. \tag{18}$$

By minimizing the aggregate lower-level loss $\sum_i g_i(x, y)$, the agents collaboratively estimate the global Bellman target $y^*$, which is then used to update $x$ in the upper-level problem.

*Simulation Environment:* We simulate a distributed RL setting with $n = 10$ agents and a state space size of $|\mathcal{S}| = 20$. Feature vectors $\phi_s \in \mathbb{R}^5$ are drawn from $\mathcal{U}[0, 1]^5$, and the transition matrix $P$ is generated by row-normalizing uniformly sampled entries. The expected rewards are drawn from $\mathcal{U}[0, 1]$, with $\gamma = 0.9$ and $\tau = 0.1$. During training, agents compute local gradients using full-batch observations (utilizing all $S$ states) corrupted by i.i.d. Gaussian noise $\xi_k \sim \mathcal{N}(0, \omega^2 I)$, where $\omega \in \{0, 1, 2, 3\}$. The communication network is an Erdős–R'enyi graph with connection probability $\nu \in \{0.1, 0.3, 0.5\}$. We evaluate the effects of noise ($\omega$) and topology ($\nu$) independently by fixing one parameter at its default value while varying the other.

*Implementation Details:* To adapt the single-level methods Push-SGD, Push-SAGA, Push-ASGD, and Push-Pull to this problem, we integrate them with the standard Iterative Differentiation (ITD) technique (Franceschi et al., 2018; Grazzi et al., 2020) for bilevel optimization, and refer to the resulting baselines as their double-loop (DL) variants. Specifically, DL-Push-SGD, DL-Push-SAGA, DL-Push-ASGD, and DL-Push-Pull solve the lower-level problem using $T$ inner iterations of the respective algorithm, and then estimate the hypergradient via automatic differentiation. For FAB, we set the step size to $0.1$ and $\mu = 60.0$. For the double-loop baselines (DL-Push-Pull, -SGD, -SAGA, -ASGD), the inner consensus loop is fixed at $T = 20$ iterations, with step sizes $\eta_0$ set to $0.0025$, $0.002$, $0.002$, and $0.0025$ ($\beta = 0.02$), respectively. Notably, Push-SAGA and Push-SGD exhibit nearly identical convergence. This occurs because our setup employs full-batch gradient computation, eliminating the sampling variance that SAGA is designed to reduce. Since the gradient stochasticity arises solely from additive environmental noise rather than data sub-sampling, the SAGA correction term becomes redundant, reducing its behavior to that of standard Push-SGD.

For the large-scale experiments with $n = 100$ and $n = 1000$, we fix the penalty parameter $\mu = 60.0$, the inner loop $T = 20$, and the graph connectivity probability $p = 0.05$. For the case of $n = 100$ (with noise level $\omega = 2.0$), we set the step size for FAB to $0.1$. For the double-loop baselines (DL-Push-Pull, -SGD, -SAGA, and -ASGD), the step sizes are selected as $0.0015$, $0.001$, $0.001$, and $0.0015$ (with momentum $\beta = 0.9$), respectively. For the larger network of $n = 1000$ (with higher noise $\omega = 3.0$), the FAB step size is adjusted to $0.05$. The baseline step sizes are set to $0.0005$ for DL-Push-Pull, $10^{-5}$ for both DL-Push-SGD and DL-Push-SAGA, and $0.0005$ for DL-Push-ASGD (with $\beta = 0.9$). These more conservative step sizes for the $n = 1000$ setting are essential to mitigate the heightened gradient variance caused by the increased noise level and to maintain algorithmic stability amidst the larger network scale.

## C.4. Distributed Data Hyper-Cleaning

This task is formulated as a decentralized bilevel optimization (DBO) problem. The upper level optimizes the data-cleaning policy (weighting parameters), while the lower level solves the client-specific training objectives under the learned policy.

The mathematical formulation is given by:

$$\min_{\boldsymbol{\psi},\boldsymbol{w}} F(\boldsymbol{\psi},\boldsymbol{w}) = \frac{1}{n}\sum_{i=1}^{n}\frac{1}{|\mathcal{D}_{\mathrm{val}}^{i}|}\sum_{(\mathbf{x}_j,y_j)\in\mathcal{D}_{\mathrm{val}}^{i}}\mathcal{L}\big(h(\mathbf{x}_j;\boldsymbol{w}),y_j\big),$$

$$\text{s.t. } \boldsymbol{w} \in \arg\min_{\boldsymbol{w}'} f(\boldsymbol{\psi},\boldsymbol{w}') = \frac{1}{n}\sum_{i=1}^{n}\frac{1}{|\mathcal{D}_{\mathrm{tr}}^{i}|}\sum_{(\mathbf{x}_j,y_j)\in\mathcal{D}_{\mathrm{tr}}^{i}}\sigma(\psi_j)\mathcal{L}\big(h(\mathbf{x}_j;\boldsymbol{w}'),y_j\big) + \frac{\tau}{2}\|\boldsymbol{w}'\|^2,$$

(19)

where $\mathcal{D}_{\mathrm{tr}}^{i}$ and $\mathcal{D}_{\mathrm{val}}^{i}$ denote the local training and validation datasets of client $i$, respectively; $(\mathbf{x}_j, y_j)$ represents the $j$-th data sample and its label; $\sigma(\cdot)$ is the sigmoid function; and $\mathcal{L}$ denotes the cross-entropy loss. The model parameters are denoted by $\boldsymbol{w}$, and $\tau$ is the regularization coefficient (set to $0.002$ to match the $0.001\|\cdot\|^2$ term).

**Fashion-MNIST with MLP.**    We begin by conducting a data hyper-cleaning experiment on the Fashion-MNIST dataset, which comprises $60,000$ training images and $10,000$ test images. The model architecture is a two-layer MLP $h(\cdot)$ with an input dimension of $784$ and ReLU activation.

*Network and setup:* The system comprises $n = 10$ agents connected via a default topology. We vary the connection probability $\nu \in \{0.1, 0.3, 0.5\}$. The dataset is partitioned using a Dirichlet non-IID scheme with the concentration parameter $\rho \in \{0.1, 0.5, 1.0\}$. Each agent's local data is further split into training and validation sets with a ratio of $9:1$. We introduce asymmetric label noise into the local training data with noise rates selected from $\{0.4, 0.5, 0.6\}$. The label flipping follows the mapping: $0\rightarrow6, 1\rightarrow3, 2\rightarrow4, 3\rightarrow1, 4\rightarrow2, 5\rightarrow7, 6\rightarrow0, 7\rightarrow5, 8\rightarrow9$, and $9\rightarrow8$.

*Implementation Details:* All models are trained for 200 epochs with a batch size of $64$ and $\tau = 0.002$. The method-specific hyperparameter configurations are as follows: Push-Pull ($\eta = 0.03$), Push-ASGD ($\eta = 0.03, \beta = 0.02$), and Push-SAGA ($\eta = 0.02$). For our proposed method FAB, we set $\eta_x = 0.001$, $\eta_y = \eta_z = 0.03$, and $\lambda = 1.0$.

**IMDB with BERT.**    We further evaluate the framework on the IMDB sentiment analysis task, a standard benchmark consisting of $50,000$ movie reviews ($25,000$ for training and $25,000$ for testing) labeled as either positive or negative. For the model architecture, we utilize a pretrained `bert-base-uncased` backbone with a maximum sequence length of $128$. To perform binary classification, we append a linear classification head to the output representations of the backbone, mapping the extracted features to the $2$ sentiment classes.

*Network and setup:* The system comprises $n = 10$ agents connected via a default topology. We vary the connection probability $\nu \in \{0.1, 0.5\}$. The dataset is partitioned using a Dirichlet non-IID scheme with the concentration parameter $\rho \in \{0.5, 1.0\}$. Each agent's local data is further split into training and validation sets with a ratio of $9:1$. We introduce asymmetric label noise into the local training data with noise rates selected from $\{0.3, 0.4\}$.

*Implementation Details:* Training is performed for 20 epochs with a batch size of $4$ and $\tau = 0.002$. We compare FAB against several single-level baselines. The specific hyperparameter configurations are: Push-SGD ($\eta = 0.005$), Push-ASGD ($\eta = 0.005, \beta = 0.9$), Push-Pull ($\eta = 0.005$), Push-SAGA ($\eta = 0.005$), and for our method FAB, we set $\eta_x = 0.01$, $\eta_y = \eta_z = 0.005$, and $\lambda = 1.0$.

### C.5. Ablation Study

**Impact of network scale and matrix weights.**    We conduct ablation studies on a decentralized policy evaluation task within a synthetic MDP environment ($|S| = 20$, feature dimension $d = 5$, discount factor $\gamma = 0.9$, $\ell_2$-regularization $\lambda = 0.1$). We investigate two specific scenarios:

- *Impact of network scale (n):* To analyze scalability (Figure 6(a)), we vary the number of agents $n \in \{10, 15, 20, 25\}$. The communication topology is a directed ring with fixed edge weights $w = 0.2$ (self-loop weight $0.8$). Data heterogeneity is maintained by repeating a base set of $5$ feature matrices and reward patterns across the expanded network.

- *Impact of matrix weights:* To evaluate the sensitivity to communication quality (Figure 6(b)), we fix $n = 10$ and employ a directed ring with alternating weights. Specifically, even-indexed agents use a communication weight and odd-indexed agents use $0.5$, where $a, b$ varies in $\{0.05, 0.10, 0.15, 0.20\}$. A smaller $\epsilon$ indicates weaker connectivity.

*Implementation Details:* All experiments are deterministic (noise-free) to isolate the topological effects. We set the penalty parameter $\mu = 60.0$ and step sizes $\eta_x = \eta_y = \eta_z = 0.1$. The scale experiments run for $50,000$ iterations, while the weight experiments run for $25,000$ iterations. Results are averaged over 10 independent trials with early stopping enabled (relative error $< 10^{-2}$).

**Comparative study of algorithmic designs.** We conduct ablation studies on a decentralized policy evaluation task within a synthetic MDP environment ($|S| = 20$, feature dimension $d = 5$, discount factor $\gamma = 0.9$, $\ell_2$-regularization $\lambda = 0.1$).

*Network and stochastic environment.* The setup comprises $n = 10$ agents connected via a time-varying directed communication graph. At each iteration, the network topology is generated using the Erdős–Rényi model with an edge creation probability of $p = 0.3$, ensuring that the graph remains strongly connected on average. We introduce stochasticity by injecting Gaussian noise into the reward observations at each time step, characterized by a standard deviation of $\omega = 1.0$.

*Baselines.* We compare our method, FAB, against four distinct baselines to evaluate its robustness and efficiency: (1) PushSum_FAB: A variant of FAB that replaces the Push-Pull mechanism with the Push-Pull protocol (ratio consensus); (2) Static_FAB: An ablation baseline that assumes a fixed topology matrix, thereby ignoring dynamic link failures during updates; (3) Centralized_FAB: A single-machine oracle that aggregates global gradients without communication constraints, serving as the performance upper bound; (4) PushPull_SOBA: A state-of-the-art distributed bilevel algorithm based on implicit differentiation. To ensure a fair comparison on directed graphs, we adapted PushPull_SOBA to utilize the same Push-Pull communication protocol as FAB.

*Implementation Details.* All algorithms are trained for $30,000$ iterations. To ensure convergence in the stochastic setting, we apply an inverse-time learning rate decay schedule $\eta_k = \eta_0/(1 + \delta k)$ with a decay rate $\delta = 10^{-5}$. The algorithm-specific hyperparameters (initial step size $\eta_0$ and penalty parameter $\mu$) were tuned individually for optimal performance: Specific settings were tuned for each method: FAB ($\eta_0$=0.15, $\mu$=60.0), PushSum_FAB ($\eta_0$=0.001, $\mu$=30.0), Static_FAB ($\eta_0$=0.001, $\mu$=2.0), Centralized_FAB ($\eta_0$=0.002, $\mu$=60.0), and PushPull_SOBA ($\eta_{x,y}$=$\eta_v$=0.002). Notably, FAB sustains a significantly larger learning rate (0.15 vs. $\sim 10^{-3}$) due to the superior stability of the Push-Pull mechanism. In contrast, PushSum_FAB requires a conservative step size to mitigate the numerical instability inherent in ratio consensus, while Static_FAB is limited by the bias from topological mismatch.

# D. Proof of Theorem 3.4

## D.1. Preliminaries

For the convergence analysis, we first introduce some notations, definitions, and preliminary results for both the bilevel optimization problem and distributed optimization.

**Smoothness and Approximation Properties of Bilevel Optimization.** We consider the distributed bilevel optimization problem formulated as follows:

$$\min_{x \in \mathbb{R}^{d_x}} \quad \mathcal{F}^*(x) := F(x, y^*(x)) = \frac{1}{n} \sum_{i=1}^{n} f_i(x, y^*(x)),$$

$$\text{s.t.} \quad y^*(x) = \arg\min_{y \in \mathbb{R}^{d_y}} G(x, y) := \frac{1}{n} \sum_{i=1}^{n} g_i(x, y). \tag{20}$$

To address the implicit function $y^*(x)$ or the lower-level constraint, we introduce the global penalty function $\mathcal{L} : \mathbb{R}^{d_x} \times \mathbb{R}^{d_y} \times \mathbb{R}^{d_y} \to \mathbb{R}$, defined as follows:

$$\mathcal{L}(x, y, z) := F(x, y) + \lambda \big[ G(x, y) - G(x, z) \big], \tag{21}$$

where $\lambda > 0$ is the penalty parameter. This leads to the unconstrained minimax problem $\min_{x,y} \max_z \mathcal{L}(x, y, z)$. Following the theoretical analysis in Kwon et al., 2023 (2023), we first define the regularized best-response map $y_\lambda^*(x)$ and the optimal penalty objective function $\mathcal{L}_\lambda^*(x)$ as follows:

$$y_\lambda^*(x) := \arg\min_y \mathcal{L}(x, y, z),$$

$$\mathcal{L}_\lambda^*(x) := \mathcal{L}\big(x, y_\lambda^*(x), y^*(x)\big) = \frac{1}{n} \sum_{i=1}^{n} \mathcal{L}_i\big(x, y_\lambda^*(x), y^*(x)\big). \tag{22}$$

The following lemma characterizes the smoothness of the optimal penalty objective function and quantifies the approximation gap between the original problem (1) and the value-function based penalty reformulation (3).

**Lemma D.1** (Chen et al., 2025a, Lemma 4.1). *Under Assumption 3.3, for any $\lambda \geq 2L_{f,1}/\mu$, the function $\mathcal{L}_\lambda^*$ is $L_{*,1}$-smooth, that is,*

$$\|\nabla \mathcal{L}_\lambda^*(x) - \nabla \mathcal{L}_\lambda^*(\theta)\| \leq L_{*,1} \|x - \theta\|,$$

*where the smoothness constant is defined as follows:*

$$L_{*,1} := \left( L_{f,1} + \frac{4L_{f,1}L_{g,1}}{\mu} + \frac{L_{f,0}L_{g,2}}{\mu} + \frac{L_{f,0}L_{g,1}L_{g,2}}{\mu^2} + L_{g,1} \right) \left( \frac{1}{\mu} + \frac{2L_{g,1}}{\mu^2} \right) \left( L_{f,1} + \frac{L_{f,0}L_{g,2}}{\mu} \right).$$

*Furthermore, the gradient of the optimal penalty function $\mathcal{L}_\lambda^*$ can uniformly approximate the hypergradient (i.e., the gradient of $\mathcal{F}^*$) of the original problem (1) as follows:*

$$\|\nabla \mathcal{L}_\lambda^*(x) - \nabla \mathcal{F}^*(x)\| \leq \frac{\sqrt{\mathcal{C}_{gap}}}{\lambda}, \tag{23}$$

*where $\mathcal{C}_{gap}$ is a constant defined as follows:*

$$\mathcal{C}_{gap} := \left( L_{f,1} + \frac{L_{g,2}L_{g,1}}{\mu} + \frac{L_{g,0}L_{g,1}L_{g,2}}{2\mu^2} + \frac{L_{g,0}L_{g,2}}{2\mu} \right)^2 \frac{L_{f,0}^2}{\mu^2}. \tag{24}$$

**Local Gradients and Weighted Norms.** In the distributed setting, the local penalty function for agent $i$ is given by:

$$\mathcal{L}_i(x, y, z) := f_i(x, y) + \lambda \big[ g_i(x, y) - g_i(x, z) \big].$$

In this step, each agent computes the local gradients $d_{\xi,i}^k$ at the most recent local decision variables and tracking variables $t_{\xi,i}^k$ for $\xi \in \{x,y,z\}$. The local gradients are given by:

$$d_{z,i}^k = \lambda \nabla_y g_i(x_i^k, z_i^k), \tag{25a}$$

$$d_{y,i}^k = \nabla_y f_i(x_i^k, y_i^k) + \lambda \nabla_y g_i(x_i^k, y_i^k), \tag{25b}$$

$$d_{x,i}^k = \nabla_x f_i(x_i^k, y_i^k) + \lambda \nabla_x g_i(x_i^k, y_i^k) - \lambda \nabla_x g_i(x_i^k, z_i^k). \tag{25c}$$

Accordingly, the tracking variables are updated using the dynamic consensus mechanism:

$$t_{\xi,i}^{k+1} = \sum_{j \in \mathcal{N}_{i,\text{in}}^k \cup \{i\}} b_{ij}^k t_{\xi,j}^k + d_{\xi,i}^{k+1} - d_{\xi,i}^k, \quad \xi \in \{x,y,z\}.$$

For conciseness, we define the stacked variables for the decision and auxiliary sequences as follows:

$$\mathbf{x}^k := (x_1^k, \ldots, x_n^k) \in \mathbb{R}^{n \times d_x}, \quad \mathbf{y}^k := (y_1^k, \ldots, y_n^k) \in \mathbb{R}^{n \times d_y}, \quad \mathbf{z}^k := (z_1^k, \ldots, z_n^k) \in \mathbb{R}^{n \times d_y}. \tag{26}$$

To accommodate the heterogeneity of the time-varying graph, we introduce weighted norms. For any generic vector $\mathbf{u} = (u_1, \ldots, u_n) \in \mathbb{R}^{n \times d}$ and a positive weight vector $\mathbf{a} = (a_1, \ldots, a_n) \in \mathbb{R}_{++}^n$, we define:

$$\|\mathbf{u}\|_{\mathbf{a}} := \sqrt{\sum_{i=1}^n a_i \|u_i\|^2}, \quad \|\mathbf{u}\|_{\mathbf{a}^{-1}} := \sqrt{\sum_{i=1}^n \frac{\|u_i\|^2}{a_i}}.$$

These norms satisfy the following relations:

$$\|\mathbf{u}\|^2 \leq \frac{1}{\min_i a_i} \|\mathbf{u}\|_{\mathbf{a}}^2, \quad \text{for all } \mathbf{a} > \mathbf{0}, \tag{27}$$

$$\|\mathbf{u}\|^2 \leq \|\mathbf{u}\|_{\mathbf{a}^{-1}}^2, \quad \text{for all } \mathbf{a} > \mathbf{0} \text{ satisfying } \langle \mathbf{a}, \mathbf{1}_n \rangle = 1. \tag{28}$$

**Graph Topology and Matrix Properties.** We recall some definitions related to the graph topology and the properties of the mixing matrices used in the algorithm.

**Definition D.2** (Graph Diameter). The diameter of a strongly connected directed graph $\mathcal{G}$, denoted by $\mathrm{D}(\mathcal{G})$, is the maximum length of the shortest paths between any pair of distinct nodes in $\mathcal{G}$.

**Definition D.3** (Maximal Edge-Utility). For a strongly connected directed graph $\mathcal{G} = ([n], \mathcal{E})$, the maximal edge-utility $\mathrm{K}(\mathcal{G})$ is the maximum value of $K(\mathcal{P})$ taken over all possible shortest-path coverings $\mathcal{P} \in \mathcal{S}(\mathcal{G})$.

The algorithm involves a sequence of row-stochastic matrices $\{A^k\}$ and column-stochastic matrices $\{B^k\}$. Their limiting behaviors are characterized by the following lemmas.

**Lemma D.4** (Nguyen et al., 2025, Lemma 5.4). *Given a sequence of row-stochastic matrices $\{A^k\}_{k \geq 0}$, there exists a sequence of stochastic vectors $\{\alpha_k\}_{k \geq 0}$ such that $\alpha_{k+1}^\top = \alpha_k^\top A^k$ for all $k \geq 0$. Moreover, $[\alpha_{k+1}]_i \geq \frac{a^n}{n}$ for all $i \in [n]$ and $k \geq 0$.*

**Lemma D.5** (Nedić et al., 2025, Lemma 3.4). *Given a sequence of column-stochastic matrices $\{B^k\}_{k \geq 0}$, there exists a sequence of stochastic vectors $\{\beta_k\}_{k \geq 0}$ such that $\beta_{k+1} = B^k \beta_k$ for all $k \geq 0$. Moreover, $[\beta_{k+1}]_i \geq \frac{b^n}{n}$ for all $i \in [n]$ and $k \geq 0$.*

**Weighted Averages and Cumulative Consensus Errors.** To facilitate the convergence analysis, we define two critical error measures: the *consensus error* $D(\cdot)$ and the *gradient tracking error* $S(\cdot)$. For any variable $\xi \in \{x,y,z\}$, let $\hat{\xi}^k := \sum_{i=1}^n [\alpha_k]_i \xi_i^k$ denote the $\alpha_k$-weighted average. The error measures are defined as:

$$D(\boldsymbol{\xi}^k, \alpha_k) := \sum_{i=1}^n [\alpha_k]_i \|\xi_i^k - \hat{\xi}^k\|^2,$$

$$S(\mathbf{t}_\xi^k, \beta_k) := \sum_{j=1}^n [\beta_k]_j \left\| \frac{t_{\xi,j}^k}{[\beta_k]_j} - \sum_{\ell=1}^n t_{\xi,\ell}^k \right\|^2, \tag{29}$$

where $\boldsymbol{\xi} \in \{\mathbf{x}, \mathbf{y}, \mathbf{z}\}$. Note that the tracking error $S$ measures the deviation of the scaled tracking variables from the total sum of gradients. The aggregate consensus error and aggregate gradient tracking are denoted by

$$
\begin{aligned}
\mathbf{V}_D^k &:= D(\mathbf{x}^k, \alpha_k) + D(\mathbf{y}^k, \alpha_k) + D(\mathbf{z}^k, \alpha_k) \\
\mathbf{V}_S^k &:= S(\mathbf{x}^k, \beta_k) + S(\mathbf{y}^k, \beta_k) + S(\mathbf{z}^k, \beta_k)
\end{aligned}
\tag{30}
$$

Finally, based on the algorithm updates and the properties of the mixing matrices, the dynamics of the weighted averages evolve as follows (see Nedić et al., 2025, Proposition 5.1):

$$
\hat{x}^{k+1} = \hat{x}^k - \eta_x^k \sum_{i=1}^n [\alpha_{k+1}]_i t_{x,i}^k, \quad \hat{y}^{k+1} = \hat{y}^k - \eta_y^k \sum_{i=1}^n [\alpha_{k+1}]_i t_{y,i}^k, \quad \hat{z}^{k+1} = \hat{z}^k - \eta_z^k \sum_{i=1}^n [\alpha_{k+1}]_i t_{z,i}^k.
\tag{31}
$$

We also define the aggregated gradients evaluated at these weighted averages for reference in the analysis:

$$
\begin{aligned}
\hat{d}_z^k &= \lambda \nabla_y G(\hat{x}^k, \hat{z}^k), & \text{(32a)} \\
\hat{d}_y^k &= \nabla_y F(\hat{x}^k, \hat{y}^k) + \lambda \nabla_y G(\hat{x}^k, \hat{y}^k), & \text{(32b)} \\
\hat{d}_x^k &= \nabla_x F(\hat{x}^k, \hat{y}^k) + \lambda \nabla_x G(\hat{x}^k, \hat{y}^k) - \lambda \nabla_x G(\hat{x}^k, \hat{z}^k). & \text{(32c)}
\end{aligned}
$$

### D.2. Proof Sketch

Now, we provide a proof sketch of Theorem 3.4 to facilitate a quick understanding of our theoretical analysis for the proposed algorithm. The proof proceeds in three main steps.

**Step 1: Bounding the hypergradient $\nabla \mathcal{F}^*(\bar{x}^k)$.** First, we estimate the gradient of $\mathcal{F}^*(x)$ at the averaged sequence $\bar{x}^k$ by using the gradient of the optimal penalty function $\mathcal{L}_\lambda^*(x)$ (defined in (22)) at the auxiliary sequence $\hat{x}^k$ (defined in (29)).

Indeed, leveraging the smoothness properties of $\mathcal{F}^*$ (Lemma 2.2 in Ghadimi & Wang (2018)) and the relationship between the gradients of $\mathcal{F}^*(x)$ and $\mathcal{L}_\lambda^*(x)$ in Lemma D.1, we establish the following bound:

$$
\|\nabla \mathcal{F}^*(\bar{x}^k)\|^2 \le 4 \|\nabla \mathcal{L}_\lambda^*(\hat{x}^k)\|^2 + \frac{2L_{*,2}^2}{\underline{\alpha}} \mathbf{V}_D^k + \frac{4\mathcal{C}_{gap}}{\lambda^2},
\tag{33}
$$

where $L_{*,2}$ and $\mathcal{C}_{gap}$ are constants detailed in Theorem D.13, and $\mathbf{V}_D^k$ represents the cumulative consensus error defined in (30). This can be derived from (88) with $\mathbf{V}_D^k \ge D(\mathbf{x}^k, \alpha_k)$. The bound in (33) highlights the key challenge: to bound $\|\nabla \mathcal{F}^*(\bar{x}^k)\|^2$, we must simultaneously ensure the decay of $\|\nabla_x \mathcal{L}_\lambda^*(\hat{x}^k)\|^2$, tightly control the consensus error $\mathbf{V}_D^k$, and manage the value of the penalty parameter $\lambda$.

**Step 2: Constructing a proper Lyapunov function and establishing its descent property.** To address the challenge related to the bound in (33), we construct a proper Lyapunov function $\Phi^{(b)}$ and establish its descent property. We define the Lyapunov function as follows:

$$
\Phi^{(b)}(\hat{x}^k) := \mathcal{L}_\lambda^*(\hat{x}^k) + C_{b,1}\|\hat{z}^k - y^*(\hat{x}^k)\|^2 + C_{b,2}\|\hat{y}^k - y_\lambda^*(\hat{x}^k)\|^2 + C_{b,3}\frac{n}{\underline{\alpha}}\mathbf{V}_D^k + C_{b,4}\mathbf{V}_S^k,
\tag{34}
$$

where $\underline{\alpha} := a^n$ and $\mathbf{V}_S^k$ tracks the gradient estimation errors defined in (30). Here and after, without loss of generality, we assume that the lower bound of $F(x, y)$ is 0, ensuring that $\mathcal{L}_\lambda^*(\hat{x}^k)$ is nonnegative, as $z$ takes the value $y^*(x)$ in the definition of $\mathcal{L}_\lambda^*(x)$. If the lower bound of $F(x, y)$ is not 0, we subtract this lower bound from $\Phi^{(b)}(\hat{x}^k)$.

To decouple the error recursion, the coupling coefficients are chosen as follows:

$$
C_{b,1} := \frac{\eta_x^k \lambda}{\eta_z^k} \mathcal{C}_{b,1}, \quad C_{b,2} := \frac{\eta_x^k \lambda}{\eta_y^k} \mathcal{C}_{b,2}, \quad C_{b,3} := \lambda \mathcal{C}_{b,3}, \quad C_{b,4} := \frac{1}{\lambda}.
\tag{35}
$$

Here, $\mathcal{C}_{b,1}$, $\mathcal{C}_{b,2}$, and $\mathcal{C}_{b,3}$ are time-invariant constants that will be carefully selected in (78). Under an appropriate regime of step sizes and the penalty parameter, we establish the descent property for this Lyapunov function as follows:

$$
\Phi^{(b)}(\hat{x}^{k+1}) - \Phi^{(b)}(\hat{x}^k) \le -\frac{1}{4}\underline{\alpha}\underline{\beta}\eta_x^k \|\nabla_x \mathcal{L}_\lambda^*(\hat{x}^k)\|^2 - \frac{1}{5}\underline{c}\mathcal{C}_{b,3}\lambda\frac{n}{\underline{\alpha}}\mathbf{V}_D^k,
\tag{36}
$$

where $\underline{\beta} := b^n$ and $\underline{c}$ is a constant related to the mixing matrix, defined in (75). The proof of (36) can be found in Lemma D.12. This descent property not only guarantees the decay of $\|\nabla_x \mathcal{L}_\lambda^*(\hat{x}^k)\|^2$ and the consensus error $\mathbf{V}_D^k$, but also provides a sufficiently large negative term $-\frac{1}{5}\underline{c}\mathcal{C}_{b,3}\lambda\frac{n}{\underline{\alpha}}\mathbf{V}_D^k$ to dominate the positive error $\frac{2L_{*,2}^2}{\underline{\alpha}}\mathbf{V}_D^k$ appearing in (33) of Step 1.

**Step 3: Establishing the convergence rate.** Finally, we combine the gradient bound from Step 1 with the descent property from Step 2. By rearranging (36) and summing over $k = 0, \ldots, K-1$, we derive:

$$\frac{1}{K}\sum_{k=0}^{K-1}\left\|\nabla_x\mathcal{L}_\lambda^*(\hat{x}^k)\right\|^2 \leq \frac{4\big(\Phi^{(b)}(\hat{x}^0) - \Phi^{(b)}(\hat{x}^K)\big)}{K\underline{\alpha}\,\underline{\beta}\eta_x^k} - \frac{4\underline{c}}{5}\mathcal{C}_{b,3}\lambda\frac{n}{\underline{\alpha}}\frac{1}{K}\sum_{k=0}^{K-1}\mathbf{V}_D^k, \tag{37}$$

where we use the fact that $-\frac{1}{\underline{\alpha}\underline{\beta}\eta_x^k} \leq -1$ (since all $\underline{\alpha}$, $\underline{\beta}$ and $\eta_x^k$ are positive constants bounded by 1).

By controlling $\lambda$ to ensure that the term $\frac{4\underline{c}}{5}\mathcal{C}_{b,3}\lambda\frac{n}{\underline{\alpha}}$ is significantly larger than $\frac{2L_{*,2}^2}{\underline{\alpha}}$ in (33), and substituting this result into (33), we exploit the telescoping property and derive the final convergence rate:

$$\min_{0\leq k < K-1}\left\|\nabla_x\mathcal{F}^*(\bar{x}^k)\right\|^2 \leq \frac{1}{K}\sum_{k=0}^{K-1}\left\|\nabla_x\mathcal{F}^*(\bar{x}^k)\right\|^2 \leq \mathcal{O}\left(\frac{1}{\lambda^2}\right) + \mathcal{O}\left(\frac{1}{\underline{\alpha}\,\underline{\beta}K\eta_x^k}\right).$$

By setting the step sizes $\eta_x^k, \eta_y^k, \eta_z^k = \mathcal{O}\left(K^{-1/3}\right)$ and the penalty parameter $\lambda = \mathcal{O}\left(K^{1/3}\right)$, both terms on the right-hand side scale as $\mathcal{O}(K^{-2/3})$. For detailed proofs, please refer to Subsections D.3–D.7.

### D.3. Descent Property of the Optimal Penalty Function

We now turn our attention to the descent behavior of the optimal penalty function $\mathcal{L}_\lambda^*$. Since the algorithm updates the decision variables using the tracking variable $t_{x,i}^k$ rather than the true gradient $\nabla\mathcal{L}_\lambda^*$, the following analysis quantifies the discrepancy introduced by this approximation, explicitly retaining the consensus and tracking error terms.

Invoking the $L_{*,1}$-smoothness of $\mathcal{L}_\lambda^*$ (Lemma D.1), we derive the following one-step descent inequality.

**Lemma D.6.** *Suppose Assumptions 3.1, 3.2, and 3.3 hold and $\lambda \geq 2L_{f,1}/\mu$. Given the $L_{*,1}$-smoothness of $\mathcal{L}_\lambda^*$, the consecutive iterates generated by Algorithm 1 satisfy the following inequality:*

$$\mathcal{L}_\lambda^*(\hat{x}^{k+1}) - \mathcal{L}_\lambda^*(\hat{x}^k) \leq -\frac{1}{\sum_{i=1}^n[\alpha_{k+1}]_i n[\beta_k]_i}\frac{1}{2\eta_x^k}\|\hat{x}^{k+1} - \hat{x}^k\|^2 + \frac{L_{*,1}}{2}\|\hat{x}^{k+1} - \hat{x}^k\|^2$$
$$+ \frac{\eta_x^k}{2\sum_{i=1}^n[\alpha_{k+1}]_i n[\beta_k]_i}\left(3\lambda^2 L_{g,1}^2\Omega_k'\|\hat{z}^k - y^*(\hat{x}^k)\|^2 + 6(L_{f,1}^2 + \lambda^2 L_{g,1}^2)\Omega_k'\|\hat{y}^k - y_\lambda^*(\hat{x}^k)\|^2\right.$$
$$\left. + 6S(\mathbf{t}_x^k, \beta_k) + (6L_{f,1}^2 + 12\lambda^2 L_{g,1}^2)\frac{n}{\min(\alpha_k)}\mathbf{V}_D^k\right) - \frac{\eta_x^k}{2}\sum_{i=1}^n[\alpha_{k+1}]_i n[\beta_k]_i\left\|\nabla_x\mathcal{L}_\lambda^*(\hat{x}^k)\right\|^2,$$

*where $\hat{\xi}^k := \sum_{i=1}^n[\alpha_k]_i\xi_i^k$ denotes the $\alpha_k$-weighted average for $\xi \in \{x, y, z\}$. The weight vectors $\alpha_k$ and $\beta_k$ are specified in Lemmas D.4 and D.5, respectively. For brevity, we define the scaling factor $\Omega_k' := n^2\max_i([\alpha_{k+1}]_i)^2[\beta_k]_i$. Here, $\mathbf{V}_D^k$ denotes the aggregate consensus error defined in (30), while $S(\cdot)$ represents the gradient tracking error detailed in (29).*

*Proof.* By the $L_{*,1}$-smoothness of $\mathcal{L}_\lambda^*$ (Lemma D.1), we have:

$$\begin{aligned}
&\mathcal{L}_\lambda^*(\hat{x}^{k+1}) - \mathcal{L}_\lambda^*(\hat{x}^k)\\
&\leq \langle\nabla_x\mathcal{L}_\lambda^*(\hat{x}^k), \hat{x}^{k+1} - \hat{x}^k\rangle + \frac{L_{*,1}}{2}\|\hat{x}^{k+1} - \hat{x}^k\|^2\\
&= \frac{1}{\sum_{i=1}^n[\alpha_{k+1}]_i n[\beta_k]_i}\left\langle\sum_{i=1}^n[\alpha_{k+1}]_i n[\beta_k]_i\nabla_x\mathcal{L}_\lambda^*(\hat{x}^k), \hat{x}^{k+1} - \hat{x}^k\right\rangle + \frac{L_{*,1}}{2}\|\hat{x}^{k+1} - \hat{x}^k\|^2
\end{aligned} \tag{38}$$

Using the algebraic identity $\langle u, v \rangle = \frac{1}{2}\|v\|^2 + \frac{1}{2}\|u\|^2 - \frac{1}{2}\|u-v\|^2$ and the update direction $\hat{x}^{k+1} - \hat{x}^k = -\eta_x^k \sum [\alpha_{k+1}]_i t_{x,i}^k$ in (31), we obtain:

$$
\left\langle \sum_{i=1}^n [\alpha_{k+1}]_i n[\beta_k]_i \nabla_x \mathcal{L}_\lambda^*(\hat{x}^k), \hat{x}^{k+1} - \hat{x}^k \right\rangle
$$
$$
= \frac{\eta_x^k}{2} \left\| \sum_{i=1}^n [\alpha_{k+1}]_i n[\beta_k]_i \nabla_x \mathcal{L}_\lambda^*(\hat{x}^k) - \sum_{i=1}^n [\alpha_{k+1}]_i t_{x,i}^k \right\|^2 - \frac{1}{2\eta_x^k}\|\hat{x}^{k+1} - \hat{x}^k\|^2 \tag{39}
$$
$$
- \frac{\eta_x^k}{2} \left\| \sum_{i=1}^n [\alpha_{k+1}]_i n[\beta_k]_i \nabla_x \mathcal{L}_\lambda^*(\hat{x}^k) \right\|^2 .
$$

Next, we bound the error term $\left\| \sum_{i=1}^n [\alpha_{k+1}]_i n[\beta_k]_i \nabla_x \mathcal{L}_\lambda^*(\hat{x}^k) - \sum_{i=1}^n [\alpha_{k+1}]_i t_{x,i}^k \right\|^2$.

$$
\left\| \sum_{i=1}^n [\alpha_{k+1}]_i n[\beta_k]_i \nabla_x \mathcal{L}_\lambda^*(\hat{x}^k) - \sum_{i=1}^n [\alpha_{k+1}]_i t_{x,i}^k \right\|^2
$$
$$
\leq 3 \left\| \sum_{i=1}^n [\alpha_{k+1}]_i n[\beta_k]_i \left( \nabla_x \mathcal{L}_\lambda^*(\hat{x}^k) - \nabla_x \mathcal{L}_\lambda(\hat{x}^k, y_\lambda^*(\hat{x}^k), \hat{z}^k) \right) \right\|^2
$$
$$
+ 3 \left\| \sum_{i=1}^n [\alpha_{k+1}]_i n[\beta_k]_i \left( \nabla_x \mathcal{L}_\lambda(\hat{x}^k, y_\lambda^*(\hat{x}^k), \hat{z}^k) - \nabla_x \mathcal{L}_\lambda(\hat{x}^k, \hat{y}^k, \hat{z}^k) \right) \right\|^2
$$
$$
+ 3 \left\| \sum_{i=1}^n [\alpha_{k+1}]_i \left( n[\beta_k]_i \nabla_x \mathcal{L}_\lambda(\hat{x}^k, \hat{y}^k, \hat{z}^k) - t_{x,i}^k \right) \right\|^2 .
$$

By the Lipschitz properties of the gradients and utilizing Jensen's inequality (noting $\sum[\beta_k]_i = 1$), the first two terms are bounded by the optimality gaps of $y$ and $z$. Specifically:

$$
\left\| \sum_{i=1}^n [\alpha_{k+1}]_i n[\beta_k]_i \nabla_x \mathcal{L}_\lambda^*(\hat{x}^k) - \sum_{i=1}^n [\alpha_{k+1}]_i t_{x,i}^k \right\|^2 \leq 6(L_{f,1}^2 + \lambda^2 L_{g,1}^2) \sum_{i=1}^n n^2 \max_i([\alpha_{k+1}]_i)^2 [\beta_k]_i \left\| \hat{z}^k - y^*(\hat{x}^k) \right\|^2
$$
$$
+ 3\lambda^2 L_{g,1}^2 \sum_{i=1}^n n^2 \max_i([\alpha_{k+1}]_i)^2 [\beta_k]_i \left\| \hat{y}^k - y_\lambda^*(\hat{x}^k) \right\|^2 \tag{40}
$$
$$
+ 3 \sum_{i=1}^n [\beta_k]_i \left\| \frac{t_{x,i}^k}{[\beta_k]_i} - \nabla_x \mathcal{L}_\lambda(\hat{x}^k, \hat{y}^k, \hat{z}^k) \right\|^2 .
$$

For the term $\sum_{i=1}^n [\beta_k]_i \left\| \frac{t_{x,i}^k}{[\beta_k]_i} - n\nabla_x \mathcal{L}_\lambda(\hat{x}^k, \hat{y}^k, \hat{z}^k) \right\|^2$ in (40), according to the definitions of error measures $D(\cdot)$ and $S(\cdot)$ in (29), we have

$$
\sum_{i=1}^n [\beta_k]_i \left\| \frac{t_{x,i}^k}{[\beta_k]_i} - \nabla_x \mathcal{L}_\lambda(\hat{x}^k, \hat{y}^k, \hat{z}^k) \right\|^2
$$
$$
\leq 2 \sum_{i=1}^n [\beta_k]_i \left\| \frac{t_{x,i}^k}{[\beta_k]_i} - \sum_{\ell=1}^n t_{x,\ell}^k \right\|^2 + 2 \sum_{i=1}^n [\beta_k]_i \left\| \sum_{\ell=1}^n t_{x,\ell}^k - n\nabla_x \mathcal{L}_\lambda(\hat{x}^k, \hat{y}^k, \hat{z}^k) \right\|^2 \tag{41}
$$
$$
\leq 2S(\mathbf{x}^k, \beta_k) + 2 \left\| \sum_{\ell=1}^n d_{x,\ell}^k - n\nabla_x \mathcal{L}_\lambda(\hat{x}^k, \hat{y}^k, \hat{z}^k) \right\|^2
$$
$$
\leq 2S(\mathbf{x}^k, \beta_k) + (6L_{f,1}^2 + 12\lambda^2 L_{g,1}^2) \frac{n}{\min(\alpha_k)} \mathbf{V}_D^k
$$

Substituting these bounds in (40) and (41) back into (39) and then back into (38), we obtain:

$$
\mathcal{L}_\lambda^*(\hat{x}^{k+1}) - \mathcal{L}_\lambda^*(\hat{x}^k)
$$
$$
\leq -\frac{1}{\sum_{i=1}^n [\alpha_{k+1}]_i n [\beta_k]_i} \frac{1}{2\eta_x^k} \|\hat{x}^{k+1} - \hat{x}^k\|^2 + \frac{L_{*,1}}{2} \|\hat{x}^{k+1} - \hat{x}^k\|^2
$$
$$
+ \frac{1}{\sum_{i=1}^n [\alpha_{k+1}]_i n [\beta_k]_i} \frac{\eta_x^k}{2} \left( 3\lambda^2 L_{g,1}^2 \sum_{i=1}^n n^2 \max_i ([\alpha_{k+1}]_i)^2 [\beta_k]_i \left\| \hat{z}^k - y^*(\hat{x}^k) \right\|^2 \right.
$$
$$
+ 3\lambda^2 L_{g,1}^2 \sum_{i=1}^n n^2 \max_i ([\alpha_{k+1}]_i)^2 [\beta_k]_i \left\| \hat{y}^k - y_\lambda^*(\hat{x}^k) \right\|^2 + 6 S(\mathbf{x}^k, \beta_k)
$$
$$
\left. + (6 L_{f,1}^2 + 12\lambda^2 L_{g,1}^2) \frac{n}{\min(\alpha_k)} \mathbf{V}_D^k + \left\| \sum_{i=1}^n [\alpha_{k+1}]_i n [\beta_k]_i \nabla_x \mathcal{L}_\lambda^*(\hat{x}^k) \right\|^2 \right).
$$

This completes the proof. □

### D.4. Contraction Properties of Auxiliary Variables

The descent inequality established in Lemma D.6 relies heavily on the proximity of the aggregated auxiliary iterates $\hat{y}^k$ and $\hat{z}^k$ to their respective optimal response maps $y_\lambda^*(\hat{x}^k)$ and $y^*(\hat{x}^k)$. Specifically, the negative gradient term must dominate the error terms involving $\|\hat{y}^k - y_\lambda^*(\hat{x}^k)\|^2$ and $\|\hat{z}^k - y^*(\hat{x}^k)\|^2$ to ensure global convergence.

Since the lower-level problems are strongly convex (Assumption 3.3), performing gradient descent steps on $y$ and $z$ induces a contraction toward the optimal solutions. However, this contraction is perturbed by two factors: the inexactness of the descent directions (due to consensus and tracking errors) and the "drift" of the optimal targets caused by the update of the upper-level variable $\hat{x}$.

The following lemma quantifies this behavior, providing recursive bounds for the approximation errors.

**Lemma D.7.** *Under Assumptions 3.1, 3.2, and 3.3, and provided that the penalty parameter satisfies $\lambda \geq \frac{2L_{f,1}}{\mu}$, the iterates $\hat{y}^k$ and $\hat{z}^k$ generated by Algorithm 1 satisfy the following inequalities for any positive scalars $\{\delta_{k,i}\}_{i=1}^4$:*

*1. Contraction of the unregularized estimator $\hat{z}^k$:*

$$
\left\| \hat{z}^{k+1} - y^*(\hat{x}^{k+1}) \right\|^2 \leq (1 + \delta_{k,1}) \left[ (1 + \delta_{k,2}) q_{z,k}(\eta_z^k) \left\| \hat{z}^k - y^*(\hat{x}^k) \right\|^2 \right.
$$
$$
\left. + \left( 1 + \frac{1}{\delta_{k,2}} \right) (\eta_z^k)^2 \left( 2 S(\mathbf{t}_z^k, \beta_k) + 6\lambda^2 L_{g,1}^2 \frac{n}{\min(\alpha_k)} \mathbf{V}_D^k \right) \right]
$$
$$
+ \frac{L_{g,1}^2}{\mu^2} \left( 1 + \frac{1}{\delta_{k,1}} \right) \left\| \hat{x}^{k+1} - \hat{x}^k \right\|^2. \tag{42}
$$

*2. Contraction of the regularized estimator $\hat{y}^k$:*

$$
\left\| \hat{y}^{k+1} - y_\lambda^*(\hat{x}^{k+1}) \right\|^2 \leq (1 + \delta_{k,3}) \left[ (1 + \delta_{k,4}) q_{y,k}(\eta_y^k) \left\| \hat{y}^k - y_\lambda^*(\hat{x}^k) \right\|^2 \right.
$$
$$
\left. + \left( 1 + \frac{1}{\delta_{k,4}} \right) (\eta_y^k)^2 \left( 2 S(\mathbf{t}_y^k, \beta_k) + 8(L_{f,1}^2 + \lambda^2 L_{g,1}^2) \frac{n}{\min(\alpha_k)} \mathbf{V}_D^k \right) \right]
$$
$$
+ \frac{9 L_{g,1}^2}{\mu^2} \left( 1 + \frac{1}{\delta_{k,3}} \right) \left\| \hat{x}^{k+1} - \hat{x}^k \right\|^2. \tag{43}
$$

*where $\mathbf{V}_D^k$ is defined in (30), $S(\mathbf{t}_\xi^k, \beta_k)$ is defined in (29). The weight vectors $\alpha_k$ and $\beta_k$ are specified in Lemmas D.4 and*

D.5, respectively. Here, the effective contraction rates $q_{z,k}$ and $q_{y,k}$ are defined as:

$$q_{z,k}(\eta_z^k) := \max \left\{ \left(1 - \lambda \eta_z^k n \min(\beta_k) \mu\right)^2, \left(1 - \lambda \eta_z^k n \min(\beta_k) L_{g,1}\right)^2 \right\},$$

$$q_{y,k}(\eta_y^k) := \max \left\{ \left(1 - \lambda \eta_y^k n \min(\beta_k) \mu\right)^2, \left(1 - \lambda \eta_y^k n \min(\beta_k) 6 L_{g,1}\right)^2 \right\}.$$

*Proof.* For the gap between $\hat{z}^{k+1}$ and $y^*(\hat{x}^{k+1})$, we decompose the error term as follows:

$$\begin{aligned}
\left\|\hat{z}^{k+1} - y^*(\hat{x}^{k+1})\right\|^2 &= \left\|\hat{z}^{k+1} - y^*(\hat{x}^k) + y^*(\hat{x}^k) - y^*(\hat{x}^{k+1})\right\|^2 \\
&= \left\|\hat{z}^{k+1} - y^*(\hat{x}^k)\right\|^2 + \left\|y^*(\hat{x}^{k+1}) - y^*(\hat{x}^k)\right\|^2 \\
&\quad + 2\left\langle \hat{z}^{k+1} - y^*(\hat{x}^k), y^*(\hat{x}^k) - y^*(\hat{x}^{k+1})\right\rangle.
\end{aligned} \tag{44}$$

For the last term (the cross term) in (44), applying Young's inequality with an arbitrary parameter $\delta_{k,1} > 0$ yields:

$$2\left\langle \hat{z}^{k+1} - y^*(\hat{x}^k), y^*(\hat{x}^k) - y^*(\hat{x}^{k+1})\right\rangle \leq \delta_{k,1}\left\|\hat{z}^{k+1} - y^*(\hat{x}^k)\right\|^2 + \frac{1}{\delta_{k,1}}\left\|y^*(\hat{x}^{k+1}) - y^*(\hat{x}^k)\right\|^2. \tag{45}$$

Substituting (45) into (44) and applying the Lipschitz continuity of $y^*(x)$, i.e., $\|y^*(x) - y^*(x')\| \leq \frac{L_{g,1}}{\mu}\|x - x'\|$ (see Lemma 2.2 in Ghadimi & Wang (2018)), we obtain:

$$\left\|\hat{z}^{k+1} - y^*(\hat{x}^{k+1})\right\|^2 \leq (1 + \delta_{k,1})\left\|\hat{z}^{k+1} - y^*(\hat{x}^k)\right\|^2 + \left(1 + \frac{1}{\delta_{k,1}}\right)\frac{L_{g,1}^2}{\mu^2}\left\|\hat{x}^{k+1} - \hat{x}^k\right\|^2. \tag{46}$$

For the term $\left\|\hat{z}^{k+1} - y^*(\hat{x}^k)\right\|^2$ in (45), we substitute the update rule of $\hat{z}^{k+1}$ and introduce the gradient term $\hat{d}_z^k$ in (32a):

$$\begin{aligned}
\left\|\hat{z}^{k+1} - y^*(\hat{x}^k)\right\|^2 &= \left\|\hat{z}^k - y^*(\hat{x}^k) - \eta_z^k \sum_{i=1}^n [\alpha_{k+1}]_i n[\beta_k]_i \hat{d}_z^k \right. \\
&\quad \left. + \eta_z^k \sum_{i=1}^n [\alpha_{k+1}]_i n[\beta_k]_i \hat{d}_z^k - \eta_z^k \sum_{i=1}^n [\alpha_{k+1}]_i t_{z,i}^k \right\|^2 \\
&\leq (1 + \delta_{k,2})\left\|\hat{z}^k - y^*(\hat{x}^k) - \eta_z^k \sum_{i=1}^n [\alpha_{k+1}]_i n[\beta_k]_i \hat{d}_z^k\right\|^2 \\
&\quad + \left(1 + \frac{1}{\delta_{k,2}}\right)(\eta_z^k)^2 \sum_{i=1}^n [\alpha_{k+1}]_i \left\|n[\beta_k]_i \hat{d}_z^k - t_{z,i}^k\right\|^2,
\end{aligned} \tag{47}$$

where the last inequality follows from Young's inequality and Jensen's inequality (due to the convexity of $\|\cdot\|^2$ and $\sum_i [\alpha_{k+1}]_i = 1$).

Next, we analyze the contraction of the first term in (47). Given that $\mathcal{L}(x, \cdot, z)$ is $\frac{\lambda \mu}{2}$-strongly convex and $3\lambda L_{g,1}$-smooth, applying Theorem 3 of Chapter 1 in Polyak (1987), we have:

$$\left\|\hat{z}^k - y^*(\hat{x}^k) - \eta_z^k \sum_{i=1}^n [\alpha_{k+1}]_i n[\beta_k]_i \hat{d}_z^k\right\|^2 \leq q_{z,k}(\eta_z^k)\left\|\hat{z}^k - y^*(\hat{x}^k)\right\|^2, \tag{48}$$

where the contraction factor is defined as

$$q_{z,k}(\eta_z^k) := \max \left\{ \left(1 - \frac{1}{2}\lambda \eta_z^k n \min(\beta_k) \mu\right)^2, \left(1 - 3\lambda \eta_z^k n \min(\beta_k) L_{g,1}\right)^2 \right\} < 1.$$

For the term $\sum_{i=1}^{n}[\alpha_{k+1}]_i\big\|n[\beta_k]_i\hat{d}_z^k - t_{z,i}^k\big\|^2$ in (47),

$$
\sum_{i=1}^{n}[\alpha_{k+1}]_i\big\|n[\beta_k]_i\hat{d}_z^k - t_{z,i}^k\big\|^2
$$

$$
\leq 2\sum_{i=1}^{n}[\beta_k]_i\Big\|\frac{t_{z,i}^k}{[\beta_k]_i} - \sum_{\ell=1}^{n}t_{z,\ell}^k\Big\|^2 + 2\sum_{i=1}^{n}[\beta_k]_i\Big\|\sum_{\ell=1}^{n}t_{z,\ell}^k - n\hat{d}_z^k\Big\|^2
$$

$$
\leq 2S(\mathbf{t}_z,\beta_k) + 6\lambda^2 L_{g,1}^2\frac{n}{\min(\alpha_k)}\mathbf{V}_D^k. \tag{49}
$$

Then we have

$$
\big\|\hat{z}^{k+1} - y^*(\hat{x}^{k+1})\big\|^2 \leq (1+\delta_{k,1})\Big((1+\delta_{k,2})q_k(\eta_z^k)\big\|\hat{z}^k - y^*(\hat{x}^k)\big\|^2 + (1+\frac{1}{\delta_{k,2}})\eta_z^{k\,2}\big(2S(\mathbf{t}_z,\beta_k)
$$

$$
+ 6\lambda^2 L_{g,1}^2\frac{n}{\min(\alpha_k)}\mathbf{V}_D^k\Big) + \frac{L_{g,1}^2}{\mu^2}\big(1+\frac{1}{\delta_{k,1}}\big)\big\|\hat{x}^{k+1} - \hat{x}^k\big\|^2.
$$

Similarly, for the gap between $\hat{y}^{k+1}$ and $y_\lambda^*(\hat{x}^{k+1})$ on the server, we decompose the error as:

$$
\big\|\hat{y}^{k+1} - y_\lambda^*(\hat{x}^{k+1})\big\|^2 = \big\|\hat{y}^{k+1} - y_\lambda^*(\hat{x}^k) + y_\lambda^*(\hat{x}^k) - y_\lambda^*(\hat{x}^{k+1})\big\|^2
$$

$$
= \big\|\hat{y}^{k+1} - y_\lambda^*(\hat{x}^k)\big\|^2 + \big\|y_\lambda^*(\hat{x}^{k+1}) - y_\lambda^*(\hat{x}^k)\big\|^2
$$

$$
+ 2\langle\hat{y}^{k+1} - y_\lambda^*(\hat{x}^k), y_\lambda^*(\hat{x}^k) - y_\lambda^*(\hat{x}^{k+1})\rangle. \tag{50}
$$

For the last term (cross term) in (50), we have:

$$
2\langle\hat{y}^{k+1} - y_\lambda^*(\hat{x}^k), y_\lambda^*(\hat{x}^k) - y_\lambda^*(\hat{x}^{k+1})\rangle \leq \delta_{k,3}\big\|\hat{y}^{k+1} - y_\lambda^*(\hat{x}^k)\big\|^2 + \frac{9L_{g,1}^2}{\mu^2\delta_{k,3}}\big\|\hat{x}^{k+1} - \hat{x}^k\big\|^2, \tag{51}
$$

where the above inequality follows from Young's inequality and the Lipschitz continuity of $y_\lambda^*(x)$ with constant $3L_{g,1}/\mu$ (see Lemma 3.2 in Kwon et al. (2023)). Then the inequality (51) can be reformulated as

$$
\big\|\hat{y}^{k+1} - y_\lambda^*(\hat{x}^{k+1})\big\|^2 \leq (1+\delta_{k,3})\big\|\hat{y}^{k+1} - y_\lambda^*(\hat{x}^k)\big\|^2 + \frac{9L_{g,1}^2}{\mu^2}\big(1+\frac{1}{\delta_{k,3}}\big)\big\|\hat{x}^{k+1} - \hat{x}^k\big\|^2. \tag{52}
$$

For the term $\big\|\hat{y}^{k+1} - y_\lambda^*(\hat{x}^k)\big\|^2$ in inequality (52),

$$
\big\|\hat{y}^{k+1} - y_\lambda^*(\hat{x}^k)\big\|^2
$$

$$
\leq\Big\|\hat{y}^k - y_\lambda^*(\hat{x}^k) - \eta_y^k\sum_{i=1}^{n}[\alpha_{k+1}]_i n[\beta_k]_i\hat{d}_y^k + \eta_y^k\sum_{i=1}^{n}[\alpha_{k+1}]_i n[\beta_k]_i\hat{d}_y^k - \eta_y^k\sum_{i=1}^{n}[\alpha_{k+1}]_i t_{y,i}^k\Big\|^2
$$

$$
\leq (1+\delta_{k,4})\Big\|\hat{y}^k - y_\lambda^*(\hat{x}^k) - \eta_y^k\sum_{i=1}^{n}[\alpha_{k+1}]_i n[\beta_k]_i\hat{d}_y^k\Big\|^2
$$

$$
+ (1+\frac{1}{\delta_{k,4}})\eta_y^{k\,2}\sum_{i=1}^{n}[\alpha_{k+1}]_i\big\|n[\beta_k]_i\hat{d}_y^k - t_{y,i}^k\big\|^2 \tag{53}
$$

Analogous to the derivation for $\hat{z}^k$ in (48), the contraction inequality for the $\hat{y}$-update is given by:

$$
\Big\|\hat{y}^k - y_\lambda^*(\hat{x}^k) - \eta_y^k\sum_{i=1}^{n}[\alpha_{k+1}]_i n[\beta_k]_i\hat{d}_y^k\Big\|^2 \leq q_{y,k}(\eta_y^k)\big\|\hat{y}^k - y_\lambda^*(\hat{x}^k)\big\|^2, \tag{54}
$$

where the contraction factor is defined as

$$
q_{y,k}(\eta_y^k) := \max\Big\{\big(1 - \lambda\eta_y^k n\min(\beta_k)\mu\big)^2, \big(1 - \lambda\eta_y^k n\min(\beta_k)L_{g,1}\big)^2\Big\} < 1.
$$

For the term $\sum_{i=1}^{n}[\alpha_{k+1}]_i\left\|n[\beta_k]_i\hat{d}_y^k - t_{y,i}^k\right\|^2$ in (53),

$$
\sum_{i=1}^{n}[\alpha_{k+1}]_i\left\|n[\beta_k]_i\hat{d}_y^k - t_{y,i}^k\right\|^2
$$
$$
\leq 2\sum_{i=1}^{n}[\beta_k]_i\left\|\frac{t_{y,i}^k}{[\beta_k]_i} - \sum_{\ell=1}^{n}t_{y,\ell}^k\right\|^2 + 2\sum_{i=1}^{n}[\beta_k]_i\left\|\sum_{\ell=1}^{n}t_{y,\ell}^k - n\hat{d}_y^k\right\|^2
$$
$$
\leq 2S(\mathbf{t}_y,\beta_k) + 4(L_{f,1}^2 + \lambda^2 L_{g,1}^2)\frac{n}{\min(\alpha_k)}\mathbf{V}_D^k. \tag{55}
$$

Then we have

$$
\left\|\hat{y}^{k+1} - y_\lambda^*(\hat{x}^{k+1})\right\|^2 \leq (1+\delta_{k,3})\Bigg((1+\delta_{k,4})q_k(\eta_y^k)\left\|\hat{y}^k - y_\lambda^*(\hat{x}^k)\right\|^2 + (1+\frac{1}{\delta_{k,4}})\eta_y^{k\,2}\big(2S(\mathbf{t}_y,\beta_k)
$$
$$
+ 8(L_{f,1}^2 + \lambda^2 L_{g,1}^2)\frac{n}{\min(\alpha_k)}\mathbf{V}_D^k\big)\Bigg) + \frac{9L_{g,1}^2}{\mu^2}\big(1+\frac{1}{\delta_{k,3}}\big)\left\|\hat{x}^{k+1} - \hat{x}^k\right\|^2
$$

$\square$

## D.5. Analysis of Weighted Dispersion

Before establishing the global convergence of Algorithm 1, we must quantify the behavior of the internal error processes. Specifically, establishing bounds on the consensus error $D(\cdot,\alpha_k)$ and the gradient tracking error $S(\cdot,\beta_k)$ is crucial for two reasons:

*1.The consensus of decision variables* : We need to ensure that the local decision variables asymptotically agree with their weighted averages (i.e., $D \to 0$).

*2. The consensus of tracking variables* : We must verify that the auxiliary tracking variables accurately estimate the global gradients (i.e., $S \to 0$).

In this section, we derive the recurrence relationships for these quantities, showing that they contract linearly up to a perturbation term controlled by the step sizes. We define the *stacked weighted average vector* as:

$$
\hat{\mathbf{x}}^k := (\hat{x}^k,\dots,\hat{x}^k)^\top \in \mathbb{R}^{n\times d_x}, \quad \hat{\mathbf{y}}^k := (\hat{y}^k,\dots,\hat{y}^k)^\top \in \mathbb{R}^{n\times d_y}, \quad \hat{\mathbf{z}}^k := (\hat{z}^k,\dots,\hat{z}^k)^\top \in \mathbb{R}^{n\times d_y}, \tag{56}
$$

where $\hat{\xi}^k = \sum[\alpha_k]_i\xi_i^k$, with $\xi \in x,y,z$, which can be found in (29).

**Lemma D.8.** *Let Assumptions 3.1, 3.2, and 3.3 hold. For any variable $\xi \in \{x,y,z\}$ and its stacked form $\boldsymbol{\xi} \in \{\mathbf{x},\mathbf{y},\mathbf{z}\}$, for all $k \geq 0$, we have:*

$$
D(\boldsymbol{\xi}^{k+1},\alpha_{k+1}) \leq (1-c_k)D(\boldsymbol{\xi}^k,\alpha_k)
$$
$$
+ \frac{2}{c_k}(\eta_\xi^k)^2\max_{j\in[n]}([\alpha_{k+1}]_j[\beta_k]_j)\left(S(\mathbf{t}_\xi^k,\beta_k) + \left\|\sum_{\ell=1}^{n}d_{\xi,\ell}^k\right\|^2\right), \tag{57}
$$

*where $D(\boldsymbol{\xi}^k,\alpha_k)$ and $S(\mathbf{t}_\xi^k,\beta_k)$ are defined in (29), the stacked variables $\boldsymbol{\xi}^k$ follow (26), and the local gradients $d_{\xi,\ell}^k$ are given in (25). The weight vectors $\alpha_k$ and $\beta_k$ are specified in Lemmas D.4 and D.5, respectively. Furthermore, the contraction parameter $c_k$ is defined as*

$$
c_k := \frac{\min(\alpha_{k+1})a^2}{\max^2(\alpha_k)\mathrm{D}(\mathcal{G}_k)\mathrm{K}(\mathcal{G}_k)}, \quad 0 < c_k < 1,
$$

*with $\mathrm{D}(\mathcal{G}^k)$ and $\mathrm{K}(\mathcal{G}^k)$ given in Definitions D.2 and D.3.*

*Proof.* We define the intermediate mixing variables $z_{\xi,i}^k$ and the corresponding stacked vectors $\mathbf{z}_\xi^k$ for any generic variable $\xi \in \{x,y,z\}$ as:

$$
z_{\xi,i}^k := \sum_{j=1}^{n}[A^k]_{ij}\xi_j^k, \quad \mathbf{z}_\xi^k := (z_{\xi,1}^k,\dots,z_{\xi,n}^k)^\top.
$$

Then, the update rules for the decision and auxiliary variables can be written compactly as:

$$\boldsymbol{\xi}^{k+1} = \mathbf{z}_\xi^k - \eta_\xi^k \mathbf{t}_\xi^k, \quad \text{for } \xi \in \{x, y, z\}. \tag{58}$$

Additionally, let $\mathbf{v}_\xi^k$ be the stacked vector where each block is the weighted average of the tracking variable $\mathbf{t}_\xi^k$ with respect to weights $\alpha_{k+1}$:

$$\mathbf{v}_\xi^k := \left( \sum_{j=1}^n [\alpha_{k+1}]_j t_{\xi,j}^k, \ldots, \sum_{j=1}^n [\alpha_{k+1}]_j t_{\xi,j}^k \right)^\top \in \mathbb{R}^{n \times d_\xi}.$$

Using these definitions and Eq. (31), the dynamics of the weighted averages satisfy:

$$\hat{\boldsymbol{\xi}}^{k+1} = \hat{\boldsymbol{\xi}}^k - \eta_\xi^k \mathbf{v}_\xi^k, \quad \text{for } \hat{\boldsymbol{\xi}} \in \{\hat{\mathbf{x}}, \hat{\mathbf{y}}, \hat{\mathbf{z}}\}. \tag{59}$$

Subtracting (59) from (58), we obtain the recurrence relation for the consensus error:

$$\boldsymbol{\xi}^{k+1} - \hat{\boldsymbol{\xi}}^{k+1} = \mathbf{z}_\xi^k - \hat{\boldsymbol{\xi}}^k - \eta_\xi^k (\mathbf{t}_\xi^k - \mathbf{v}_\xi^k). \tag{60}$$

Taking the squared $\alpha_{k+1}$-norm on both sides of (60) and applying Young's inequality with parameter $\delta_{k,5} > 0$, we obtain:

$$\begin{aligned}
D(\boldsymbol{\xi}^{k+1}, \alpha_{k+1}) \leq &(1 + \delta_{k,5}) \sum_{i=1}^n [\alpha_{k+1}]_i \big\| z_{\xi,i}^k - \hat{\xi}^k \big\|^2 \\
&+ \left( 1 + \frac{1}{\delta_{k,5}} \right) (\eta_\xi^k)^2 \sum_{i=1}^n [\alpha_{k+1}]_i \bigg\| t_{\xi,i}^k - \sum_{j=1}^n [\alpha_{k+1}]_j t_{\xi,j}^k \bigg\|^2.
\end{aligned} \tag{61}$$

For the mixing term $\sum_{i=1}^n [\alpha_{k+1}]_i \big\| z_{\xi,i}^k - \hat{\xi}^k \big\|^2$, invoking Eq. (19) from Nedić (2025) yields:

$$\sum_{i=1}^n [\alpha_{k+1}]_i \big\| z_{\xi,i}^k - \hat{\xi}^k \big\|^2 \leq (1 - c_k) D(\boldsymbol{\xi}^k, \alpha_k), \tag{62}$$

where the contraction coefficient $c_k = \frac{\min(\alpha_{k+1})a^2}{\max^2(\alpha_k)D(\mathbb{G}_k)K(\mathbb{G}_k)}$, here $0 < c_k < 1$.

Substituting (62) into the previous recursion, we obtain:

$$\begin{aligned}
D(\boldsymbol{\xi}^{k+1}, \alpha_{k+1}) \leq &(1 + \delta_{k,5})(1 - c_k) D(\boldsymbol{\xi}^k, \alpha_k) \\
&+ \left( 1 + \frac{1}{\delta_{k,5}} \right) (\eta_\xi^k)^2 \sum_{i=1}^n [\alpha_{k+1}]_i \bigg\| t_{\xi,i}^k - \sum_{j=1}^n [\alpha_{k+1}]_j t_{\xi,j}^k \bigg\|^2.
\end{aligned} \tag{63}$$

Next, we bound the second term on the R.H.S. of (63). By relating the distributions $\alpha_{k+1}$ and $\beta_k$, and using the property that the variance is bounded by the second moment, we have:

$$\begin{aligned}
\sum_{i=1}^n [\alpha_{k+1}]_i \bigg\| t_{\xi,i}^k - \sum_{j=1}^n [\alpha_{k+1}]_j t_{\xi,j}^k \bigg\|^2 &\leq \sum_{i=1}^n [\alpha_{k+1}]_i \| t_{\xi,i}^k \|^2 \\
&= \sum_{i=1}^n [\alpha_{k+1}]_i [\beta_k]_i \frac{\| t_{\xi,i}^k \|^2}{[\beta_k]_i} \\
&\leq \max_{j \in [n]} ([\alpha_{k+1}]_j [\beta_k]_j) \sum_{i=1}^n \frac{\| t_{\xi,i}^k \|^2}{[\beta_k]_i}.
\end{aligned} \tag{64}$$

Now, we rewrite the summation term on the R.H.S. of (64). By viewing $\frac{t^k_{\xi,i}}{[\beta_k]_i}$ as a random variable distributed according to $\beta_k$, we apply the bias-variance decomposition $\mathbb{E}[\|X\|^2] = \text{Var}(X) + \|\mathbb{E}[X]\|^2$:

$$
\begin{aligned}
\sum_{i=1}^n \frac{\|t^k_{\xi,i}\|^2}{[\beta_k]_i} &= \sum_{i=1}^n [\beta_k]_i \left\| \frac{t^k_{\xi,i}}{[\beta_k]_i} \right\|^2 \\
&= \sum_{i=1}^n [\beta_k]_i \left\| \frac{t^k_{\xi,i}}{[\beta_k]_i} - \sum_{\ell=1}^n t^k_{\xi,\ell} \right\|^2 + \left\| \sum_{\ell=1}^n t^k_{\xi,\ell} \right\|^2 \\
&= S(\mathbf{t}^k_\xi, \beta_k) + \left\| \sum_{\ell=1}^n t^k_{\xi,\ell} \right\|^2 .
\end{aligned}
\tag{65}
$$

Combining (64) and (65), using the gradient tracking conservation property $\sum_{\ell=1}^n t^k_{\xi,\ell} = \sum_{\ell=1}^n d^k_{\xi,\ell}$, and setting $\delta_{k,5} = c_k$ ($0 < c_k < 1$), we arrive at the desired bound in Lemma D.8.

$\square$

Next, we analyze the descent property of the tracking error $S(\mathbf{t}^{k+1}_\xi, \beta_{k+1})$. We define the global sum of the tracking variables as $s^k_\xi := \sum_{\ell=1}^n t^k_{\xi,\ell}$ and let $\mathbf{s}^k_\xi := (s^k_\xi, \ldots, s^k_\xi)^\top$. Additionally, we define $\mathbf{d}^k_\xi := (d^k_{\xi,1}, \ldots, d^k_{\xi,n})^\top$, where the elements are given in (25).

**Lemma D.9.** *Let Assumptions 3.1–3.3 hold. For all $k \geq 0$, we have:*

$$
S(\mathbf{t}^{k+1}_x, \beta_{k+1}) \leq (1 - e_k) S(\mathbf{t}^k_x, \beta_k) + \frac{4\gamma}{e_k}(3L^2_{f,1} + 6\lambda^2 L^2_{g,1})\mathbf{V}^k_c,
\tag{66}
$$

$$
S(\mathbf{t}^{k+1}_y, \beta_{k+1}) \leq (1 - e_k) S(\mathbf{t}^k_y, \beta_k) + \frac{8\gamma}{e_k}(L^2_{f,1} + \lambda^2 L^2_{g,1})\mathbf{V}^k_c,
\tag{67}
$$

$$
S(\mathbf{t}^{k+1}_z, \beta_{k+1}) \leq (1 - e_k) S(\mathbf{t}^k_z, \beta_k) + \frac{4\gamma}{e_k}\lambda^2 L^2_{g,1}\mathbf{V}^k_c,
\tag{68}
$$

*where $S(\mathbf{t}^k_\xi, \beta_k)$ is defined in (29), $\beta_k$ is given in Lemma D.5, and $\mathbf{V}^k_c := \|\mathbf{x}^{k+1} - \mathbf{x}^k\|^2 + \|\mathbf{y}^{k+1} - \mathbf{y}^k\|^2 + \|\mathbf{z}^{k+1} - \mathbf{z}^k\|^2$ denotes the sum of squared iterate differences, where the stacked variables $\boldsymbol{\xi}^k$, for $\boldsymbol{\xi} \in \{\mathbf{x}, \mathbf{y}, \mathbf{z}\}$, are defined in (26). Additionally, $\gamma := \max_k(n + \frac{1}{\min(\beta_{k+1})})$, and the coefficient $e_k \in (0, 1)$ is defined as*

$$
e_k = \frac{\min^2(\beta_k) b^2}{\max^2(\beta_k) \max(\beta_{k+1}) \mathrm{D}(\mathcal{G}^k) \mathrm{K}(\mathcal{G}^k)},
$$

*where $\mathrm{D}(\mathcal{G}^k)$ and $\mathrm{K}(\mathcal{G}^k)$ are given in Definitions D.2 and D.3, respectively.*

*Proof.* Define the mixed intermediate vectors $\mathbf{w}^k_\xi$ for any $\xi \in \{x, y, z\}$ as:

$$
\mathbf{w}^k_\xi := [w^k_{\xi,1}, \ldots, w^k_{\xi,n}]^\top, \quad \text{with } w^k_{\xi,i} = \sum_{j=1}^n b^k_{ij} t^k_{\xi,j}.
$$

Taking the $\beta_{k+1}$-induced norm on both sides of the preceding equality and using the relation between $S(\mathbf{t}^{k+1}_x, \beta_{k+1})$ and the weighted norm, we have:

$$
\begin{aligned}
S(\mathbf{t}^{k+1}_x, \beta_{k+1}) &= \left\| \mathbf{w}^k_x \mathrm{diag}^{-1}(\beta_{k+1}) - \mathbf{s}^k_x + (\mathbf{s}^k_x - \mathbf{s}^{k+1}_x) + (\mathbf{d}^{k+1}_x - \mathbf{d}^k_x)\mathrm{diag}^{-1}(\beta_{k+1}) \right\|^2_{\beta_{k+1}} \\
&\leq (1 + \delta_{k,6}) \left\| \mathbf{w}^k_x \mathrm{diag}^{-1}(\beta_{k+1}) - \mathbf{s}^k_x \right\|^2_{\beta_{k+1}} \\
&\quad + \left( 1 + \frac{1}{\delta_{k,6}} \right) \left( 2\|\mathbf{s}^k_x - \mathbf{s}^{k+1}_x\|^2_{\beta_{k+1}} + 2\|(\mathbf{d}^{k+1}_x - \mathbf{d}^k_x)\mathrm{diag}^{-1}(\beta_{k+1})\|^2_{\beta_{k+1}} \right).
\end{aligned}
\tag{69}
$$

We next consider the term $\|\mathbf{w}_x^k \text{diag}^{-1}(\beta_{k+1}) - \mathbf{s}_x^k\|_{\beta_{k+1}}$. Using the definitions $\mathbf{w}_x^k = (w_{x,1}^k, \ldots, w_{x,n}^k)^\top$ and $\mathbf{s}_x^k = (s_x^k, \ldots, s_x^k)^\top$, we have:

$$\|\mathbf{w}_x^k \text{diag}^{-1}(\beta_{k+1}) - \mathbf{s}_x^k\|_{\beta_{k+1}}^2 = \sum_{i=1}^n [\beta_{k+1}]_i \left\| \frac{w_{x,i}^k}{[\beta_{k+1}]_i} - s_x^k \right\|^2.$$

Invoking the mixing contraction property (e.g., Lemma 5.5 in Nedić (2025)), yields:

$$\sum_{i=1}^n [\beta_{k+1}]_i \left\| \frac{w_{x,i}^k}{[\beta_{k+1}]_i} - \sum_{\ell=1}^n t_{x,\ell}^k \right\|^2 \le (1-e_k) \sum_{i=1}^n [\beta_k]_i \left\| \frac{t_{x,i}^k}{[\beta_k]_i} - \sum_{\ell=1}^n t_{x,\ell}^k \right\|^2,$$

where the contraction coefficient is given by $e_k = \frac{\min^2(\beta_k)b^2}{\max^2(\beta_k)\max(\beta_{k+1})\text{D}(G_k)\text{K}(G_k)}$, here $0 < e_k < 1$. Consequently, we obtain:

$$\|\mathbf{w}_x^k \text{diag}^{-1}(\beta_{k+1}) - \mathbf{s}_x^k\|_{\beta_{k+1}}^2 \le (1-e_k) \sum_{i=1}^n [\beta_k]_i \left\| \frac{t_{x,i}^k}{[\beta_k]_i} - \sum_{\ell=1}^n t_{x,\ell}^k \right\|^2 = (1-e_k)S(\mathbf{t}_x^k, \beta_k).$$

Substituting the preceding contraction relation back into (69), we have:

$$
\begin{aligned}
S(\mathbf{t}_x^{k+1}, \beta_{k+1}) \le {} & (1+\delta_{k,6})(1-e_k)S(\mathbf{t}_x^k, \beta_k) \\
& + 2\left(1 + \frac{1}{\delta_{k,6}}\right)\|\mathbf{s}_x^k - \mathbf{s}_x^{k+1}\|_{\beta_{k+1}}^2 \\
& + 2\left(1 + \frac{1}{\delta_{k,6}}\right)\|(\mathbf{d}_x^{k+1} - \mathbf{d}_x^k)\text{diag}^{-1}(\beta_{k+1})\|_{\beta_{k+1}}^2.
\end{aligned}
\tag{70}
$$

Next, we analyze the drift terms. First, for the consensus term of sums $\|\mathbf{s}_x^k - \mathbf{s}_x^{k+1}\|_{\beta_{k+1}}^2$, using the fact that $\sum_{i=1}^n [\beta_{k+1}]_i = 1$, we have:

$$\|\mathbf{s}_x^k - \mathbf{s}_x^{k+1}\|_{\beta_{k+1}}^2 = \sum_{i=1}^n [\beta_{k+1}]_i \|s_x^{k+1} - s_x^k\|^2 = \|s_x^{k+1} - s_x^k\|^2.$$

Since $B^k$ is column stochastic, the total mass is conserved up to the gradient change, i.e., $s_x^k = \sum_{\ell=1}^n t_{x,\ell}^k = \sum_{\ell=1}^n d_{x,\ell}^k$. This implies:

$$\|s_x^{k+1} - s_x^k\|^2 = \left\| \sum_{\ell=1}^n (d_{x,\ell}^{k+1} - d_{x,\ell}^k) \right\|^2 \le n\|\mathbf{d}_x^{k+1} - \mathbf{d}_x^k\|^2,$$

where the last inequality follows from Cauchy-Schwarz inequality $\|\sum v_i\|^2 \le n \sum \|v_i\|^2$.

For the weighted gradient difference term, we have:

$$\|(\mathbf{d}_x^{k+1} - \mathbf{d}_x^k)\text{diag}^{-1}(\beta_{k+1})\|_{\beta_{k+1}}^2 = \sum_{i=1}^n \frac{\|d_{x,i}^{k+1} - d_{x,i}^k\|^2}{[\beta_{k+1}]_i} \le \frac{1}{\min(\beta_{k+1})}\|\mathbf{d}_x^{k+1} - \mathbf{d}_x^k\|^2.$$

Substituting these bounds back into the recurrence of $S(\mathbf{t}_x^{k+1}, \beta_{k+1})$, we obtain:

$$
\begin{aligned}
S(\mathbf{t}_x^{k+1}, \beta_{k+1}) \le {} & (1+\delta_{k,6})(1-e_k)S(\mathbf{t}_x^k, \beta_k) \\
& + 2\left(n + \frac{1}{\min(\beta_{k+1})}\right)\left(1 + \frac{1}{\delta_{k,6}}\right)\|\mathbf{d}_x^{k+1} - \mathbf{d}_x^k\|^2.
\end{aligned}
$$

Similarly, for the tracking errors of $y$ and $z$, we have:

$$
\begin{aligned}
S(\mathbf{t}_y^{k+1}, \beta_{k+1}) \le {} & (1+\delta_{k,6})(1-e_k)S(\mathbf{t}_y^k, \beta_k) \\
& + 2\left(n + \frac{1}{\min(\beta_{k+1})}\right)\left(1 + \frac{1}{\delta_{k,6}}\right)\|\mathbf{d}_y^{k+1} - \mathbf{d}_y^k\|^2, \\
S(\mathbf{t}_z^{k+1}, \beta_{k+1}) \le {} & (1+\delta_{k,6})(1-e_k)S(\mathbf{t}_z^k, \beta_k) \\
& + 2\left(n + \frac{1}{\min(\beta_{k+1})}\right)\left(1 + \frac{1}{\delta_{k,6}}\right)\|\mathbf{d}_z^{k+1} - \mathbf{d}_z^k\|^2.
\end{aligned}
$$

Finally, invoking the $L$-smoothness of $f_i$ and $g_i$ (Assumption 3.3) and recalling $\mathbf{V}_c^k := \|\mathbf{x}^{k+1} - \mathbf{x}^k\|^2 + \|\mathbf{y}^{k+1} - \mathbf{y}^k\|^2 + \|\mathbf{z}^{k+1} - \mathbf{z}^k\|^2$, we bound the gradient differences as:

$$\|\mathbf{d}_x^{k+1} - \mathbf{d}_x^k\|^2 \le (3L_{f,1}^2 + 6\lambda^2 L_{g,1}^2)\mathbf{V}_c^k,$$
$$\|\mathbf{d}_y^{k+1} - \mathbf{d}_y^k\|^2 \le 2(L_{f,1}^2 + \lambda^2 L_{g,1}^2)\mathbf{V}_c^k,$$
$$\|\mathbf{d}_z^{k+1} - \mathbf{d}_z^k\|^2 \le \lambda^2 L_{g,1}^2 \mathbf{V}_c^k.$$

Substituting these gradient bounds into the respective error recursions and setting $\delta_{k,6} = e_k$ completes the proof. □

To bound the terms $\|\mathbf{x}^{k+1} - \mathbf{x}^k\|^2$, $\|\mathbf{y}^{k+1} - \mathbf{y}^k\|^2$, and $\|\mathbf{z}^{k+1} - \mathbf{z}^k\|^2$ appearing in $\mathbf{V}_c^k$, the following lemma establishes the necessary estimates for the iterate differences of both decision and auxiliary variables.

**Lemma D.10.** *Let Assumptions 3.1, 3.2 and 3.3 hold. For any variable $\xi \in \{x, y, z\}$ and its stacked form $\boldsymbol{\xi} \in \{\mathbf{x}, \mathbf{y}, \mathbf{z}\}$, for all $k \ge 0$, we have:*

$$\left\|\boldsymbol{\xi}^{k+1} - \boldsymbol{\xi}^k\right\|^2 \le 3\left(\frac{c_k}{\min(\alpha_{k+1})} + \frac{1}{\min(\alpha_k)}\right) D(\boldsymbol{\xi}^k, \alpha_k)$$
$$+ 3(\eta_\xi^k)^2 \left(S(\mathbf{t}_\xi^k, \beta_k) + \left\|\sum_{i=1}^n d_{\xi,i}^k\right\|^2\right). \tag{71}$$

*Here, $\boldsymbol{\xi}^k$ is defined in (26), and $d_{\xi,i}^k$ follows (25). The scalars $\alpha_k$ and $c_k$ are given in Lemmas D.4 and D.8, respectively, while $D(\boldsymbol{\xi}^k, \alpha_k)$ and $S(\mathbf{t}_\xi^k, \beta_k)$ are defined in (29) and (30).*

*Proof.* Recalling the compact update rule from (58), we have $\boldsymbol{\xi}^{k+1} = \mathbf{z}_\xi^k - \eta_\xi^k \mathbf{t}_\xi^k$. Subtracting $\boldsymbol{\xi}^k$ from both sides and introducing the intermediate weighted average $\hat{\boldsymbol{\xi}}^k$ (where $\hat{\boldsymbol{\xi}}^k = \mathbf{1} \otimes \hat{\xi}^k$), we decompose the difference as:

$$\boldsymbol{\xi}^{k+1} - \boldsymbol{\xi}^k = \mathbf{z}_\xi^k - \boldsymbol{\xi}^k - \eta_\xi^k \mathbf{t}_\xi^k$$
$$= (\mathbf{z}_\xi^k - \hat{\boldsymbol{\xi}}^k) + (\hat{\boldsymbol{\xi}}^k - \boldsymbol{\xi}^k) - \eta_\xi^k \mathbf{t}_\xi^k.$$

Using the inequality $\|a + b + c\|^2 \le 3(\|a\|^2 + \|b\|^2 + \|c\|^2)$, we obtain:

$$\|\boldsymbol{\xi}^{k+1} - \boldsymbol{\xi}^k\|^2 \le 3\|\mathbf{z}_\xi^k - \hat{\boldsymbol{\xi}}^k\|^2 + 3\|\boldsymbol{\xi}^k - \hat{\boldsymbol{\xi}}^k\|^2 + 3(\eta_\xi^k)^2\|\mathbf{t}_\xi^k\|^2. \tag{72}$$

We now bound each term on the R.H.S. of (72):

Using the norm equivalence $\|\mathbf{u}\|^2 \le \frac{1}{\min(\alpha)}\|\mathbf{u}\|_\alpha^2$ (from (27)) and the mixing contraction property (Eq. (62)), we have:

$$\|\mathbf{z}_\xi^k - \hat{\boldsymbol{\xi}}^k\|^2 \le \frac{1}{\min(\alpha_{k+1})}\|\mathbf{z}_\xi^k - \hat{\boldsymbol{\xi}}^k\|_{\alpha_{k+1}}^2 \le \frac{c_k}{\min(\alpha_{k+1})}D(\boldsymbol{\xi}^k, \alpha_k).$$

Similarly, applying the norm equivalence to the consensus error yields:

$$\|\boldsymbol{\xi}^k - \hat{\boldsymbol{\xi}}^k\|^2 \le \frac{1}{\min(\alpha_k)}\|\boldsymbol{\xi}^k - \hat{\boldsymbol{\xi}}^k\|_{\alpha_k}^2 = \frac{1}{\min(\alpha_k)}D(\boldsymbol{\xi}^k, \alpha_k).$$

Utilizing the relationship between the Euclidean norm and the $\beta^{-1}$-weighted norm (Eq. (28)), and the decomposition derived in (65), we have:

$$\|\mathbf{t}_\xi^k\|^2 \le \|\mathbf{t}_\xi^k\|_{\beta_k^{-1}}^2 = \sum_{i=1}^n \frac{\|t_{\xi,i}^k\|^2}{[\beta_k]_i} = S(\mathbf{t}_\xi^k, \beta_k) + \left\|\sum_{i=1}^n d_{\xi,i}^k\right\|^2,$$

where the last equality uses the conservation property $\sum t_{\xi,i}^k = \sum d_{\xi,i}^k$.

Substituting these three bounds back into (72) yields the desired result. □

**Lemma D.11.** *Under Assumptions 3.1–3.3, the squared norm of the sum of local gradients is bounded as follows:*

$$\left\|\sum_{i=1}^{n} d_{x,i}^k\right\|^2 \leq (12L_{f,1}^2 + 24\lambda^2 L_{g,1}^2)\frac{n}{\min(\alpha_k)}\mathbf{V}_D^k + 4n^2\lambda^2 L_{g,1}^2\|\hat{z}^k - y^*(\hat{x}^k)\|^2$$

$$+ 8n^2(L_{f,1}^2 + \lambda^2 L_{g,1}^2)\|\hat{y}^k - y_\lambda^*(\hat{x}^k)\|^2 + 4n^2\|\nabla_x\mathcal{L}_\lambda^*(\hat{x}^k)\|^2; \tag{73a}$$

$$\left\|\sum_{i=1}^{n} d_{y,i}^k\right\|^2 \leq 4(L_{f,1}^2 + \lambda^2 L_{g,1}^2)\frac{n}{\min(\alpha_k)}\mathbf{V}_D^k$$

$$+ 4n^2(L_{f,1}^2 + \lambda^2 L_{g,1}^2)\|\hat{y}^k - y_\lambda^*(\hat{x}^k)\|^2; \tag{73b}$$

$$\left\|\sum_{i=1}^{n} d_{z,i}^k\right\|^2 \leq 2\lambda^2 L_{g,1}^2\frac{n}{\min(\alpha_k)}\mathbf{V}_D^k + 2n^2\lambda^2 L_{g,1}^2\|\hat{z}^k - y^*(\hat{x}^k)\|^2, \tag{73c}$$

*where $\mathbf{V}_D^k$ denotes the cumulative consensus error as defined in (30), and $\alpha_k$ is given in Lemma D.4.*

*Proof.* We derive the bounds for each variable separately. The bound for $\left\|\sum_{i=1}^{n} d_{x,i}^k\right\|^2$ is established by isolating the consensus violation and the optimality residuals from the true gradient component:

$$\left\|\sum_{i=1}^{n} d_{x,i}^k\right\|^2 \leq 4\left\|\sum_{i=1}^{n} d_{x,i}^k - n\nabla_x\mathcal{L}_\lambda(\hat{x}^k, \hat{y}^k, \hat{z}^k)\right\|^2$$

$$+ 4n^2\|\nabla_x\mathcal{L}_\lambda(\hat{x}^k, \hat{y}^k, \hat{z}^k) - \nabla_x\mathcal{L}_\lambda(\hat{x}^k, \hat{y}^k, y^*(\hat{x}^k))\|^2$$

$$+ 4n^2\|\nabla_x\mathcal{L}_\lambda(\hat{x}^k, \hat{y}^k, y^*(\hat{x}^k)) - \nabla_x\mathcal{L}_\lambda^*(\hat{x}^k)\|^2$$

$$+ 4n^2\|\nabla_x\mathcal{L}_\lambda^*(\hat{x}^k)\|^2.$$

Applying Lipschitz continuity of the gradients (Assumption 3.3), we obtain:

$$\left\|\sum_{i=1}^{n} d_{x,i}^k\right\|^2 \leq (12L_{f,1}^2 + 24\lambda^2 L_{g,1}^2)\frac{n}{\min(\alpha_k)}\mathbf{V}_D^k$$

$$+ 4n^2\lambda^2 L_{g,1}^2\|\hat{z}^k - y^*(\hat{x}^k)\|^2$$

$$+ 8n^2(L_{f,1}^2 + \lambda^2 L_{g,1}^2)\|\hat{y}^k - y_\lambda^*(\hat{x}^k)\|^2$$

$$+ 4n^2\|\nabla_x\mathcal{L}_\lambda^*(\hat{x}^k)\|^2,$$

where we used the fact that $\nabla_x\mathcal{L}_\lambda$ is Lipschitz continuous with respect to $y$ with constant $L_{f,1}+\lambda L_{g,1}$, and $(L_{f,1}+\lambda L_{g,1})^2 \leq 2(L_{f,1}^2 + \lambda^2 L_{g,1}^2)$.

Using the optimality condition $\nabla_y\mathcal{L}_\lambda(\hat{x}^k, y_\lambda^*(\hat{x}^k), \hat{z}^k) = 0$, we have:

$$\left\|\sum_{i=1}^{n} d_{y,i}^k\right\|^2 \leq 2\left\|\sum_{i=1}^{n} d_{y,i}^k - n\nabla_y\mathcal{L}_\lambda(\hat{x}^k, \hat{y}^k, \hat{z}^k)\right\|^2$$

$$+ 2n^2\|\nabla_y\mathcal{L}_\lambda(\hat{x}^k, \hat{y}^k, \hat{z}^k) - \nabla_y\mathcal{L}_\lambda(\hat{x}^k, y_\lambda^*(\hat{x}^k), \hat{z}^k)\|^2$$

$$\leq 4(L_{f,1}^2 + \lambda^2 L_{g,1}^2)\frac{n}{\min(\alpha_k)}\mathbf{V}_D^k$$

$$+ 4n^2(L_{f,1}^2 + \lambda^2 L_{g,1}^2)\|\hat{y}^k - y_\lambda^*(\hat{x}^k)\|^2.$$

Similarly, comparing the gradient at the current iterate and the optimal response $y^*(\hat{x}^k)$ (note: $\nabla_z$ does not depend on $y$):

$$\left\|\sum_{i=1}^n d_{z,i}^k\right\|^2 \le 2\left\|\sum_{i=1}^n d_{z,i}^k - n\nabla_z \mathcal{L}_\lambda(\hat{x}^k, \hat{y}^k, \hat{z}^k)\right\|^2$$
$$+ 2n^2\left\|\nabla_z \mathcal{L}_\lambda(\hat{x}^k, \hat{y}^k, \hat{z}^k) - \nabla_z \mathcal{L}_\lambda(\hat{x}^k, \hat{y}^k, y^*(\hat{x}^k))\right\|^2$$
$$\le 2\lambda^2 L_{g,1}^2 \frac{n}{\min(\alpha_k)}\mathbf{V}_D^k + 2n^2\lambda^2 L_{g,1}^2\left\|\hat{z}^k - y^*(\hat{x}^k)\right\|^2.$$

$\square$

## D.6. Descent in the Lyapunov Function

To establish the convergence of the proposed algorithm, we construct a suitable Lyapunov function and investigate its descent properties. We begin by introducing the necessary assumptions on the weight vectors and defining the auxiliary notations. Specifically, we assume that the weight vectors derived from the mixing matrices conform to the following bounds:

$$\frac{\underline{\alpha}}{n} \le [\alpha_k]_i \le \frac{\overline{\alpha}}{n} \le 1, \quad \frac{\underline{\beta}}{n} \le [\beta_k]_i \le \frac{\overline{\beta}}{n} \le 1, \tag{74}$$

where $\underline{\alpha} := a^n$ and $\underline{\beta} := b^n$. We further define the auxiliary contraction parameters $\delta_{k,1} = \delta_{k,2} := 1 - \lambda\eta_z^k\underline{q}$ and $\delta_{k,3} = \delta_{k,4} := 1 - \lambda\eta_y^k\underline{q}$.

Furthermore, following Nedić et al., 2025, Eq. (40), we denote $\underline{c}$ and $\underline{e}$ as the uniform lower bounds for the contraction coefficients $c_k$ and $e_k$ defined in Lemma D.8 and Lemma D.9, respectively. In summary, these constants are defined as:

$$\underline{q} := \min\left\{n\min(\beta_k)\mu,\ n\min(\beta_k)L_{g,1}\right\}, \quad \underline{c} := \min_k\{c_k\}, \quad \underline{e} := \min_k\{e_k\}. \tag{75}$$

We construct the Lyapunov function for the bilevel algorithm FAB as a combination of the objective value, the approximation errors, and the consensus and tracking measures:

$$\Phi^{(b)}(\hat{x}^k) := \mathcal{L}_\lambda^*(\hat{x}^k) + C_{b,1}\|\hat{z}^k - y^*(\hat{x}^k)\|^2 + C_{b,2}\|\hat{y}^k - y_\lambda^*(\hat{x}^k)\|^2$$
$$+ C_{b,3}\frac{n}{\underline{\alpha}}\mathbf{V}_D^k + C_{b,4}\mathbf{V}_S^k, \tag{76}$$

where $\mathbf{V}_D^k$ and $\mathbf{V}_S^k$ are defined in (30). To decouple the error dynamics, the coupling coefficients are carefully chosen as:

$$C_{b,1} := \frac{\eta_x^k\lambda}{\eta_z^k}\mathcal{C}_{b,1}, \quad C_{b,2} := \frac{\eta_x^k\lambda}{\eta_y^k}\mathcal{C}_{b,2}, \quad C_{b,3} := \lambda\mathcal{C}_{b,3}, \quad C_{b,4} := \frac{1}{\lambda}, \tag{77}$$

with the time-invariant base constants given by:

$$\mathcal{C}_{b,1} := \frac{4L_{g,1}^2 n^2\overline{\alpha}^2\overline{\beta}}{\underline{q}\,\underline{\alpha}\underline{\beta}^2}, \quad \mathcal{C}_{b,2} := \frac{13L_{g,1}^2 n^2\overline{\alpha}^2\overline{\beta}}{\underline{q}\,\underline{\alpha}\underline{\beta}^2}, \quad \mathcal{C}_{b,3} := \frac{672\gamma L_{g,1}^2}{n\underline{e}}. \tag{78}$$

We are now in a position to analyze the evolution of the Lyapunov function. By evaluating the difference $\Phi^{(b)}(\hat{x}^{k+1}) - \Phi^{(b)}(\hat{x}^k)$ and substituting the descent property of the penalty function along with the contraction lemmas derived previously, we obtain the following expansion:

**Lemma D.12.** *Under Assumptions 3.1–3.3, suppose the penalty parameter $\lambda$ and the fixed step-sizes $\{\eta_x^k, \eta_y^k, \eta_z^k\}$ satisfy the following conditions.*

*First, the penalty parameter is bounded by:*

$$\frac{1}{\lambda} \le \min\left\{1, \frac{L_{g,1}}{L_{f,1}}, \frac{\mu}{2L_{f,1}}\right\}. \tag{79}$$

*Second, the step-sizes satisfy the absolute bounds:*

$$\max\{\eta_x^k, \eta_y^k, \eta_z^k\} \le \min\left\{1, \frac{(\overline{\alpha}\overline{\beta})^2}{16n^2\mathcal{C}_{aux}}, \frac{1}{8n^2\mathcal{C}_{aux}}, \frac{1}{4\overline{\alpha}\overline{\beta}L_{*,1}}\right\},$$ (80a)

$$\lambda\max\{\eta_x^k, \eta_y^k, \eta_z^k\} \le \min\left\{\frac{1}{\underline{q}}, \frac{\overline{\alpha}^2\overline{\beta}}{12\underline{\alpha}\underline{\beta}\mathcal{C}_{aux}}, \frac{\underline{\alpha}^2\underline{\beta}\underline{c}\mathcal{C}_{b,3}}{90nL_{g,1}^2}, \frac{\underline{\alpha}\underline{\beta}\underline{c}q\mathcal{C}_{b,3}}{120n\mathcal{C}_{b,1}L_{g,1}^2},\right.$$
$$\left.\frac{\underline{\alpha}\underline{\beta}\underline{c}q\mathcal{C}_{b,3}}{320n\mathcal{C}_{b,2}L_{g,1}^2}, \frac{\underline{\alpha}\underline{\beta}\underline{e}}{15}, \frac{\underline{q}\underline{\beta}\underline{e}}{20\mathcal{C}_{b,2}}, \frac{\underline{\alpha}\underline{\beta}}{16n^2\mathcal{C}_{aux}}\right\}.$$ (80b)

*Third, the step-sizes satisfy the relative ratio bounds:*

$$\frac{\lambda(\eta_x^k)^2}{\eta_y^k} \le \frac{\overline{\alpha}^2\overline{\beta}}{24\underline{\alpha}\underline{\beta}\mathcal{C}_{aux}}\left(\frac{\mu}{L_{g,1}}\sqrt{\frac{q}{432\underline{\alpha}\mathcal{C}_{b,2}}}\right)^3, \quad \frac{\eta_x^k}{\eta_y^k} \le \frac{\mu}{L_{g,1}}\sqrt{\frac{q}{432\underline{\alpha}\mathcal{C}_{b,2}}},$$ (81a)

$$\frac{\lambda(\eta_x^k)^2}{\eta_z^k} \le \frac{\overline{\alpha}^2\overline{\beta}}{6\underline{\alpha}\underline{\beta}\mathcal{C}_{aux}}\left(\frac{\mu}{L_{g,1}}\sqrt{\frac{q}{12\underline{\alpha}\mathcal{C}_{b,1}}}\right)^3, \quad \frac{\eta_x^k}{\eta_z^k} \le \frac{\mu}{L_{g,1}}\sqrt{\frac{q}{12\underline{\alpha}\mathcal{C}_{b,1}}},$$ (81b)

*where the sets of constants $\{\underline{\alpha}, \overline{\alpha}, \underline{\beta}, \overline{\beta}\}$, $\{q, \underline{c}, \underline{e}\}$, and $\{\mathcal{C}_{b,1}, \mathcal{C}_{b,2}, \mathcal{C}_{b,3}\}$ are defined in (74), (75), and (78), respectively. Furthermore, the auxiliary constant is defined as: $\mathcal{C}_{aux} := \frac{2C_{b,3}\overline{\alpha}\overline{\beta}}{\underline{c}} + \frac{168\gamma L_{g,1}^2}{\underline{e}}$. Then, the sequence generated by Algorithm 1 satisfies the descent inequality:*

$$\Phi^{(b)}(\hat{x}^{k+1}) - \Phi^{(b)}(\hat{x}^k) \le -\frac{1}{4}\underline{\alpha}\underline{\beta}\eta_x^k\|\nabla_x\mathcal{L}_\lambda^*(\hat{x}^k)\|^2 - \frac{1}{5}\underline{c}\mathcal{C}_{b,3}\lambda\frac{n}{\underline{\alpha}}\mathbf{V}_D^k.$$ (82)

*Proof.* Considering the definition of the Lyapunov function $\Phi^{(b)}$ and invoking Lemmas D.6, D.7, D.8, D.9, D.10 and D.11, and provided that $\frac{1}{\lambda} \le 1$ and $\max\{\eta_x^k, \eta_y^k, \eta_z^k\} \le 1$, we derive the following expansion for the difference $\Phi^{(b)}(\hat{x}^{k+1}) - \Phi^{(b)}(\hat{x}^k)$:

$$\Phi^{(b)}(\hat{x}^{k+1}) - \Phi^{(b)}(\hat{x}^k) \le C_{\Phi,1}^{(b)}\|\hat{x}^{k+1} - \hat{x}^k\|^2 + C_{\Phi,2}^{(b)}\|\hat{z}^k - y^*(\hat{x}^k)\|^2$$
$$+ C_{\Phi,3}^{(b)}\|\hat{y}^k - y_\lambda^*(\hat{x}^k)\|^2 + C_{\Phi,4}^{(b)}\frac{n}{\underline{\alpha}}\mathbf{V}_D^k$$
$$+ C_{\Phi,5}^{(b)}\mathbf{V}_S^k + C_{\Phi,6}^{(b)}\|\nabla_x\mathcal{L}_\lambda^*(\hat{x}^k)\|^2.$$ (83)

We now proceed to analyze the coefficients of each term in the expansion (83) to establish the descent property.

*1. Coefficient of the drift term $\|\hat{x}^{k+1} - \hat{x}^k\|^2$.* Collecting all terms involving the decision variable difference, we obtain the coefficient:

$$C_{\Phi,1}^{(b)} = -\frac{1}{\underline{\alpha}\,\underline{\beta}}\frac{1}{2\eta_x^k} + \frac{L_{*,1}}{2} + C_{b,1}\frac{2L_{g,1}^2}{\lambda\eta_z^k\underline{q}\,\underline{\beta}\mu^2} + C_{b,2}\frac{72L_{g,1}^2}{\lambda\eta_y^k\underline{q}\,\underline{\beta}\mu^2}.$$

*2. Coefficient of the unregularized estimation error $\|\hat{z}^k - y^*(\hat{x}^k)\|^2$.* Grouping the terms associated with the lower-level variable $z$, the coefficient is:

$$C_{\Phi,2}^{(b)} = \frac{\eta_x^k}{2}\frac{6\lambda^2L_{g,1}^2n^2\overline{\alpha}^2\overline{\beta}}{\underline{\alpha}\underline{\beta}} - C_{b,1}\lambda\eta_z^k\underline{q}\,\underline{\beta}$$
$$+ \left(\frac{2}{\underline{c}}C_{b,3}\overline{\alpha}\,\overline{\beta}\eta_x^{k^2} + 56C_{b,4}\frac{\gamma}{\underline{e}}\lambda^2L_{g,1}^23\eta_x^{k^2}\right)4n^2\lambda^2L_{g,1}^2$$
$$+ \left(\frac{2}{\underline{c}}C_{b,3}\overline{\alpha}\,\overline{\beta}\eta_z^{k^2} + 56C_{b,4}\frac{\gamma}{\underline{e}}\lambda^2L_{g,1}^23\eta_z^{k^2}\right)2n^2\lambda^2L_{g,1}^2.$$

3. *Coefficient of the regularized estimation error* $\|\hat{y}^k - y^*_\lambda(\hat{x}^k)\|^2$. *Similarly, for the variable $y$, the coefficient is:*

$$
\begin{aligned}
C^{(b)}_{\Phi,3} = {}& \frac{12\eta^k_x \lambda^2 L^2_{g,1} n^2 \overline{\alpha}^2 \overline{\beta}}{\underline{\alpha}\,\underline{\beta}} - C_{b,2}\lambda\eta^k_y \underline{q}\,\underline{\beta} \\
& + \left( \frac{2}{\underline{c}}C_{b,3}\overline{\alpha}\,\overline{\beta}\eta^{k^2}_x + 56C_{b,4}\frac{\gamma}{\underline{e}}\lambda^2 L^2_{g,1} 3\eta^{k^2}_x \right) 4n^2\lambda^2 L^2_{g,1} \\
& + \left( \frac{2}{\underline{c}}C_{b,3}\overline{\alpha}\,\overline{\beta}\eta^{k^2}_y + 56C_{b,4}\frac{\gamma}{\underline{e}}\lambda^2 L^2_{g,1} 3\eta^{k^2}_y \right) 8n^2\lambda^2 L^2_{g,1}.
\end{aligned}
$$

4. *Coefficient of the consensus error* $\frac{n}{\underline{\alpha}}\mathbf{V}^k_D$. *The coefficient for the consensus error consists of the negative contraction term from the graph topology and positive perturbations:*

$$
\begin{aligned}
C^{(b)}_{\Phi,4} = {}& \frac{1}{\underline{\alpha}\,\underline{\beta}}18\lambda^2 L^2_{g,1}\frac{\eta^k_x}{2} + C_{b,1}\frac{12}{\lambda\underline{q}\,\underline{\beta}}\eta^k_z\lambda^2 L^2_{g,1} \\
& + C_{b,2}\frac{32}{\lambda\underline{q}\,\underline{\beta}}\eta^k_y\lambda^2 L^2_{g,1} - C_{b,3}\underline{c} + 336C_{b,4}\frac{\gamma}{\underline{e}}\lambda^2 L^2_{g,1} \\
& + \left( \frac{2}{\underline{c}}C_{b,3}\overline{\alpha}\,\overline{\beta} + 168C_{b,4}\frac{\gamma}{\underline{e}}\lambda^2 L^2_{g,1} \right) 36\lambda^2 L^2_{g,1}(\eta^{k^2}_x + \eta^{k^2}_y + \eta^{k^2}_z).
\end{aligned}
$$

5. *Coefficient of the tracking error* $\mathbf{V}^k_S$. *The coefficient for the gradient tracking error is:*

$$
\begin{aligned}
C^{(b)}_{\Phi,5} = {}& \frac{3}{\underline{\alpha}\,\underline{\beta}}\eta^k_x + C_{b,1}\frac{4}{\lambda\underline{q}\,\underline{\beta}}\eta^k_z + C_{b,2}\frac{4}{\lambda\underline{q}\,\underline{\beta}}\eta^k_y - C_{b,4}\underline{e} \\
& + \left( \frac{2}{\underline{c}}C_{b,3}\overline{\alpha}\,\overline{\beta} + 168C_{b,4}\frac{\gamma}{\underline{e}}\lambda^2 L^2_{g,1} \right) (\eta^{k^2}_x + \eta^{k^2}_y + \eta^{k^2}_z).
\end{aligned}
$$

6. *Coefficient of the gradient norm* $\|\nabla_x \mathcal{L}^*_\lambda(\hat{x}^k)\|^2$. *Finally, the coefficient for the descent term is:*

$$
C^{(b)}_{\Phi,6} = \left( \frac{2}{\underline{c}}C_{b,3}\overline{\alpha}\,\overline{\beta} + 168C_{b,4}\frac{\gamma}{\underline{e}}\lambda^2 L^2_{g,1} \right) 4n^2\eta^{k^2}_x - \frac{\eta^k_x}{2}\underline{\alpha}\,\underline{\beta}.
$$

By ensuring that the step-sizes and the penalty parameter $\lambda$ satisfy conditions (79)–(81), we guarantee that the coefficients $C^{(b)}_{\Phi,1}$, $C^{(b)}_{\Phi,2}$, $C^{(b)}_{\Phi,3}$, and $C^{(b)}_{\Phi,5}$ are non-positive, while $C^{(b)}_{\Phi,4}$ and $C^{(b)}_{\Phi,6}$ are sufficiently negative to establish the desired descent inequality (82).

$\square$

## D.7. Convergence Rate of FAB

Building upon the Lyapunov descent property established in Lemma D.12, we are now well-positioned to derive the global convergence rate of Algorithm 1. The following theorem demonstrates that the algorithm converges to a stationary point of the hyper-objective function at a sublinear rate.

**Theorem D.13.** *Let Assumptions 3.1–3.3 hold. Suppose that the step sizes and the penalty parameter satisfy the conditions in Lemma D.12, subject to the following additional condition on $\lambda$:*

$$
\frac{1}{\lambda} \leq \frac{4n\underline{c}\mathcal{C}_{b,3}}{5L^2_{*,2}}, \tag{84}
$$

*where $\mathcal{C}_{b,3}$ and $\underline{c}$ are constants defined in (78) and (75), respectively, and the smoothness constant $L_{*,2}$ is given by:*

$$
L_{*,2} := \frac{\left( 2L_{f,1} + \frac{L_{f,1}L_{g,1}}{\mu} + L_{f,0}\left( \frac{2L_{g,2}}{\mu} + \frac{L_{g,2}L_{g,1}}{\mu^2} \right) \right) L_{g,1}}{\mu} + L_{f,1} + L_{f,0}\left( \frac{L_{g,2}L_{f,0}}{\mu} + \frac{L_{g,2}L^2_{f,0}}{\mu^2} \right).
$$

*Then, we have:*

$$\left\|\nabla_x \mathcal{F}^*(\bar{x}^k)\right\|^2 + \frac{8\underline{c}n}{5a^n}\mathcal{C}_{b,3}\lambda \mathbf{V}_D^k \leq \frac{4\mathcal{C}_{gap}}{\lambda^2} + \frac{16\big(\Phi^{(b)}(\hat{x}^k) - \Phi^{(b)}(\hat{x}^{k+1})\big)}{(ab)^n \eta_x^k}, \tag{85}$$

*where $\mathcal{C}_{gap}$ is a constant defined in (24). Furthermore, if the step sizes are chosen such that $\eta_x^k, \eta_y^k, \eta_z^k = \mathcal{O}(K^{-1/3})$ and the penalty parameter is set to $\lambda = \mathcal{O}(K^{1/3})$, then after $K$ iterations, the minimum squared norm of the hypergradient satisfies:*

$$\min_{0 \leq k < K-1} \|\nabla \mathcal{F}^*(\bar{x}^k)\|^2 = \mathcal{O}\left((ab)^{-n} K^{-\frac{2}{3}}\right), \tag{86}$$

*where $a, b \in (0,1)$ are defined in Assumption 3.2.*

*Proof.* Invoking the $L_{*,2}$-smoothness of the hyper-objective function $\mathcal{F}^*$, as established in Lemma 2.2 of Ghadimi & Wang (2018), we obtain:

$$\left\|\nabla_x \mathcal{F}^*(\hat{x}^k) - \nabla_x \mathcal{F}^*(\bar{x}^k)\right\|^2 \leq L_{*,2}^2 \|\hat{x}^k - \bar{x}^k\|^2 \leq L_{*,2}^2 \frac{1}{n}\sum_{i=1}^n \|x_i^k - \hat{x}^k\|^2$$

$$\leq \frac{L_{*,2}^2}{\underline{\alpha}} D(\mathbf{x}^k, \alpha_k). \tag{87}$$

Using the inequality $\|a + b\|^2 \leq 2\|a\|^2 + 2\|b\|^2$, we decompose the gradient norm of $\bar{x}^k$:

$$\left\|\nabla_x \mathcal{F}^*(\bar{x}^k)\right\|^2 \leq 2\left\|\nabla_x \mathcal{F}^*(\hat{x}^k)\right\|^2 + 2\left\|\nabla_x \mathcal{F}^*(\hat{x}^k) - \nabla_x \mathcal{F}^*(\bar{x}^k)\right\|^2$$

$$\overset{(87)}{\leq} 2\left\|\nabla_x \mathcal{F}^*(\hat{x}^k)\right\|^2 + \frac{2L_{*,2}^2}{\underline{\alpha}} D(\mathbf{x}^k, \alpha_k)$$

$$\overset{Lemma\ D.1}{\leq} 4\left\|\nabla_x \mathcal{L}_\lambda^*(\hat{x}^k)\right\|^2 + \frac{2L_{*,2}^2}{\underline{\alpha}} D(\mathbf{x}^k, \alpha_k) + \frac{4\mathcal{C}_{gap}}{\lambda^2}. \tag{88}$$

Next, by rearranging the terms in inequality (82), we obtain

$$\left\|\nabla_x \mathcal{L}_\lambda^*(\hat{x}^k)\right\|^2 \leq \frac{4\big(\Phi^{(b)}(\hat{x}^k) - \Phi^{(b)}(\hat{x}^{k+1})\big)}{\underline{\alpha}\underline{\beta}\eta_x^k} - \frac{4\underline{c}}{5}\mathcal{C}_{b,3}\lambda\frac{n}{\underline{\alpha}}\mathbf{V}_D^k,$$

where the last term utilizes the bound $-\frac{1}{\underline{\alpha}\underline{\beta}\eta_x^k} \leq -1$ (since all parameters are positive constants bounded by 1). Substituting this back into the bound for $\nabla \mathcal{F}^*(\bar{x}^k)$:

$$\left\|\nabla_x \mathcal{F}^*(\bar{x}^k)\right\|^2 \leq \frac{4\mathcal{C}_{gap}}{\lambda^2} + \frac{16\big(\Phi^{(b)}(\hat{x}^k) - \Phi^{(b)}(\hat{x}^{k+1})\big)}{K\underline{\alpha}\underline{\beta}\eta_x^k} + \left[\frac{2L_{*,2}^2}{\underline{\alpha}} D(\mathbf{x}^k, \alpha_k) - \frac{16\underline{c}}{5}\mathcal{C}_{b,3}\lambda\frac{n}{\underline{\alpha}}\mathbf{V}_D^k\right]. \tag{89}$$

If we choose $\lambda$ sufficiently large such that the condition

$$\frac{1}{\lambda} \leq \frac{4n\underline{c}\mathcal{C}_{b,3}}{5L_{*,2}^2}$$

holds, it implies that

$$\frac{2L_{*,2}^2}{\underline{\alpha}} \leq \frac{8\underline{c}}{5}\mathcal{C}_{b,3}\lambda\frac{n}{\underline{\alpha}}.$$

Noting that $\mathbf{V}_D^k \geq D(\mathbf{x}^k, \alpha_k)$ by definition, the last term in the inequality becomes non-positive and can be dropped, yielding (85).

Next, summing (85) over $k$, we have:

$$\frac{1}{K}\sum_{k=0}^{K-1}\left\|\nabla_x \mathcal{F}^*(\bar{x}^k)\right\|^2 \leq \frac{4\mathcal{C}_{gap}}{\lambda^2} + \frac{16\big(\Phi^{(b)}(\hat{x}^0) - \Phi^{(b)}(\hat{x}^K)\big)}{K\underline{\alpha}\underline{\beta}\eta_x^k}. \tag{90}$$

Finally, set a warm-start for $x, y, z$ and taking $\eta_x^k = \eta_y^k = \eta_z^k = \mathcal{O}\left(K^{-1/3}\right)$ and $\lambda = \mathcal{O}\left(K^{1/3}\right)$, both terms on the right-hand side scale as $\mathcal{O}(K^{-2/3})$. Specifically, $\lambda^{-2} \sim K^{-2/3}$ and $(K\eta)^{-1} \sim K^{-2/3}$. Therefore:

$$\min_{0 \le k < K-1} \left\| \nabla_x \mathcal{F}^*(\bar{x}^k) \right\|^2 \le \frac{1}{K} \sum_{k=0}^{K-1} \left\| \nabla_x \mathcal{F}^*(\bar{x}^k) \right\|^2 = \mathcal{O}\left( (ab)^{-n} K^{-\frac{2}{3}} \right).$$

$\square$

### D.8. Proof of Proposition 3.5

**Proposition D.14.** *Under the same conditions as Theorem 3.4, for any decision or auxiliary variable $\xi \in \{x, y, z\}$, the average consensus error satisfies:*

$$\min_{0 \le k < K} \frac{1}{n} \sum_{i=1}^n \|\xi_i^k - \bar{\xi}^k\|^2 = \mathcal{O}\left((ab)^{-n} K^{-1}\right). \tag{91}$$

*Proof.* Summing the descent inequality (85) over $k$, we have:

$$\frac{8\underline{c}}{5} \mathcal{C}_{b,3} \lambda \frac{n}{\underline{\alpha}} \frac{1}{K} \sum_{k=0}^{K-1} \mathbf{V}_D^k \le \frac{4\mathcal{C}_{gap}}{\lambda^2} + \frac{16\left(\Phi^{(b)}(\hat{x}^0) - \Phi^{(b)}(\hat{x}^K)\right)}{K \underline{\alpha} \underline{\beta} \eta_x^k}.$$

By the definition of the consensus measure $\mathbf{V}_D^k$, we recall the norm relationship $\frac{1}{n} \sum_{i=1}^n \|\xi_i^k - \bar{\xi}^k\|^2 \le \frac{n}{\underline{\alpha}} \mathbf{V}_D^k$ for any $\xi \in \{x, y, z\}$. Dividing both sides of the inequality by the coefficient $\frac{8}{5} \underline{c} \mathcal{C}_{b,3} \lambda$ and substituting the norm bound, we obtain:

$$\begin{aligned}
\min_{0 \le k < K} \frac{1}{n} \sum_{i=1}^n \|\xi_i^k - \bar{\xi}^k\|^2 &\le \frac{1}{K} \sum_{k=0}^{K-1} \frac{1}{n} \sum_{i=1}^n \|\xi_i^k - \bar{\xi}^k\|^2 \\
&\le \frac{1}{K} \sum_{k=0}^{K-1} \frac{n}{\underline{\alpha}} \mathbf{V}_D^k \\
&\le \frac{5}{8\underline{c}\mathcal{C}_{b,3}} \left( \frac{4\mathcal{C}_{gap}}{\lambda^3} + \frac{16\left(\Phi^{(b)}(\hat{x}^0) - \Phi^{(b)}(\hat{x}^K)\right)}{K \underline{\alpha} \underline{\beta} \lambda \eta_x^k} \right).
\end{aligned} \tag{92}$$

Finally, substituting the parameters $\eta_x^k = \eta_y^k = \eta_z^k = \mathcal{O}\left(K^{-1/3}\right)$ and $\lambda = \mathcal{O}\left(K^{1/3}\right)$ into the bound yields the desired convergence rate of $\mathcal{O}\left((ab)^{-n} K^{-1}\right)$, which completes the proof. $\square$

## E. Proof of Theorem 3.6

We consider the single-level optimization problem formulated as follows:

$$\min_{x \in \mathbb{R}^{d_x}} F(x) := \frac{1}{n} \sum_{i=1}^n f_i(x).$$

Let $\bar{x}^k := \frac{1}{n} \sum_{i=1}^n x_i^k$ denote the global average of the decision variables. Due to the row-stochastic property of the push matrices $\{B^k\}$ and the initialization $t_{x,i}^0 = \nabla f_i(x_i^0)$, the average of the tracking variables preserves the average gradient, i.e., $\bar{t}_x^k := \frac{1}{n} \sum_{i=1}^n t_{x,i}^k = \frac{1}{n} \sum_{i=1}^n \nabla f_i(x_i^k)$.

**Proof Sketch of Theorem 3.6** The proof strategy for the single-level Push-Pull algorithm parallels that of FAB but admits significant simplifications due to the absence of the lower-level constraint. We outline the three key steps below, highlighting the structural differences from the bilevel analysis.

*Step 1: Bounding the hypergradient $\nabla F(\bar{x}^k)$.* First, we relate the gradient of the objective at the averaged sequence $\bar{x}^k$ to the gradient at the auxiliary sequence $\hat{x}^k$. Invoking the smoothness properties of $F$, we obtain the following bound:

$$\left\| \nabla F(\bar{x}^k) \right\|^2 \le 2 \left\| \nabla_x F(\hat{x}^k) \right\|^2 + \frac{2L_{f,1}^2}{\underline{\alpha}} D(\mathbf{x}^k, \alpha_k), \tag{93}$$

where $D(\mathbf{x}^k, \alpha_k)$ represents the cumulative consensus error defined in (29), which arises from (109). Key difference: Unlike the bilevel formulation in FAB, this bound is not affected by a penalty parameter $\lambda$ or the approximation error of a lower-level solution. This absence of $\mathcal{O}(\lambda^{-2})$ terms allows for a faster fundamental convergence rate.

*Step 2: Constructing a proper Lyapunov function and establishing its descent property.* To control the terms in (93), we construct a Lyapunov function $\Phi^{(s)}$ tailored for the single-level objective:

$$\Phi^{(s)}(\hat{x}^k) := F(\hat{x}^k) + \mathcal{C}_{s,1} D(\mathbf{x}^k, \alpha_k) + \mathcal{C}_{s,2} S(\mathbf{t}_x^k, \beta_k), \tag{94}$$

where $\mathcal{C}_{s,\cdot}$ denote time-invariant constants defined in (101). Key difference: Crucially, these coupling coefficients do not depend on the step size $\eta$ or a penalty parameter, unlike the complex coupling required in FAB (Eq. (77)). This reflects the simpler dynamics where consensus and optimization can be balanced with constant weights.

Under the step size condition provided in Theorem 3.6, we establish the descent property:

$$\Phi^{(s)}(\hat{x}^{k+1}) - \Phi^{(s)}(\hat{x}^k) \leq -\frac{\eta_x^k}{4}\underline{\alpha}\underline{\beta} \left\| \nabla_x F(\hat{x}^k) \right\|^2 - \frac{\mathcal{C}_{s,1}\underline{c}n}{4\underline{\alpha}} D(\mathbf{x}^k, \alpha_k) \tag{95}$$

Similarly, we leverage the term $-\frac{\mathcal{C}_{s,1}\underline{c}}{4}\frac{n}{\underline{\alpha}}D(\mathbf{x}^k, \alpha_k)$ to control the $\mathcal{O}(1)\mathbf{V}_D^k$ term in (93).

*Step 3: Establishing the convergence rate.* Rearranging (95) and summing over $k = 0, \ldots, K - 1$, we derive:

$$\frac{1}{K} \sum_{k=0}^{K-1} \left\| \nabla_x F(\bar{x}^k) \right\|^2 \leq \frac{4}{\underline{\alpha}\underline{\beta}\eta_x^k} \left( \Phi^{(s)}(\hat{x}^0) - \Phi^{(s)}(\hat{x}^K) \right) - \sum_{k=0}^{K-1} \mathcal{C}_{s,1}\underline{c}\frac{n}{\underline{\alpha}}D(\mathbf{x}^k, \alpha_k).$$

Here the last term utilizes the bound $-\frac{1}{\underline{\alpha}\underline{\beta}\eta_x^k} \leq -1$ (since all parameters are positive constants bounded by 1) and we can control the step size $\eta_x^k$ to ensure that the term $\mathcal{C}_{s,1}\frac{4\underline{c}}{\eta_x^k}\frac{n}{\underline{\alpha}}D(\mathbf{x}^k, \alpha_k)$ is significantly larger than $\frac{2L_{f,1}^2}{\underline{\alpha}}D(\mathbf{x}^k, \alpha_k)$ in (33). Substituting this back into (33) and accounting for the error cancellation, we obtain the final convergence rate for the original objective:

$$\frac{1}{K} \sum_{k=0}^{K-1} \left\| \nabla_x F(\bar{x}^k) \right\|^2 \leq \frac{\mathcal{O}(1)\left(\Phi^{(s)}(\hat{x}^0) - \Phi^{(s)}(\hat{x}^K)\right)}{K\underline{\alpha}\underline{\beta}\eta_x^k}. \tag{96}$$

Since the step size $\eta_x^k$ can be chosen independently of the time horizon $K$ (e.g., a constant or a standard decay schedule), the right-hand side scales as $\mathcal{O}(K^{-1})$. Key difference: This yields a standard non-convex convergence rate of $\mathcal{O}(K^{-1})$, which is faster than the $\mathcal{O}(K^{-2/3})$ rate of FAB. The slower rate in FAB is the inevitable price paid for approximating the lower-level solution via the penalty method. This completes the proof sketch of Theorem 3.6.

Following the analysis framework established previously, we first derive the descent inequality for the objective function.

**Descent property of the objective function.**

**Lemma E.1.** *Suppose Assumptions 3.1, 3.2 holds. Given the $L_{f,1}$-smoothness of $F$, the consecutive iterates generated by Algorithm 2 satisfy:*

$$\begin{aligned}
F(\hat{x}^{k+1}) - F(\hat{x}^k) \leq\ & -\frac{1}{\sum_{i=1}^n [\alpha_{k+1}]_i n[\beta_k]_i}\frac{1}{2\eta_x^k}\|\hat{x}^{k+1} - \hat{x}^k\|^2 + \frac{L_{f,1}}{2}\|\hat{x}^{k+1} - \hat{x}^k\|^2 \\
& + \frac{1}{\sum_{i=1}^n [\alpha_{k+1}]_i n[\beta_k]_i}\frac{\eta_x^k}{2}\left(6S(\mathbf{x}^k, \beta_k) + 6L_{f,1}^2\frac{n}{\min(\alpha_k)}D(\mathbf{x}^k, \alpha_k)\right) \\
& - \frac{\eta_x^k}{2}\sum_{i=1}^n [\alpha_{k+1}]_i n[\beta_k]_i \left\| \nabla_x F(\hat{x}^k) \right\|^2,
\end{aligned}$$

*where $\hat{x}^k := \sum_{i=1}^n [\alpha_k]_i x_i^k$ denotes the $\alpha_k$-weighted average. The weight vectors $\alpha_k$ and $\beta_k$ are specified in Lemmas D.4 and D.5, respectively. Here, the terms $D(\cdot)$ and $S(\cdot)$ are defined in (29).*

*Proof.* The proof follows directly from the proof of Lemma D.6. □

**Analysis of Weighted Dispersion.**

**Lemma E.2.** *Let Assumptions 3.1 and 3.2 hold, and assume that each function $f_i$ is $L_{f,1}$-smooth. For the sequences generated by Algorithm 2, the following inequality holds for all $k \geq 0$:*

$$D(\mathbf{x}^{k+1}, \alpha_{k+1}) \leq (1 - c_k)D(\mathbf{x}^k, \alpha_k)$$
$$+ \frac{2}{c_k}(\eta_x^k)^2 \max_{j \in [n]}([\alpha_{k+1}]_j[\beta_k]_j)\left(S(\mathbf{t}_x^k, \beta_k) + \left\|\sum_{\ell=1}^n \nabla f(x_\ell^k)\right\|^2\right), \tag{97}$$

*where $D(\mathbf{x}^k, \alpha_k)$ and $S(\mathbf{t}_x^k, \beta_k)$ are defined in (29), the stacked variables $\mathbf{x}^k$ follow (26). The weight vectors $\alpha_k$ and $\beta_k$ are specified in Lemmas D.4 and D.5, respectively. Furthermore, the contraction parameter $c_k$ is defined as*

$$c_k := \frac{\min(\alpha_{k+1})a^2}{\max^2(\alpha_k)\mathrm{D}(\mathcal{G}_k)\mathrm{K}(\mathcal{G}_k)}, \quad 0 < c_k < 1,$$

*with $\mathrm{D}(\mathcal{G}^k)$ and $\mathrm{K}(\mathcal{G}^k)$ given in Definitions D.2 and D.3.*

*Proof.* The proof follows directly from the proof of Lemma D.8. $\qquad\square$

**Lemma E.3.** *Let Assumptions 3.1 and 3.2 hold, and assume that each function $f_i$ is $L_{f,1}$-smooth. For all $k \geq 0$, the tracking error satisfies:*

$$S(\mathbf{t}_x^{k+1}, \beta_{k+1}) \leq (1 - e_k)S(\mathbf{t}_x^k, \beta_k) + \frac{4\gamma L_{f,1}^2}{e_k}\|\mathbf{x}^{k+1} - \mathbf{x}^k\|^2, \tag{98}$$

*where $S(\mathbf{t}_x^k, \beta_k)$ is defined in (29), $\beta_k$ is given in Lemma D.5, and the stacked variables $\mathbf{x}^k$ are defined in (26). Additionally, $\gamma := \max_k(n + \frac{1}{\min(\beta_{k+1})})$, and the coefficient $e_k \in (0, 1)$ is defined as*

$$e_k = \frac{\min^2(\beta_k)b^2}{\max^2(\beta_k)\max(\beta_{k+1})\mathrm{D}(\mathcal{G}^k)\mathrm{K}(\mathcal{G}^k)},$$

*where $\mathrm{D}(\mathcal{G}^k)$ and $\mathrm{K}(\mathcal{G}^k)$ are given in Definitions D.2 and D.3, respectively.*

*Proof.* The proof follows directly from the proof of Lemma D.9. $\qquad\square$

**Lemma E.4.** *Let Assumptions 3.1 and 3.2 hold, and assume that each function $f_i$ is $L_{f,1}$-smooth. For any variable $x$ and $\mathbf{x}$ for all $k \geq 0$, we have:*

$$\|\mathbf{x}^{k+1} - \mathbf{x}^k\|^2 \leq 3\left(\frac{c_k}{\min(\alpha_{k+1})} + \frac{1}{\min(\alpha_k)}\right)D(\mathbf{x}^k, \alpha_k)$$
$$+ 3(\eta_x^k)^2\left(S(\mathbf{t}_x^k, \beta_k) + \left\|\sum_{i=1}^n \nabla f(x_i^k)\right\|^2\right). \tag{99}$$

*Here, $\mathbf{x}^k$ is defined in (26). The scalars $\alpha_k$ and $c_k$ are given in Lemmas D.4 and D.8, respectively, while $D(\mathbf{x}^k, \alpha_k)$ and $S(\mathbf{t}_x^k, \beta_k)$ are defined in (29) and (30).*

*Proof.* The proof follows directly from the proof of Lemma D.10. $\qquad\square$

**Lemma E.5.** *Let Assumptions 3.1 and 3.2 hold, and assume that each function $f_i$ is $L_{f,1}$-smooth. Then the squared norms of the sum of tracking variables are bounded as follows:*

$$\left\|\nabla f(x_i^k)\right\|^2 \leq 12L_{f,1}^2\frac{n}{\min(\alpha_k)}D(\mathbf{x}^k, \alpha_k) + 4n^2\left\|\nabla_x F(\hat{x}^k)\right\|^2, \tag{100}$$

*where $D(\mathbf{x}^k, \alpha_k)$ denotes the consensus error as defined in (29), and $\alpha_k$ is given in Lemma D.4.*

*Proof.* The proof follows directly from the proof of Lemma D.11. $\qquad\square$

**Descent in the Lyapunov function.** We define the Lyapunov function for the single-level objective $F$ as:

$$\Phi^{(s)}(\hat{x}^k) := F(\hat{x}^k) + \mathcal{C}_{s,1} D(\mathbf{x}^k, \alpha_k) + \mathcal{C}_{s,2} S(\mathbf{t}_x^k, \beta_k), \tag{101}$$

where thethe time-invariant base constants are given by:

$$\mathcal{C}_{s,1} := \frac{\beta \underline{c}}{32 \overline{\alpha} \overline{\beta} n^3}, \quad \mathcal{C}_{s,2} := 1. \tag{102}$$

Based on the established error bounds, we derive the following descent lemma.

**Lemma E.6.** *Suppose that Assumptions 3.1 and 3.2 hold, and each function $f_i$ is $L_{f,1}$-smooth. If the step-size $\eta_x^k$ is chosen such that $\eta_x^k \geq \underline{\eta}$ (with $\underline{\eta} > 0$) and satisfies the following upper bound condition:*

$$\eta_x^k \leq \min\left\{ \frac{1}{\overline{\alpha}\overline{\beta} L_{f,1}}, \quad \frac{\mathcal{C}_{s,1} \underline{c} e n}{384 \gamma L_{f,1}^2}, \quad \sqrt{\frac{\mathcal{C}_{s,1} \underline{c} \underline{\alpha} \underline{\beta}}{48 L_{f,1}^2}}, \right.$$
$$\sqrt{\frac{\mathcal{C}_{s,1} \underline{c}}{192 L_{f,1}^2 \left( \mathcal{C}_{s,1} \frac{2}{\underline{c}} \overline{\beta} + \frac{12 \gamma L_{f,1}^2}{e} \right)}}, \quad \frac{e \underline{\alpha} \underline{\beta}}{12}, \tag{103}$$
$$\left. \frac{\underline{\alpha} \underline{\beta}}{16 n^2 \left( \mathcal{C}_{s,1} \frac{2}{\underline{c}} n \underline{\beta} + \frac{12 \gamma L_{f,1}^2}{e} \right)}, \quad \frac{e \underline{\alpha} \underline{\beta}}{384 \gamma L_{f,1}^2 n^2} \right\},$$

*where the sets of constants $\{\underline{\alpha}, \overline{\alpha}, \underline{\beta}, \overline{\beta}\}$, $\{\underline{q}, \underline{c}, \underline{e}\}$, and $\{\mathcal{C}_{b,1}, \mathcal{C}_{b,2}, \mathcal{C}_{b,3}\}$ are defined in (74), (75), and (102), respectively. Then the Lyapunov function satisfies the following descent inequality:*

$$\Phi^{(s)}(\hat{x}^{k+1}) - \Phi^{(s)}(\hat{x}^k) \leq -\frac{\eta_x^k}{4} \underline{\alpha} \underline{\beta} \left\| \nabla_x F(\hat{x}^k) \right\|^2 - \frac{\mathcal{C}_{s,1} \underline{c} n}{4 \underline{\alpha}} D(\mathbf{x}^k, \alpha_k). \tag{104}$$

*Proof.* Substituting the contraction properties of the consensus and tracking errors, and utilizing the bounds on the gradient difference $\| \sum \nabla f(x_\ell^k) \|^2$, we expand the difference of the Lyapunov function as follows:

$$\Phi^{(s)}(\hat{x}^{k+1}) - \Phi^{(s)}(\hat{x}^k)$$
$$\leq -\frac{1}{2\eta_x^k \overline{\alpha}\overline{\beta}} \|\hat{x}^{k+1} - \hat{x}^k\|^2 + \frac{L_{f,1}}{2} \|\hat{x}^{k+1} - \hat{x}^k\|^2$$
$$+ \frac{\eta_x^k}{2\underline{\alpha}\underline{\beta}} \left( 6 S(\mathbf{t}_x^k, \beta_k) + 6 L_{f,1}^2 \frac{n}{\underline{\alpha}} D(\mathbf{x}^k, \alpha_k) \right) - \frac{\eta_x^k}{2} \underline{\alpha} \underline{\beta} \left\| \nabla_x F(\hat{x}^k) \right\|^2$$
$$- \mathcal{C}_{s,1} \underline{c} \frac{n}{\underline{\alpha}} D(\mathbf{x}^k, \alpha_k) + \mathcal{C}_{s,1} \frac{2}{\underline{c}} (\eta_x^k)^2 \overline{\alpha} \overline{\beta} \frac{n}{\underline{\alpha}} S(\mathbf{t}_x^k, \beta_k)$$
$$- \mathcal{C}_{s,2} \underline{e} S(\mathbf{t}_x^k, \beta_k) + \mathcal{C}_{s,2} \frac{24 \gamma L_{f,1}^2}{e \underline{\alpha}} D(\mathbf{x}^k, \alpha_k) + \mathcal{C}_{s,2} \frac{12 \gamma L_{f,1}^2}{e} (\eta_x^k)^2 S(\mathbf{t}_x^k, \beta_k)$$
$$+ \left( \mathcal{C}_{s,1} \frac{n}{\underline{\alpha}} \frac{2}{\underline{c}} \overline{\alpha} \overline{\beta} + \mathcal{C}_{s,2} \frac{12 \gamma L_{f,1}^2}{e} \right) (\eta_x^k)^2 \left( 12 L_{f,1}^2 \frac{n}{\underline{\alpha}} D(\mathbf{x}^k, \alpha_k) + 4 n^2 \left\| \nabla_x F(\hat{x}^k) \right\|^2 \right).$$

We now group the terms by the error quantities $\|\hat{x}^{k+1} - \hat{x}^k\|^2$, $D(\mathbf{x}^k, \alpha_k)$, $S(\mathbf{t}_x^k, \beta_k)$, and the gradient norm $\|\nabla_x F(\hat{x}^k)\|^2$.

1. Term $\|\hat{x}^{k+1} - \hat{x}^k\|^2$: The coefficient is

$$C_{\Phi,1}^{(s)} := -\frac{1}{2\eta_x^k \overline{\alpha}\overline{\beta}} + \frac{L_{f,1}}{2}.$$

2. Term $D(\mathbf{x}^k, \alpha_k)$: The coefficient is

$$C_{\Phi,2}^{(s)} := \frac{3 L_{f,1}^2 n}{\underline{\beta} \underline{\alpha}^2} \eta_x^k - \mathcal{C}_{s,1} \underline{c} \frac{n}{\underline{\alpha}} + \mathcal{C}_{s,2} \frac{24 \gamma L_{f,1}^2}{e \underline{\alpha}} + \left( \mathcal{C}_{s,1} \frac{2}{\underline{c}} \overline{\alpha} \overline{\beta} \frac{n}{\underline{\alpha}} + \mathcal{C}_{s,2} \frac{12 \gamma L_{f,1}^2}{e} \right) \frac{12 n L_{f,1}^2}{\underline{\alpha}} (\eta_x^k)^2.$$

3. Term $S(\mathbf{t}_x^k, \beta_k)$: The coefficient is

$$C_{\Phi,3}^{(s)} := \frac{3}{\underline{\alpha\beta}}\eta_x^k - \mathcal{C}_{s,2}\underline{e} + \mathcal{C}_{s,1}\frac{2}{\underline{c}}(\eta_x^k)^2\overline{\alpha}\overline{\beta}\frac{n}{\underline{\alpha}} + \mathcal{C}_{s,2}\frac{12\gamma L_{f,1}^2}{\underline{e}}(\eta_x^k)^2.$$

4. Term $\|\nabla_x F(\hat{x}^k)\|^2$: The coefficient is

$$C_{\Phi,4}^{(s)} := -\frac{\alpha\beta}{2}\eta_x^k + \left(\mathcal{C}_{s,1}\frac{2}{\underline{c}}\overline{\alpha}\overline{\beta}\frac{n}{\underline{\alpha}} + \mathcal{C}_{s,2}\frac{12\gamma L_{f,1}^2}{\underline{e}}\right)4n^2(\eta_x^k)^2.$$

Under these conditions in (103), the coefficients simplify to the desired bounds, yielding:

$$\Phi^{(s)}(\hat{x}^{k+1}) - \Phi^{(s)}(\hat{x}^k) \le -\frac{\eta_x^k}{4}\underline{\alpha\beta}\left\|\nabla_x F(\hat{x}^k)\right\|^2 - \frac{\mathcal{C}_{s,1}\underline{c}}{4}\frac{n}{\underline{\alpha}}D(\mathbf{x}^k, \alpha_k). \tag{105}$$

$\square$

**Convergence rate of Push-Pull**

**Theorem E.7.** *Suppose that Assumptions 3.1 and 3.2 hold, and that each function $f_i$ is $L_{f,1}$-smooth. Provided that the step sizes satisfy the conditions in Lemma E.6 and are bounded by $\eta_x^k \le \mathcal{C}_{s,1}\frac{n\underline{c}}{4L_{f,1}^2}$, where the constants $\mathcal{C}_{s,1}$ and $\underline{c}$ are defined in (102) and (75) respectively, then the algorithm achieves the following convergence bounds for the global gradient norm and the average consensus error:*

$$\min_{0\le k<K}\|\nabla F(\bar{x}^k)\|^2 = \mathcal{O}\left((ab)^{-n}K^{-1}\right), \tag{106a}$$

$$\min_{0\le k<K}\frac{1}{n}\sum_{i=1}^{n}\|x_i^k - \bar{x}^k\|^2 = \mathcal{O}\left((ab)^{-n}K^{-1}\right), \tag{106b}$$

*where $F(x) = \frac{1}{n}\sum_{i=1}^{n}f_i(x)$.*

*Proof.* From Lemma E.6, we proceed as follows.

First, summing the descent inequality in (105) over $k$ from 0 to $K-1$, we obtain:

$$\sum_{k=0}^{K-1}\left\|\nabla_x F(\hat{x}^k)\right\|^2 \le \frac{4}{\underline{\alpha\beta}\eta_x}\left(\Phi^{(s)}(\hat{x}^0) - \Phi^{(s)}(\hat{x}^{K-1})\right) - \sum_{k=0}^{K-1}\mathcal{C}_{s,1}\underline{c}\frac{n}{\underline{\alpha}}D(\mathbf{x}^k, \alpha_k), \tag{107}$$

where the last term utilizes the bound $-\frac{1}{\underline{\alpha\beta}\eta_x^k} \le -1$ (since all parameters are positive constants bounded by 1). Invoking the $L_{f,1}$-smoothness of $F$, we can bound the gradient difference terms:

$$\begin{aligned}
\left\|\nabla_x F(\hat{x}^k) - \nabla_x F(\bar{x}^k)\right\|^2 &\le L_{f,1}^2\|\hat{x}^k - \bar{x}^k\|^2 \\
&\le L_{f,1}^2\frac{1}{n}\sum_{i=1}^{n}\|x_i^k - \hat{x}^k\|^2 \\
&\le \frac{L_{f,1}^2}{\underline{\alpha}}D(\mathbf{x}^k, \alpha_k). \tag{108}
\end{aligned}$$

Applying the inequality $\|a+b\|^2 \le 2\|a\|^2 + 2\|b\|^2$, we obtain:

$$\begin{aligned}
\left\|\nabla_x F(\bar{x}^k)\right\|^2 &\le 2\left\|\nabla_x F(\hat{x}^k)\right\|^2 + 2\left\|\nabla_x F(\hat{x}^k) - \nabla_x F(\bar{x}^k)\right\|^2 \\
&\le 2\left\|\nabla_x F(\hat{x}^k)\right\|^2 + \frac{2L_{f,1}^2}{\underline{\alpha}}D(\mathbf{x}^k, \alpha_k). \tag{109}
\end{aligned}$$

Summing over $k = 0, \ldots, K - 1$ and dividing by $K$ yields:

$$\frac{1}{K}\sum_{k=0}^{K-1}\left\|\nabla_x F(\bar{x}^k)\right\|^2 \leq \frac{2}{K}\sum_{k=0}^{K-1}\left\|\nabla_x F(\hat{x}^k)\right\|^2 + \frac{2L_{f,1}^2}{\underline{\alpha} K}\sum_{k=0}^{K-1}D(\mathbf{x}^k, \alpha_k).$$

Next, substituting (107) into the inequality above, we have:

$$\frac{1}{K}\sum_{k=0}^{K-1}\left\|\nabla_x F(\bar{x}^k)\right\|^2 \leq \frac{8\big(\Phi^{(s)}(\hat{x}^0) - \Phi^{(s)}(\hat{x}^K)\big)}{K\underline{\alpha}\underline{\beta}\eta_x^k}$$
$$+ \frac{1}{K}\sum_{k=0}^{K-1}\left[\frac{2L_{f,1}^2}{\underline{\alpha}}D(\mathbf{x}^k, \alpha_k) - 2\mathcal{C}_{s,1}\underline{c}\frac{n}{\underline{\alpha}}D(\mathbf{x}^k, \alpha_k)\right].$$

If we take $\eta_x^k \leq \mathcal{C}_{s,1}\frac{n\underline{c}}{4L_{f,1}^2}$, we can obtain that:

$$\frac{1}{K}\sum_{k=0}^{K-1}\left\|\nabla_x F(\bar{x}^k)\right\|^2 \leq \frac{8\big(\Phi^{(s)}(\hat{x}^0) - \Phi^{(s)}(\hat{x}^K)\big)}{K\underline{\alpha}\underline{\beta}\eta_x^k}.$$

Considering that $\eta_x^k \geq \underline{\eta}$, we have

$$\min_{0 \leq k < K}\left\|\nabla_x F(\bar{x}^k)\right\|^2 = \mathcal{O}\left((\underline{\alpha}\underline{\beta})^{-1}K^{-1}\right). \tag{110}$$

The analysis of the consensus error $\frac{1}{n}\sum_{i=1}^n\|x_i^k - \bar{x}^k\|^2$ follows a similar line of reasoning to Proposition D.14. $\qquad\square$

---

**Algorithm 3** PushSum_FAB Algorithm

---

1: **Input:** Initial variables $\{x_i^0, y_i^0, z_i^0\}_{i=1}^n$; Initial weights $\{w_{x,i}^0, w_{y,i}^0, w_{z,i}^0\}_{i=1}^n \leftarrow 1$; Initial variables $\{(t_{x,i}^0, t_{y,i}^0, t_{z,i}^0) = (d_{x,i}^0, d_{y,i}^0, d_{z,i}^0)\}_{i=1}^n$, computed via (7); Sequences of column-stochastic matrices $\{B^k\}$; Learning rates $\{(\eta_x^k, \eta_y^k, \eta_z^k)\}$; Penalty parameter $\lambda$; Total number of iterations $K$.

2: **for** $k = 0, 1, \ldots, K - 1$ **do**

3:     **for** each agents $i \in \{1, \ldots, n\}$ in parallel **do**

4:         **Decision variable update:**

5:             $x_i^{k+1} = \sum_{j=1}^n [B^k]_{ij}(x_j^k - \eta_x^k t_{x,j}^k)$

6:             $y_i^{k+1} = \sum_{j=1}^n [B^k]_{ij}(y_j^k - \eta_y^k t_{y,j}^k)$

7:             $z_i^{k+1} = \sum_{j=1}^n [B^k]_{ij}(z_j^k - \eta_z^k t_{z,j}^k)$

8:         Update weights:

9:             $w_{\xi,i}^{k+1} = \sum_{j=1}^n [B^k]_{ij}w_{\xi,j}^k, \quad \forall \xi \in \{x, y, z\}$

10:         **Bias correction (ratio retrieval):**

11:             $\tilde{x}_i^{k+1} = x_i^{k+1}/w_{x,i}^{k+1}$

12:             $\tilde{y}_i^{k+1} = y_i^{k+1}/w_{y,i}^{k+1}$

13:             $\tilde{z}_i^{k+1} = z_i^{k+1}/w_{z,i}^{k+1}$

14:         **Local gradient evaluation:**

15:             $d_{x,i}^{k+1} = \nabla_x f_i(\tilde{x}_i^{k+1}, \tilde{y}_i^{k+1}) + \lambda\big(\nabla_x g(\tilde{x}_i^{k+1}, \tilde{y}_i^{k+1}) - \nabla_x g(\tilde{x}_i^{k+1}, \tilde{z}_i^{k+1})\big)$

16:             $d_{y,i}^{k+1} = \nabla_y f_i(\tilde{x}_i^{k+1}, \tilde{y}_i^{k+1}) + \lambda\nabla_y g_i(\tilde{x}_i^{k+1}, \tilde{y}_i^{k+1})$

17:             $d_{z,i}^{k+1} = \lambda\nabla_y g_i(\tilde{x}_i^{k+1}, \tilde{z}_i^{k+1})$

18:         **Tracking variable update:**

19:             $t_{\xi,i}^{k+1} = \sum_{j=1}^n [B^k]_{ij}t_{\xi,j}^k + (d_{\xi,i}^{k+1} - d_{\xi,i}^k), \quad \forall \xi \in \{x, y, z\}$

20:     **end for**

21: **end for**

---

---

**Algorithm 4** PushPull_SOBA Algorithm

---

1: **Input:** Initial variables $\{x_i^0, y_i^0, v_i^0\}_{i=1}^n$; Initial variables $\{(t_{x,i}^0, t_{y,i}^0, t_{v,i}^0) = (d_{x,i}^0, d_{y,i}^0, d_{v,i}^0)\}_{i=1}^n$, computed via (111); Sequences of row-stochastic matrices $\{A^k\}$ and column-stochastic matrices $\{B^k\}$; Learning rates $\{(\eta_x^k, \eta_y^k, \eta_z^k)\}$; Total number of iterations $K$.

2: **for** $k = 0, 1, \ldots, K-1$ **do**

3:     **for** each agents $i \in \{1, \ldots, n\}$ in parallel **do**

4:         **Decision variable update:**

5:             $x_i^{k+1} = \sum_{j=1}^n [A^k]_{ij} x_j^k - \eta_x^k t_{x,i}^k$

6:             $y_i^{k+1} = \sum_{j=1}^n [A^k]_{ij} y_j^k - \eta_y^k t_{y,i}^k$

7:             $v_i^{k+1} = \sum_{j=1}^n [A^k]_{ij} v_j^k - \eta_v^k t_{v,i}^k$

8:         **Local gradient evaluation:**

$$d_{y,i}^{k+1} = \nabla_y G_i(x_i^{k+1}, y_i^{k+1}) \quad \text{(Compute lower-level gradient)} \tag{111a}$$

$$d_{v,i}^{k+1} = \nabla_{yy}^2 G_i(x_i^{k+1}, y_i^{k+1}) v_i^{k+1} - \nabla_y F_i(x_i^{k+1}, y_i^{k+1}) \quad \text{(Compute auxiliary direction)} \tag{111b}$$

$$d_{x,i}^{k+1} = \nabla_x F_i(x_i^{k+1}, y_i^{k+1}) - \nabla_{xy}^2 G_i(x_i^{k+1}, y_i^{k+1}) v_i^{k+1} \quad \text{(Compute hypergradient)} \tag{111c}$$

9:         **Tracking variable update:**

10:          $t_{x,i}^{k+1} = \sum_{j=1}^n [B^k]_{ij} t_{x,j}^k + (d_{x,i}^{k+1} - d_{x,i}^k)$

11:          $t_{y,i}^{k+1} = \sum_{j=1}^n [B^k]_{ij} t_{y,j}^k + (d_{y,i}^{k+1} - d_{y,i}^k)$

12:          $t_{v,i}^{k+1} = \sum_{j=1}^n [B^k]_{ij} t_{v,j}^k + (d_{v,i}^{k+1} - d_{v,i}^k)$

13:     **end for**

14: **end for**

---

