# OpenReview forum: "FAB: A First-Order AB-based Gradient Algorithm for Distributed Bilevel Optimization over Time-Varying Directed Graphs"
_ICML.cc/2026/Conference — ICML 2026 regular_

### Official Review · Reviewer_e1j8 · 2026-03-10

**Soundness:** 4
**Presentation:** 4
**Significance:** 3
**Originality:** 4
**Overall Recommendation:** 6
**Confidence:** 4

**Summary:**

The paper proposes FAB, a first-order algorithm that integrates the Push-Pull communication strategy with a value-function penalty method to solve distributed bilevel optimization problems over time-varying directed graphs. Theoretically, it establishes an $\mathcal{O}(K^{-2/3})$ convergence rate for the nonconvex-strongly-convex setting. Additionally, the authors resolve an open question by proving an $\mathcal{O}((ab)^{-n}K^{-1})$ convergence rate for the standard single-level Push-Pull algorithm in nonconvex scenarios. Empirical evaluations on tasks like distributed RL and data hyper-cleaning demonstrate FAB's superior efficiency and robustness against existing double-loop baselines.

**Compliance With Llm Reviewing Policy:**

Affirmed.

**Final Justification:**

The authors solved my concerns.

**Key Questions For Authors:**

1. Could the authors briefly clarify the core advantage of adopting the value-function penalty approach over first-order iterative differentiation (ITD) specifically under the stringent communication constraints of time-varying directed topologies?


2. Could the authors provide a comparison showing test accuracy against the total cumulative communication volume (e.g., in MB)? This would perfectly demonstrate how the structural advantage of the proposed single-loop method inherently overcomes its 2x per-iteration communication overhead in practical distributed systems in Figure 7.


3. While the theoretical convergence rate includes an $(ab)^{-n}$ factor (suggesting poor scalability with network size $n$), the empirical results demonstrate a much more graceful, non-exponential degradation. Could the authors briefly share the intuition on why the practical performance significantly exceeds this worst-case theoretical bound?

**Limitations:**

Yes

**Strengths And Weaknesses:**

**Strengths**

Algorithmic Innovation: This work proposes FAB, the first fully first-order algorithm to address this specific problem. By elegantly integrating the Push-Pull (AB) communication strategy with a value function-based penalty method, FAB efficiently mitigates consensus bias in dynamic unbalanced networks without requiring complex double-loop structures.

Theoretical Contributions: The paper establishes a rigorous $\mathcal{O}(K^{-2/3})$ convergence rate for distributed bilevel optimization over time-varying directed graphs. As a valuable byproduct, it also resolves an open question by proving an $\mathcal{O}((ab)^{-n}K^{-1})$ convergence rate for the standard single-level Push-Pull algorithm in nonconvex settings.

Empirical Validation: The experiments are extensive and convincing. The authors evaluate FAB across diverse and large-scale tasks, including distributed hyperparameter tuning, distributed RL policy evaluation, and data hyper-cleaning tasks.

Presentation: The paper is well-written and easy to read.

**Weaknesses:**

The theoretical guarantees are established under the lower-level strong convexity assumption. However, the empirical results demonstrate FAB's effectiveness far beyond this scope, for instance, on highly non-convex tasks such as BERT fine-tuning. The paper would be even more comprehensive if the authors could add a brief remark acknowledging this favorable gap between the strict theoretical assumptions and the algorithm's broader empirical capabilities.

---

> ### Author Rebuttal · Authors · 2026-03-30
>
> We sincerely thank the reviewer for their positive remarks regarding FAB’s algorithmic design, our theoretical analysis in resolving the single-level open question, and our empirical evaluations. Below is our detailed response to the constructive feedback.
>
> ### **Response to W:**
>
> We thank the reviewer for highlighting the gap between our theory and FAB’s broader empirical capability. We will add a remark in the revision to clarify this point:
>
> - **Theoretical assumptions and challenges:** Strong convexity of the lower-level problem is standard in distributed bilevel optimization, as it ensures a well-defined hypergradient and the smoothness of the hyper-objective $\mathcal{F}^{*}(x)$. For general nonconvex lower-level problems, tools such as the Moreau envelope may be used, but the hypergradient is no longer well-defined and there is still no widely accepted stationarity measure for convergence analysis. Establishing rigorous non-asymptotic rates in this setting remains a major open problem because of the lack of an appropriate stationarity notion, the inherent nonsmoothness, and the possible nonuniqueness of lower-level solutions.
>
> - **Practical implementation:** Although our theory relies on lower-level strong convexity, FAB itself does not require it in practice. By using a value-function-based penalty reformulation, FAB operates in a single-loop gradient descent–ascent manner and avoids the fragile exactness required by iterative differentiation. It can also be naturally combined with AB/Push–Pull communication, making it more adaptable to general nonconvex tasks such as BERT fine-tuning.
>
>
> ### **Response to Q1:**
>
> We thank the reviewer for this important question. The main advantage is that FAB avoids the severe **synchronization and sequential communication bottlenecks** faced by ITD-based methods under time-varying directed topologies:
>
> - **Synchronization bottleneck in ITD-based methods:** ITD methods (e.g., unrolling) compute hypergradients by chaining through $T$ inner iterations, which requires sequential Hessian–vector tracking variables. Over time-varying directed graphs, where links are asymmetric and changing, reliably propagating these variables is slow and fragile. The communication cost also scales multiplicatively with $T$, creating a major synchronization bottleneck.
>
> - **Structural advantage of the value-function penalty:** By reformulating the bilevel problem into a single-level penalized objective via the lower-level value function, FAB turns the problem into decentralized min–max optimization. FAB then uses only concurrent first-order gradients within a single simultaneous Push–Pull loop, with no nested tracking variables and no multi-round sequential synchronization.
>
>
> ### **Response to Q2:**
>
> We appreciate this suggestion. Please see the *Comparison with Single-Level ITD Baselines* in our **Response to Question 1 for Reviewer jZHp**, where FAB achieves the highest communication efficiency among all methods in Figure 7.
>
> ### **Response to Q3:**
>
> We thank the reviewer for this insightful observation. The empirical scaling is much better than the worst-case bound $(ab)^{-n}$. We will clarify this in the revision as follows:
>
> - **Theoretical worst case:** In convergence analysis over time-varying directed graphs, one must cover adversarial topology sequences. As shown in [1], the factor $(ab)^{-n}$ comes from a pathological “adversarial token-passing” process, where information moves through only one shifting bottleneck link with minimal weights $a$ and $b$ at each step (e.g., Node 1 $\rightarrow$ Node 2 at step $k$, then Node 2 $\rightarrow$ Node 3 at step $k+1$). In such a case, information needs $n$ steps to traverse the network, and the consensus signal decays by factors $a$ and $b$ at each step, yielding exponential mixing time.
>
> - **Practical connectivity:** In practical decentralized learning, including the Erdős-Rényi graphs in our experiments, time-varying topologies usually have much stronger algebraic connectivity. Nodes communicate with multiple peers rather than along a single shifting chain, and these “random shortcuts” reduce the effective dynamic diameter, allowing mixing in $\mathcal{O}(\log n)$ steps [2] instead of suffering the worst-case exponential penalty. Hence the consensus error does not grow exponentially in practice, which explains the graceful degradation in Fig. 6(a). Although one can artificially build topology sequences that realize the $(ab)^{-n}$ bottleneck, such cases are highly pathological and far from practical dynamic learning systems.
>
> **References**:
>
> [1] A. Olshevsky and J. N. Tsitsiklis. Convergence speed in distributed consensus and averaging. SIAM Review, 2011.
>
> [2] D. J. Watts and S. H. Strogatz. Collective dynamics of ‘small-world’ networks. Nature, 1998.

---

> > ### Author Rebuttal · Reviewer_e1j8 · 2026-04-02
> >
> > I have read all rebuttals. My concerns have been solved.

---

> > > ### Author Response · Authors · 2026-04-04
> > >
> > > Thanks for your timely response. We are glad that your concerns have been addressed.

---

### Official Review · Reviewer_5coN · 2026-03-12

**Soundness:** 3
**Presentation:** 3
**Significance:** 3
**Originality:** 3
**Overall Recommendation:** 5
**Confidence:** 4

**Summary:**

This paper presents the FAB algorithm to address distributed bilevel optimization over time-varying directed graphs. The work is theoretically solid, well-motivated, and demonstrates strong performance across multiple representative deep learning tasks. While the current experiments cover several critical aspects, adding a discussion on the algorithm's robustness against node failures or communication disruptions, along with further insights into large-scale network scalability, would significantly enhance the depth and practical impact of the study.

**Compliance With Llm Reviewing Policy:**

Affirmed.

**Key Questions For Authors:**

1. In the experimental section, most deep learning tasks (e.g., BERT fine-tuning) appear to be limited to smaller networks (e.g., $n=10$). Could you provide additional results or a discussion on the convergence behavior and communication overhead when scaling to significantly larger networks (e.g., $n=1000$)? Understanding this behavior is crucial for validating the algorithm’s practical deployment in large-scale distributed systems.

2. While the paper considers time-varying directed topologies, how does the FAB algorithm perform under more extreme conditions, such as random link failures, node dropouts, or potential Byzantine attacks? If you have not tested these, a brief discussion on whether the current mechanism inherently handles such volatility—or how it might be adapted—would significantly clarify its robustness.

**Limitations:**

Yes

**Strengths And Weaknesses:**

Strengths:

1. Comprehensive content and theoretical foundation: The paper provides a rigorous theoretical analysis of the proposed "FAB" algorithm, a first-order gradient method based on the Push-Pull (AB) communication strategy, for distributed bilevel optimization over time-varying directed graphs.

2. Clear motivation: The authors effectively justify the necessity of integrating bilevel optimization into distributed learning to tackle hyperparameter tuning, supported by compelling empirical observations.

3. Extensive validation: The proposed algorithm is validated through a wide range of deep learning applications, including hyperparameter tuning, data hyper-cleaning, and reinforcement learning, covering both CV and NLP domains.

Weaknesses

1. Large-scale networks and scalability: While the reinforcement learning tasks explore up to $n=1000$ agents, other complex deep learning tasks (e.g., BERT fine-tuning) appear to be limited to smaller networks (e.g., $n=10$). It would be beneficial to discuss the convergence performance and communication overhead in large-scale network settings.

2. Robustness and Topology:The paper already evaluates performance under time-varying topologies and varying connectivity probabilities, as well as data heterogeneity and label noise. I suggest discussing the algorithm's stability under more extreme conditions, such as node failures, random communication link interruptions, or potential Byzantine attacks, to demonstrate its robustness in highly unreliable environments.

---

> ### Author Rebuttal · Authors · 2026-03-30
>
> We sincerely thank the reviewer for their positive remarks regarding the motivation, theoretical analysis, and empirical evaluations of our work. Below is our detailed response to the constructive feedback.
>
> ### **Response to W1 and Q1:**
>
> We appreciate the positive assessment. Since Weakness 1 and Question 1 both concern network scalability, we address them together below.
>
> Evaluating BERT on $n=10$ is a hardware limitation of simulating decentralized networks on one or a few nodes, not an algorithmic flaw. Simulating 100 agents with BERT-base requires maintaining 100 independent model weights, gradients, and optimizer states simultaneously, demanding 300–500 GB of VRAM. This exceeds typical academic computing capacities. When memory footprints per agent are manageable, FAB scales effortlessly. As already shown in **Figure 15**, FAB demonstrates robust convergence in Reinforcement Learning (RL) environments with up to $n=100$ and $n=1000$ agents.
>
> While the full $n=1000$ data hyper-cleaning evaluation is still running, we already validate FAB's scalability on a large $n=100$ network. The table below reports test accuracy (%) at four training stages for FAB and Push-Pull under different node scales (Shared settings: Fashion-MNIST, MLP, 200 epochs, batch size 64, noise rate 0.5, averaged over 10 seeds).
>
> | **Epoch** | **n=10, α=0.5 FAB (Ours)** | **n=10, α=0.5 Push-Pull** | **n=100, α=1.0 FAB (Ours)** | **n=100, α=1.0 Push-Pull** |
> | --- | --- | --- | --- | --- |
> | **50** | **82.55 ±  1.05** | 63.62 ±  6.40 | **70.00 ±  2.32** | 39.50±  4.59 |
> | **100** | **83.55 ±  0.39** | 66.33 ±  5.58 | **74.30 ±  2.15** | 39.30 ± 4.38 |
> | **150** | **83.99 ±  0.54** | 68.71 ±  5.10 | **77.60 ±  1.91** | 42.80 ±  7.10 |
> | **200** | **83.90 ±  0.61** | 67.91±  5.89 | **81.00 ±  2.76** | 45.70 ±  7.39 |
>
> Scaling to $n=100$ inherently degrades performance for all methods. We set $\alpha=1.0$ (weakening data heterogeneity) to strictly isolate this topological challenge. The results confirm two key facts: First, FAB consistently outperforms Push-Pull across all scales. Second, FAB is strictly more robust to network growth; while the baseline's accuracy collapses as $n$ increases, FAB suffers significantly less degradation, maintaining a dominant 81.00% final accuracy.
>
> ### **Response to W2 and Q2:**
>
> We thank the reviewer for this suggestion. We agree that evaluating robustness in unreliable environments is critical for practical decentralized deployments. We address this from both empirical and theoretical perspectives:
>
> **1. Empirical Robustness to Link Failures and Node Dropouts:**
>
> To address communication volatility, we conducted new robustness experiments. Both scenarios share a challenging base configuration: Fashion-MNIST, MLP, 10 nodes, 200 epochs, 40% label noise, Dirichlet non-IID data ($\alpha = 0.1$), highly sparse random graphs ($p = 0.1$), and results averaged over 10 random seeds.
>
> We evaluated two extreme conditions:
>
> - **Scenario A: 30% Link Failure.** Each communication edge is independently dropped with 30% probability at every round.
>
> | **Algorithm** | **Epoch 50** | **Epoch 100** | **Epoch 150** | **Epoch 200** |
> | --- | --- | --- | --- | --- |
> | **FAB (Ours)** | **74.69 ± 7.68** | **78.34 ± 5.93** | **80.04 ± 5.05** | **80.83 ± 4.42** |
> | Push-ASGD | 71.86 ± 7.71 | 76.10 ± 6.62 | 76.82 ± 6.16 | 76.95 ± 5.22 |
> | Push-Pull | 71.60 ± 6.81 | 75.29 ± 6.25 | 77.14 ± 5.88 | 77.89 ± 5.24 |
> | Push-SAGA | 68.31 ± 8.52 | 73.35 ± 6.99 | 75.44 ± 6.66 | 75.60 ± 6.48 |
> | Push-SGD | 71.30 ± 7.76 | 73.63 ± 7.11 | 74.98 ± 6.30 | 76.00 ± 5.82 |
>
> - **Scenario B: 10% Node Dropout.** Each node has an independent 10% probability of going offline at every round.
>
> | **Algorithm** | **Epoch 50** | **Epoch 100** | **Epoch 150** | **Epoch 200** |
> | --- | --- | --- | --- | --- |
> | **FAB (Ours)** | **74.08 ± 8.16** | **77.94 ± 6.16** | **79.57 ± 5.30** | **80.73 ± 4.77** |
> | Push-ASGD | 70.27 ± 7.83 | 75.37 ± 6.86 | 76.54 ± 6.19 | 76.94 ± 5.38 |
> | Push-Pull | 71.45 ± 7.78 | 75.46 ± 6.48 | 76.93 ± 6.55 | 78.01 ± 5.82 |
> | Push-SAGA | 68.29 ± 9.07 | 73.10 ± 7.24 | 75.06 ± 6.66 | 76.19 ± 6.43 |
> | Push-SGD | 70.73 ± 7.44 | 74.36 ± 6.81 | 76.05 ± 7.04 | 76.71 ± 5.80 |
>
> As demonstrated, FAB inherently absorbs these extreme volatilities. Whether facing severe message loss or temporary node dropouts, FAB consistently outperforms all baselines and converges to the highest accuracy (~80.8%), proving strong resilience.
>
> **2. Theoretical Boundary regarding Byzantine Attacks:**
>
> Like standard Push-Pull, FAB relies on exact tracking and linear mixing, so a single malicious agent can disrupt the global consensus manifold. Achieving Byzantine robustness requires replacing the linear mixing matrix with non-linear robust aggregators (e.g., coordinate-wise median, trimmed mean, or Krum) tailored for directed graphs.

---

### Official Review · Reviewer_arcJ · 2026-03-13

**Soundness:** 3
**Presentation:** 3
**Significance:** 3
**Originality:** 3
**Overall Recommendation:** 4
**Confidence:** 3

**Summary:**

This paper introduces FAB, a first-order algorithm for distributed bilevel optimization over time-varying directed networks. Basically, it helps multiple agents solve nested optimization problems, like hyperparameter tuning, while communicating over a changing network. FAB combines the Push-Pull communication method with a penalty-based reformulation, and it only uses first-order gradients, so it’s practical and efficient. The authors prove it converges with rate $O(K^{-2/3}) $and also provide a new convergence result for Push-Pull in the nonconvex case. Experiments show FAB works better than existing methods, especially in dynamic and noisy settings.

**Compliance With Llm Reviewing Policy:**

Affirmed.

**Final Justification:**

After reading the rebuttal, I am increasing my score, while I still have reservations regarding the strong connectivity assumption and the scalability due to the exponential dependence.

**Key Questions For Authors:**

1. The min–max reformulation does not appear to be convex–concave, correct? So whether simple gradient descent-ascent updates are guaranteed to be stable in this setting?

**Strengths And Weaknesses:**

Strength:
1. This paper studies distributed bilevel optimization over time-varying directed graphs, which is interesting and unexplored.
2. This paper establishes a convergence rate for Push-Pull in nonconvex upper-level and strongly convex lower-level settings over time-varying directed graphs.
3. It is applicable to federated learning, meta-learning, and distributed ML systems.

Weakness:
1. In Section 2, the penalty formulation is only a proxy for the original bilevel problem. For a finite $\lambda$, it may introduce bias or additional stationary points. In Line 156, the paper states that $\lambda$ needs to be sufficiently large, but it does not clearly specify how large $\lambda$ should be. Moreover, a large $\lambda$ can amplify consensus errors in the distributed setting. This trade-off is not fully discussed.
2. The paper assumes the graph is strongly connected at every iteration. This is a rather strong requirement for time-varying networks and may not hold in practical dynamic settings. In directed graphs in particular, it is common that one node can reach another, but the reverse path does not exist, which violates strong connectivity.
3. The convergence rate contains a factor $(ab)^{-n}$, which grows exponentially with the number of agents $n$. This raises concerns about theoretical scalability, especially for large networks.  In addition, the requirement $\lambda = O(K^{1/3})$ depends on the total number of iterations $K$, which may not be practical when $K$ is unknown in advance. Moreover, increasing $\lambda$ can amplify consensus errors, making the method potentially sensitive in distributed settings.
4. Since the theory requires $\lambda = \mathcal{O}(K^{1/3})$, it would be helpful to evaluate how sensitive the performance is to different $\lambda$ schedules. I feel increasing $\lambda$ also amplifies the penalty term, which can in turn magnify consensus and tracking errors in the distributed setting. This creates a structural trade-off: a larger $\lambda$ improves the approximation quality of the bilevel formulation, but it may worsen stability and consensus behavior across agents.

---

> ### Author Rebuttal · Authors · 2026-03-30
>
> Below is our detailed response to your constructive feedback.
>
> ### **General Response: The Effect of $\lambda$**
>
> 1) A large $\lambda$, **with a properly chosen small stepsize**, does not amplify consensus or tracking errors in our setting. By algorithmic updates in (6) and (7), it amplifies the **$L$-smoothness constant** of the penalty function in Eq. (4), which requires a smaller stepsize. Once the stepsize is reduced accordingly, both errors remain bounded.
>
> 2) Besides improving the approximation quality of the bilevel formulation, a sufficiently large $\lambda$ makes the objective in Eq. (4) not only strongly concave in $z$, **but also strongly convex in $y$**. This was also recognized and used by Kwon et al. (2023) [1]; see also Eq. (22) in Appendix D.1, where the strong convexity in $y$ defines the unique minimizer $y_\lambda^*(x)$.
>
> 3) Following the reviewer’s suggestion, we added experiments under **the same settings as Table 1**:
>
> | Fixed Penalty ($\lambda$) | 60 | 80 | 100 | 120 | 140 |
> | --- | --- | --- | --- | --- | --- |
> | **Final Cons. Error** | 9.04e-02 | 6.21e-02 | 4.38e-02 | 1.88e-02 | **1.83e-02** |
> | **Final Track. Error** | 8.93e-01 | 6.24e-01 | 4.42e-01 | 1.95e-01 | **1.90e-01** |
> | **Peak Cons. Error** | 1.59e-01 | 1.20e-01 | 9.59e-02 | 7.39e-02 | **5.86e-02** |
> | **Peak Track. Error** | 1.44e+00 | 1.10e+00 | 8.82e-01 | 6.96e-01 | **5.60e-01** |
>
> The table shows that increasing $\lambda$ **decreases** both the final and peak absolute errors.
>
> ### **Response to W1:**
> As indicated at Line 157, Lemma D.1 proves that the proxy hypergradient uniformly approximates the true hypergradient:
>
> $\||\nabla \mathcal{L} _ {\lambda}^{\ast}(x)-\nabla \mathcal{F}^{\ast}(x)\|| \le \frac{\sqrt{\mathcal{C} _ {gap}}}{\lambda}$
>
> Thus, although a finite $\lambda$ may introduce bias or extra stationary points, the required magnitude of $\lambda$ is quantified by the desired accuracy.
> As clarified in the **General Response**, a large $\lambda$ does not amplify consensus errors with a properly scaled stepsize. Hence, the trade-off is not approximation quality versus consensus, but approximation bias (favoring larger $\lambda$) versus optimization complexity (larger smoothness requires smaller stepsizes). Theorem 3.4 resolves this via $\lambda=\mathcal{O}(K^{1/3})$, balancing the $\mathcal{O}(1/\lambda)$ gap and optimization error to yield the $\mathcal{O}(K^{-2/3})$ rate.
>
> ### **Response to W2:**
> As noted in our manuscript, this assumption can be relaxed. Per-iteration strong connectivity is a standard analytical baseline in time-varying directed networks (e.g., Assumption 3.1 and Proposition 6.1 in [2]). As detailed after Assumption 3.2, our analysis extends to the weaker $C$-strongly-connected model, where only the union of graphs over $C$ consecutive iterations is connected, accommodating intermittent links. The proof extension mirrors Remark 6.2 of [2].
>
> ### **Response to W3:**
> (1) Regarding the $(ab)^{-n}$ factor, please refer to our **Response to W for Reviewer jZHp**, where we explain that this exponential dependence is a well-known worst-case bottleneck for time-varying directed consensus [1].
>
> (2) Regarding $\lambda$ and consensus errors, please refer to our **General Response**: with a properly scaled stepsize, a large $\lambda$ does not amplify consensus errors.
>
> (3) Knowing $K$ is not required operationally. **Practically**, Table 1 shows that a fixed $\lambda$ above the structural threshold already works well without knowing $K$. **Theoretically**, one may use $\lambda_k=\lambda_0 k^{1/3}$; by Corollary 4.2 of Kwon et al. (2023) [1], this preserves convergence and keeps the dominant rate $\mathcal{O}(K^{-2/3})$ up to a mild logarithmic factor $\mathcal{O}(K^{-2/3}\log K)$, removing the need for predefined $K$.
>
> ### **Response to W4**:
>
> As shown in Table 1 (Appendix B.4), FAB is robust to fixed $\lambda \in [60, 140]$, demonstrating that any sufficiently large $\lambda$ ensures good performance. As detailed in our General Response, scaling the stepsize controls the penalty term, and increasing $\lambda$ also shrinks the absolute consensus and tracking errors, alleviating stability concerns.
>
> ### **Response to Q1**:
> While not globally convex-concave, the min-max formulation has **two structural properties** guaranteeing convergence:
>
> it is strongly concave in $z$ (from $g$) and, when $\lambda \ge 2L_{f,1}/\mu$, strongly convex in $y$.
>
> As detailed in Lemma D.1, these properties bound the hypergradient approximation, ensuring that standard gradient descent-ascent (GDA) converges to a stationary point defined by the true hypergradient, consistent with recent first-order methods [1].
>
> **References:**
>
> [1] J. Kwon, D. Kwon, S. J. Wright, et al. A fully first-order method for stochastic bilevel optimization. ICML, 2023.
>
> [2] A. Nedić, D. T. A. Nguyen, and D. T. Nguyen. AB/Push-Pull method for distributed optimization in time-varying directed networks. Optimization Methods and Software, 2025.

---

> > ### Author Rebuttal · Reviewer_arcJ · 2026-04-03
> >
> > After reading the rebuttal, I am increasing my score, while I still have reservations regarding the strong connectivity assumption and the scalability due to the exponential dependence.

---

> > > ### Author Response · Authors · 2026-04-04
> > >
> > > Thanks for your timely response.
> > >
> > > Before addressing your reservations, we would like to humbly correct a typo in our initial rebuttal. In our Response to W3(1), the citation [1] inadvertently pointed to Kwon et al. (2023). It was intended to reference Olshevsky and Tsitsiklis (2011) (listed as Reference [1] in our Response to Reviewer jZHp), which establishes the exponential bottleneck for time-varying directed consensus. We apologize for any confusion this may have caused.
> > >
> > > Regarding your reservations, we would like to further clarify the following:
> > >
> > > - **On the strong connectivity assumption:** As clarified in our detailed response, our algorithm does not rely on instantaneous per-iteration connectivity in practice. It fundamentally inherits the flexibility of the $C$-strongly-connected model. In practical networks with intermittent or asymmetric packet drops, accumulating these sparse links over a moderately small window $C$ easily forms directed rings or overlapping cycles that span all nodes. Thus, temporal accumulation naturally guarantees global strong connectivity, making the algorithm highly robust under realistic network volatility. Indeed, our response to W2 and Q2 for Reviewer 5coN provides empirical evidence for this point. Specifically, even under extreme volatility, such as a 30% independent link failure rate or a 10% node dropout probability at each round, FAB still demonstrates strong performance in substantially harsher environments.
> > >
> > > - **On the exponential dependence and scalability:** We completely understand your reservation. We would like to clarify that this term reflects a worst-case mathematical bottleneck for time-varying directed consensus. Specifically, as mentioned in our correction above, Olshevsky and Tsitsiklis (2011) [1] established that the worst-case convergence time inherently scales exponentially with the network size. The underlying mathematical mechanism for this is further detailed by Touri and Nedić (2014) [2] (see Lemma 7 and Corollary 3), who showed that the minimal positive entry in the product of stochastic matrices inevitably decays as $O(\gamma^n)$, where $\gamma$ represents the lower bound of the non-zero elements in the transition matrices. Therefore, this exponential factor represents a fundamental property of the worst-case network topologies in this field. Fortunately, our large-scale empirical results suggest that practical real-world networks naturally avoid these pathological worst-case scenarios, allowing our algorithm to scale quite gracefully in practice.
> > >
> > > **References:**
> > >
> > > [1] A. Olshevsky and J. N. Tsitsiklis. Convergence speed in distributed consensus and averaging. *SIAM Review*, 2011.
> > >
> > > [2] B. Touri and A. Nedić. Product of random stochastic matrices. *IEEE Transactions on Automatic Control*, 2014.

---

### Official Review · Reviewer_jZHp · 2026-03-13

**Soundness:** 3
**Presentation:** 3
**Significance:** 3
**Originality:** 3
**Overall Recommendation:** 4
**Confidence:** 3

**Summary:**

This paper investigates fully first-order distributed bilevel optimization over time-varying directed graphs, where both asymmetric communication and the nested bilevel structure create substantial technical challenges.
The authors propose FAB, a first-order algorithm that combines the Push–Pull communication mechanism with a value-function-based penalty reformulation, thereby avoiding Hessian-vector products in the distributed bilevel setting.
Under nonconvex upper-level objectives and strongly convex lower-level objectives, the paper proves that FAB achieves an $O(K^{-2/3})$ convergence rate in terms of the hypergradient. The paper also derives an $O(K^{-1})$ convergence result for the corresponding single-level Push–Pull method over time-varying directed graphs, which is presented as resolving an open question.
Empirically, FAB is evaluated on distributed hyperparameter tuning, reinforcement-learning policy evaluation, and data hyper-cleaning, where it is reported to outperform several baselines and to be more robust to label corruption, data heterogeneity, and dynamic network effects.

**Compliance With Llm Reviewing Policy:**

Affirmed.

**Final Justification:**

The authors’ responses have adequately addressed my concerns. I would like to keep my positive score.

**Key Questions For Authors:**

- Could the authors clarify the practical communication overhead of FAB? The paper does include communication-cost plots and notes that FAB may incur substantially higher communication (e.g., roughly $2\times$ that of Push-Pull in one experiment, and the highest communication cost among the compared bilevel variants in Figure 12), so I would like to better understand how the authors view this trade-off between improved performance and additional communication.

**Limitations:**

yes

**Strengths And Weaknesses:**

**Strengths**
- The focus on time-varying directed graphs seems practically meaningful.
The paper provides a clear motivation for why such dynamic and asymmetric communication patterns arise in realistic multi-agent systems with delays, stragglers, or changing connectivity.
- The paper makes a substantial technical contribution beyond the bilevel result itself.
In particular, the analysis also yields an $O(K^{-1})$ convergence rate for single-level nonconvex Push–Pull over time-varying directed graphs, which the paper presents as resolving an open question; this seems like an independently meaningful contribution to the distributed optimization literature.

**Weaknesses**
- While the authors do acknowledge this issue, my primary concern is the exponential dependence on the network size in the main theoretical rates, through factors such as $(ab)^{-n}$.
Here, $a$ and $b$ are the uniform lower bounds on the nonzero entries of the row-/column-stochastic mixing matrices, and for sparse time-varying directed graphs, these quantities can be substantially smaller than 1.
As a result, the bound can become extremely pessimistic as $n$ grows, which significantly weakens the practical meaning of the theory.
The paper explicitly notes that linear speedup may not be guaranteed and also lists obtaining a polynomial dependence on $n$ as future work.
Still, to my understanding, this remains the main limitation of the current result.

---

> ### Author Rebuttal · Authors · 2026-03-30
>
> We sincerely thank the reviewer for their positive remarks regarding the practical relevance of our network setting and the theoretical value of our analysis. Below is our detailed response to the constructive feedback.
>
> ### **Response to  W:**
>
> We sincerely appreciate the insightful feedback. We clarify its mathematical origins and the broader field-wide context below:
>
> **1. The Theoretical Origin of the Exponential Dependence:** Without relying on doubly stochastic or weight-balanced assumptions, consensus over general time-varying directed graphs requires bounding the multi-step contraction of time-varying transition matrices. To guarantee network-wide information propagation, the lower bound of these accumulated positive entries decays exponentially with the network size n in the worst-case scenario. As rigorously proven in Theorem 8.1 of Olshevsky and Tsitsiklis (2011) [1], this exponential decay is a fundamental mathematical bottleneck for directed consensus.
>
> **2. Broader Challenges and Empirical Reality:** Overcoming this universally recognized challenge requires either (i) a foundational breakthrough in algebraic graph theory regarding stochastic matrix products, or (ii) circumventing the problem by imposing strictly stronger assumptions (e.g., fixed topologies or instantaneous strong connectivity). Nevertheless, our empirical results (Figure 6(a)) demonstrate significantly more optimistic scaling in practice. As noted in our conclusion, tightening these worst-case theoretical bounds to better reflect practical network dynamics remains a compelling direction for future research.
>
> **Reference:**
>
> [1] A. Olshevsky and J. N. Tsitsiklis. Convergence speed in distributed consensus and averaging. SIAM Review, 2011.
>
> ### **Response to Q:**
>
> We appreciate the reviewer highlighting this trade-off. While FAB incurs a roughly 2x per-iteration communication overhead, the standard metric in distributed optimization is the cumulative communication to reach a target accuracy.
>
> **1. Comparison with Single-Level ITD Baselines:** Following the reviewer’s insightful suggestion, we evaluated the cumulative communication volume required to reach specified test accuracy thresholds over 10 independent runs. The results are summarized in the table below (to be included in the revised manuscript).
>
> *(Note: Values are reported as Mean $\pm$ Std. The notation `(n/10)` indicates the number of successful runs (out of 10) in which the method reached the target accuracy.)*
>
>
> | **Algorithm** | **Acc ≥80% (MB)** | **Acc ≥82% (MB)** | **Acc ≥84% (MB)** | **Acc ≥86% (MB)** |
> | --- | --- | --- | --- | --- |
> | DL-Push-SGD | 51624.0±8555.4 | 90694.8±22475.0 (8/10) | 116554.0 (1/10) | N/A (0/10) |
> | DL-Push-SAGA | **51498.2±8669.7** | 92242.5±18208.8 (7/10) | 114038.4 (1/10) | N/A (0/10) |
> | DL-Push-ASGD | 69469.6±13968.0 | 115835.8±33680.6 (8/10) | 239070.8±71367.2 (2/10) | N/A (0/10) |
> | DL-Push-Pull | 65923.0±10908.4 | 113653.3±25917.7 | 214536.3±87180.5 (3/10) | N/A (0/10) |
> | FAB (Ours) | 57519.2±6487.4 | **85429.4±15474.4** | **172098.0±62980.4** | **334164.8±57454.0** (2/10) |
>
> From the above results, we observe that while DL-Push-SAGA is slightly more communication-efficient in the lower-accuracy regime ($\ge$ 80\%), its performance degrades significantly at higher accuracy targets. In contrast, FAB requires the lowest cumulative communication bandwidth for more stringent accuracy requirements ($\ge$ 82\% and $\ge$ 84\%).
>
> Moreover, reaching the 86% accuracy threshold represents an extreme stress test under this dynamic topology setting. FAB is the only method capable of achieving this target, whereas all baseline methods incur prohibitive communication overhead and consistently fail.
>
> **2. Evaluation Against Potential Variants:**  Following your suggestion, we report the target-error communication costs under the same setting of Figure 12:
>
> | **Algorithm** | **Comm @ Err ≤0.1 (MB)** | **Comm @ Err ≤0.05 (MB)** | **Comm @ Err ≤0.02 (MB)** | **Comm @ Err ≤0.01 (MB)** |
> | --- | --- | --- | --- | --- |
> | Static_FAB | N/A | N/A | N/A | N/A |
> | PushSum_FAB | 142.33 ± 0.004  | 196.38 ± 0.003  | 307.73 ± 0.002  | N/A |
> | PushPull_SOBA | **77.56 ± 0.006** | **90.24 ± 0.005** | **102.91 ± 0.005** | **403.98 ± 0.004** |
> | FAB (Ours) | 110.84 ± 0.004  | 150.42 ± 0.004  | 229.65 ± 0.003  | 405.57 ± 0.004  |
>
> First, as an ablation, Static_FAB fails to reach even the lowest precision, confirming our communication protocol is indispensable here. Second, while PushPull_SOBA uses less communication in the early transient phase ($\text{Err} \le 0.02$), FAB matches its efficiency in the strict high-precision regime ($\text{Err} \le 0.01$). Considering its advantages in wall-clock time and memory efficiency (Figure 12), FAB presents a more favorable overall trade-off.

---

> > ### Author Rebuttal · Reviewer_jZHp · 2026-04-03
> >
> > I have confirmed the rebuttal. The authors’ responses have adequately addressed my concerns.

---

> > > ### Author Response · Authors · 2026-04-04
> > >
> > > Thanks for your timely response. We are glad that your concerns have been addressed.

---

### Decision · Program_Chairs · 2026-04-30

**Decision:**

Accept (regular)

**Comment:**

The paper proposes a novel algorithm for bilevel decentralized learning over directed, time-varying communication topologies, and provides its theoretical convergence guarante. Notably, it also improves the best known convergence rate of the push-pull algorithm in the single-level non-convex setting. The theoretical results are supported by empirical evaluations on training an MLP on Fashion-MNIST and BERT on IMDB.

Reviewers raised several concerns about the convergence rate, in particular its exponential dependence on the number of nodes, as well as the strong assumption that the communication graph is strongly connected at every iteration. Despite these limitations, I believe the proposed algorithm and analysis are novel and relevant to the machine learning community, the algorithm performs well in practice and practically scales well to the large number of nodes and thus the paper worth acceptance.

The authors should incorporate concerns raised by the reviewers into their final version, including discussions (e.g. that practically algorithm scales well with the number of nodes).